# Multimodal cell atlas of the ageing human skeletal muscle

Yiwei Lai[1,2,15], Ignacio Ramírez-Pardo[3,4,15], Joan Isern[4,15], Juan An[1,2,5,6,15], Eusebio Perdiguero[3,4,15], Antonio L. Serrano[3,4,15], Jinxiu Li[1,2,7,15], Esther García-Domínguez[8], Jessica Segalés[3], Pengcheng Guo[1,2,9], Vera Lukesova[3], Eva Andrés[3], Jing Zuo[1,2], Yue Yuan[1,2], Chuanyu Liu[1,2], José Viña[8], Julio Doménech-Fernández[10,11], Mari Carmen Gómez-Cabrera[8], Yancheng Song[12], Longqi Liu[1,2,7], Xun Xu[1,2,7], Pura Muñoz-Cánoves[3,4,13 ✉] & Miguel A. Esteban[1,2,5,9,14 ✉]

Muscle atrophy and functional decline (sarcopenia) are common manifestations of frailty and are critical contributors to morbidity and mortality in older people[1]. Deciphering the molecular mechanisms underlying sarcopenia has major implications for understanding human ageing[2]. Yet, progress has been slow, partly due to the difficulties of characterizing skeletal muscle niche heterogeneity (whereby myofibres are the most abundant) and obtaining well-characterized human samples[3,4]. Here we generate a single-cell/single-nucleus transcriptomic and chromatin accessibility map of human limb skeletal muscles encompassing over 387,000 cells/nuclei from individuals aged 15 to 99 years with distinct fitness and frailty levels. We describe how cell populations change during ageing, including the emergence of new populations in older people, and the cell-specific and multicellular network features (at the transcriptomic and epigenetic levels) associated with these changes. On the basis of cross-comparison with genetic data, we also identify key elements of chromatin architecture that mark susceptibility to sarcopenia. Our study provides a basis for identifying targets in the skeletal muscle that are amenable to medical, pharmacological and lifestyle interventions in late life.

Increased longevity demands new approaches to promote healthy ageing. Owing to its connections with other body tissues, the skeletal muscle is a major determinant of systemic health[1,2]. Accordingly, pronounced loss of skeletal muscle mass and function associated with ageing—termed sarcopenia—is not only a disabling event but also a critical catalysing step in the accelerated degenerating cascade of older people[1]. Sarcopenia often affects individuals aged over 80 years and is more pronounced in locomotor muscles due to their constant exposure to stress[1].

Skeletal muscle comprises large multinucleated myofibres with distinct contractile and metabolic activities (slow twitch/oxidative, also known as type I myofibres; and fast twitch/glycolytic, also known as type II myofibres) controlled by the activity of motoneurons that contact the myofibres at the neuromuscular junction (NMJ)[5,6]. Muscles also contain a variety of less abundant mononucleated cells, including muscle stem cells (MuSCs, satellite cells), fibro-adipogenic progenitors (FAPs), adipocytes, fibroblast-like cells, immune cells, vascular cells and Schwann cells[3,4]. On average, lean muscle mass declines from 50% of the total body weight in young adults to 25% in individuals aged over 80 years[1]. Preservation of muscle mass and function during life requires appropriate interactions of myofibres with the nearby resident cell types[7,8]. Moreover, skeletal muscle has the ability to regenerate due to MuSCs, which are quiescent unless damage occurs[9]. Ageing negatively affects the overall multicellular cross-talk in the skeletal muscle niche as well as the relative cell numbers, and reduces the regenerative ability of MuSCs[9]. However, the underlying mechanisms remain poorly characterized at the molecular level, especially in humans, complicating the development of therapeutic approaches.

Here we aimed to generate a comprehensive transcriptomic and epigenomic cell atlas of the human locomotor skeletal muscle across different age groups and sexes, including individuals aged ≥84 years with signs of sarcopenia.

## Multimodal atlas of human skeletal muscle

To investigate the molecular changes that occur in the human skeletal muscle with ageing, we obtained hindlimb muscle biopsies from 31

[1]BGI Research, Hangzhou, China. [2]BGI Research, Shenzhen, China. [3]Department of Medicine and Life Sciences, Universitat Pompeu Fabra (UPF), Barcelona, Spain. [4]Altos Labs, San Diego Institute of Science, San Diego, CA, USA. [5]Laboratory of Integrative Biology, Guangzhou Institutes of Biomedicine and Health, Chinese Academy of Sciences, Guangzhou, China. [6]School of Life Sciences, Division of Life Sciences and Medicine, University of Science and Technology of China, Hefei, China. [7]College of Life Sciences, University of Chinese Academy of Sciences, Beijing, China. [8]Freshage Research Group, Department of Physiology, Faculty of Medicine, University of Valencia and CIBERFES, Fundación Investigación Hospital Clínico Universitario/INCLIVA, Valencia, Spain. [9]State Key Laboratory for Diagnosis and Treatment of Severe Zoonotic Infectious Diseases, Key Laboratory for Zoonosis Research of the Ministry of Education, Institute of Zoonosis, College of Veterinary Medicine, Jilin University, Jilin, China. [10]Servicio de Cirugía Ortopédica y Traumatología, Hospital Arnau de Vilanova y Hospital de Liria and Health Care Department Arnau-Lliria, Valencia, Spain. [11]Department of Orthopedic Surgery, Clinica Universidad de Navarra, Pamplona, Spain. [12]Department of Orthopedics, The First Affiliated Hospital of Guangdong Pharmaceutical University, Guangzhou, China. [13]ICREA, Barcelona, Spain. [14]The Fifth Affiliated Hospital of Guangzhou Medical University-BGI Research Center for Integrative Biology, The Fifth Affiliated Hospital of Guangzhou Medical University, Guangzhou, China. [15]These authors contributed equally: Yiwei Lai, Ignacio Ramírez-Pardo, Joan Isern, Juan An, Eusebio Perdiguero, Antonio L. Serrano, Jinxiu Li. ✉e-mail: pmunozcanoves@altoslabs.com; miguelesteban@genomics.cn

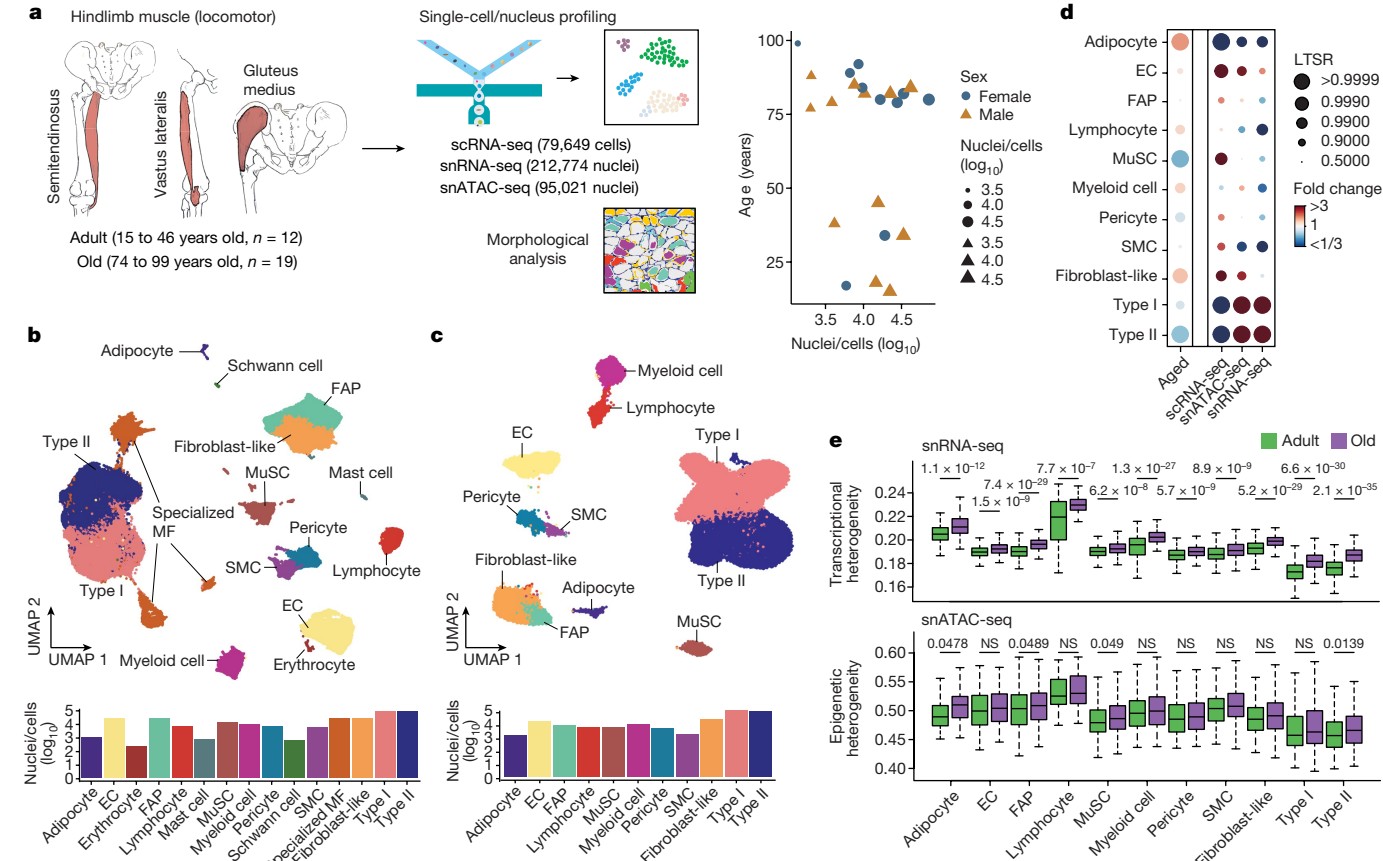

**Fig. 1 | Multimodal human locomotor skeletal muscle ageing atlas.**
**a**, Schematic of the hindlimb skeletal muscle samples analysed in this study. The samples were obtained from 12 adult and 19 older adult (old) individuals (left). The samples were processed for single-nucleus or single-cell isolation for sc/snRNA-seq and/or snATAC-seq library construction (using the DNBelab C4 kit) and sequencing (top middle), or subjected to morphological analysis (bottom middle). Right, the sex, age and profiled nuclei/cells per individual.
**b**, UMAP analysis of 292,423 sc/snRNA-seq profiles delineating 15 main skeletal muscle cell populations (top). Bottom, the number of nuclei/cells sequenced for each cell type. Dots and bars are coloured by cell type. MF, myofibre. **c**, UMAP analysis of 95,021 snATAC-seq profiles delineating 11 main skeletal muscle cell populations (top) at the chromatin level based on gene-activity scores of established marker genes. Bottom, the number of nuclei sequenced for each cell type. Dots and bars are coloured by cell type. **d**, The relative proportional changes of each cell type with ageing (column 1) and each single-cell modality (columns 2–4) considering co-variable factors as sex, ethnicity, omics technology and sequencing batch. The colour scale represents the fold change, and the dot size shows the probability of change (local true sign rate (LTSR)) calculated using a generalized linear mixed model with a Poisson outcome[14]. **e**, Quantification of the transcriptional (top) and epigenetic (bottom) heterogeneity by age group and cell type. $n = 300$ cells obtained by downsampling from the total captured cells in each cell type. For cell types with fewer than 300 cells, all cells were included for analysis. For the box plots, the centre line shows the median, the box limits show the upper and lower quartiles, and the whiskers show 1.5× the interquartile range. For **e**, $P$ values were calculated using two-tailed Mann–Whitney $U$-tests.

participants (17 male and 14 female) from Spain and China, who were divided into two age groups: adults (aged 15 to 46 years, $n = 12$) and older adults (aged 74 to 99 years, $n = 19$) of both sexes, with median ages of 36 and 84 years, respectively (Fig. 1a and Supplementary Table 1). We assessed muscle functionality using (1) the Barthel index, which measures the ability of an individual to carry out daily living activities and their degree of autonomy[10]; and (2) the Charlson index, which predicts life expectancy on the basis of a person's comorbidities[11]. Ageing was inversely and directly correlated with Barthel index and Charlson index scores, respectively, in both sexes (Extended Data Fig. 1a and Supplementary Table 1). Each biopsy was divided into various samples, which were (1) fixed with paraformaldehyde for histology; (2) snap-frozen in liquid nitrogen for single-nucleus RNA-sequencing (snRNA-seq) and single-nucleus assay for transposase-accessible chromatin using sequencing (snATAC-seq); and/or (3) freshly dissociated in single-cell suspensions for single-cell RNA sequencing (scRNA-seq). Morphological analysis confirmed the integrity of the tissue architecture in all cases, and of overt myofibre atrophy in older individuals (Extended Data Fig. 1b). Senescent cells, as determined by senescence-associated β-galactosidase (SA-β-gal) staining, were not detected in the myofibre

area in either adult or older adult muscle samples (Extended Data Fig. 1c). Previous evidence has demonstrated the scarcity of senescent cells in both mouse and human unperturbed muscles[12]. We performed snRNA-seq and snATAC-seq analysis of whole samples, and scRNA-seq analysis of isolated mononucleated cells (Fig. 1a). After quality control, the overall dataset contained 387,444 nuclei/cells corresponding to 22 individuals: 212,774 for snRNA-seq, 79,649 for scRNA-seq and 95,021 for snATAC-seq (Supplementary Table 2).

Uniform manifold approximation and projection (UMAP) visualization of the scRNA-seq and snRNA-seq (sc/snRNA-seq) datasets showed clusters representative of type I and II and specialized myonuclei in the multinucleated myofibre compartment[13]. Within the mononucleated cells, the major muscle-resident cell types were MuSCs, stromal cells (FAPs, fibroblast-like cells and adipocytes), vascular cells (pericytes, smooth muscle cells (SMCs) and endothelial cells (ECs)), immune cells (myeloid, lymphoid and mast cells) and glial cells (Schwann cells)[3,4] (Fig. 1b and Extended Data Fig. 1d). Analysis of snATAC-seq data showed robust identification of the main cell types (Fig. 1c and Extended Data Fig. 1e,f). Integration of the sc/snRNA-seq and snATAC-seq results showed a high correlation, indicating no obvious biases due to method,

age, sex, ethnicity or muscle group (Extended Data Fig. 2a–c). A generalized linear mixed model[14] that considered potentially clinically relevant factors (sex and ethnicity) and technical factors (omics dataset and sequencing batch) revealed age-related decreases in myonuclei, especially in type II myofibres, MuSCs and pericytes, and age-related increases in adipocytes, fibroblast-like cells and immune cells (Fig. 1d). Cell proportion analyses for each individual in all omics datasets depicted similar results irrespective of sex (Extended Data Fig. 2d). These analyses also highlighted that the snRNA-seq and snATAC-seq data are enriched for myonuclei, whereas scRNA-seq mostly captured mononucleated cells. Immunofluorescence validated the progressive changes of MuSCs, FAPs/fibroblast-like cells, adipocytes and immune cells with ageing (Extended Data Fig. 3). Notably, we noticed that most of the cell types showed increased transcriptional heterogeneity among individual cells/nuclei, which is an emerging feature of ageing[15] (Fig. 1e). This was associated with variations in the levels of chromatin accessibility at these loci in the snATAC-seq data, together pointing to increased epigenetic instability that could facilitate cell identity drifts.

## Changes in myonucleus composition

Our collection of human skeletal muscle samples constitutes a powerful resource tool to elucidate the molecular drivers and processes underlying muscle wasting in older people. We first dissected the heterogeneity of myonuclei in different ages by scoring the snRNA-seq data based on known myofibre-type-specific markers[6] (Supplementary Table 3). In addition to *MYH7*+ myonuclei (type I, *TNNT1*+), we identified the two known type II myonuclei (*TNNT3*+) subtypes expressing either *MYH2* (type IIA) or *MYH1* (type IIX), as well as hybrid myonuclei simultaneously expressing two *MYH* genes, across individuals (Fig. 2a and Extended Data Fig. 4a–e). Consistent with previous knowledge[6], ageing induced a general decrease in type II myonuclei accompanied by a relative increase in type I myonuclei in both sexes, which translated into structural changes in the myofibres, as confirmed by immunofluorescence analysis (Fig. 2b and Extended Data Fig. 4f). The decrease in type II myonuclei was more marked for the IIX subtype, followed by hybrid IIA/IIX myonuclei, and the extent of the changes was highly correlated with the age of the individual (Fig. 2b and Extended Data Fig. 4g). We drew similar conclusions after analysing the snATAC-seq dataset.

Further snRNA-seq subclustering identified the presence of myonuclei specialized at the myotendinous junction and the NMJ in both main myofibre types (Fig. 2c and Extended Data Fig. 5a,b). Myotendinous junction myonuclei exhibited enrichment in genes associated with cell–matrix interactions (*COL22A1*, *ADGBR4*), whereas NMJ myonuclei showed enrichment in genes linked to synaptic transmission responses (*PHLDB2*, *CHRNE*). Importantly, subclustering identified other populations enriched in either adult or older adult muscle. For example, *ENOX1*+ myonuclei specific for type II myofibres were enriched in the adult group (median: adult, 9.13%; older adult, 4.21%). By contrast, *TNNT2*+, *ID1*+, *DCLK1*+ and *SAA2*+ myonuclei were enriched in the older group: (1) *TNNT2*+ and *DCLK1*+ myonuclei were primarily present in type I myofibres (*TNNT2*+: adult, 0.08%; older adult, 2.45%; *DCLK1*+: adult, 0.27%; older adult, 2.27%); (2) *ID1*+, in both types of myofibre (type I: adult, 0.17%; older adult, 1.47%; type II: adult, 0.47%; older adult, 0.76%); and (3) *SAA2*+ populations, mainly in type II (adult, 0.10%; older adult, 0.98%) (Fig. 2c,d). Most of these myonuclear subpopulations were also detected by snATAC-seq subclustering and showed the same trend after ageing (Extended Data Fig. 5c). All subpopulations were confirmed by Hotspot analysis[16], which clusters gene expression profiles into modules (Fig. 2e–g, Extended Data Fig. 5d–f and Supplementary Table 4).

Consistent with the protective role of NADPH oxidases in skeletal muscle[17], *ENOX1*+ myonuclei may represent a healthy type II myofibre population, as supported by the high expression levels of genes related to carbohydrate metabolism necessary for fast-twitch contraction (Fig. 2f and Extended Data Fig. 5a,b). Cardiac troponin T

(TNNT2) expression has been associated with denervation and ageing[18]. *TNNT2*+ myonuclei were enriched in genes associated with cardiac muscle contraction (*MYH6*, *TNNT2*), suggesting a loss of skeletal muscle sarcomere specification. TNNT2 expression in older myofibres was confirmed by immunofluorescence analysis (Fig. 2h). *DCLK1* encodes doublecortin-like kinase 1, which is involved in microtubule assembly and dynamics and is highly expressed in dystrophic regenerative (Reg-Myon) myonuclei[19]. ID1 is a transcription factor (TF) involved in BMP signalling that is associated with muscle atrophy in mice[20]. Serum amyloid A2 (encoded by *SAA2*) is a major acute-phase protein that is highly expressed in response to inflammation and chronic tissue injury[21]. *ID1*+, *DCLK1*+ and *SAA2*+ older myonuclei expressed high levels of NMJ-related genes (*CHRNA1*, *CHRNG*, *MUSK*, *COLQ*) and cell adhesion genes, such as members of the *PCDHG* gene family[22] (Fig. 2f and Extended Data Fig. 5a,b), which may indicate a compensatory response for the loss of innervation. These subpopulations were also enriched for stress and pro-inflammatory genes (*FOS*, *JUN*, *EGR1*)[23] and proteolysis genes (*FBXO32*, *CTSD*)[24]. The increased presence of myofibres with signs of denervation in older muscle was validated by immunofluorescence analysis of NCAM1[5] (Extended Data Fig. 5g).

## General and myofibre-type-specific deterioration

To assess the stepwise transcriptional changes that skeletal myofibres undergo with ageing, we first determined the common differentially expressed genes (DEGs) between adult (aged ≤46 years), 'old' (aged 74–82 years) and 'very old' (aged ≥84 years) type I and type II myonucleus populations and performed functional enrichment analysis. The shared effects of ageing in older myonuclei comprised a downregulation of genes related to metabolism, including glucose metabolic processes (*SLC2A4*, *PFKFB1*) and TFs regulating lipid metabolism (*PPARGC1A*, *PPARA*), and sarcomeric genes, such as myosin and troponin genes (Extended Data Fig. 6a,b). There was a strong correlation between an individual's age and the downregulation of expression and chromatin accessibility of sarcomeric genes (Extended Data Fig. 6c and Supplementary Table 5a,b). We also observed a general dysregulation of the circadian machinery in aged myonuclei: core clock genes such as *PER1*, *PER2* and *RORA* were downregulated, whereas *CLOCK* and *BMAL1* (also known as *ARNTL*) were upregulated, consistent with circadian misalignment with ageing[25] (Extended Data Fig. 6b). Although transcriptional changes generally had a good match in the snATAC-seq data, circadian genes did not, indicating regulation at other levels. Other shared effects included upregulation of myofibre-atrophy-related processes, such as protein catabolism (lysosome, autophagy and the ubiquitin-proteasome system) and FOXO signalling[24] (Extended Data Fig. 6a). Moreover, older myonuclei displayed an increased enrichment in TGFβ signalling and homophilic cell adhesion, suggesting an altered interaction with the myofibre environment. Importantly, comparative analysis by age groups revealed that the activation of pro-inflammatory signalling (TNF)[26] was persistently high in the myonuclei of individuals aged ≥84 years. Moreover, we observed a positive correlation with ageing of genes associated with muscle weakness such as increased *PCDHGA1* and *AMPD3*[22,27] transcription and chromatin accessibility, albeit with higher variability at the level of chromatin accessibility (Extended Data Fig. 6c).

To study the directionality of transcriptional variation in the myofibres, we analysed the pseudotime cell trajectories, observing a defined path with ageing in both type I and type II myonuclei (Fig. 3a). The trajectory end points of these myonuclei corresponded to the transcriptional profiles of the new populations that emerged mostly in aged muscle. This trend was also evident when plotting specific skeletal muscle functions (grouped as scores) progressively affected by ageing, such as the sarcomeric apparatus or atrophy-related genes (Fig. 3b, Extended Data Fig. 6d and Supplementary Table 3). Notably, the trajectory of type I myonuclei with ageing was progressive, while that of type II myonuclei

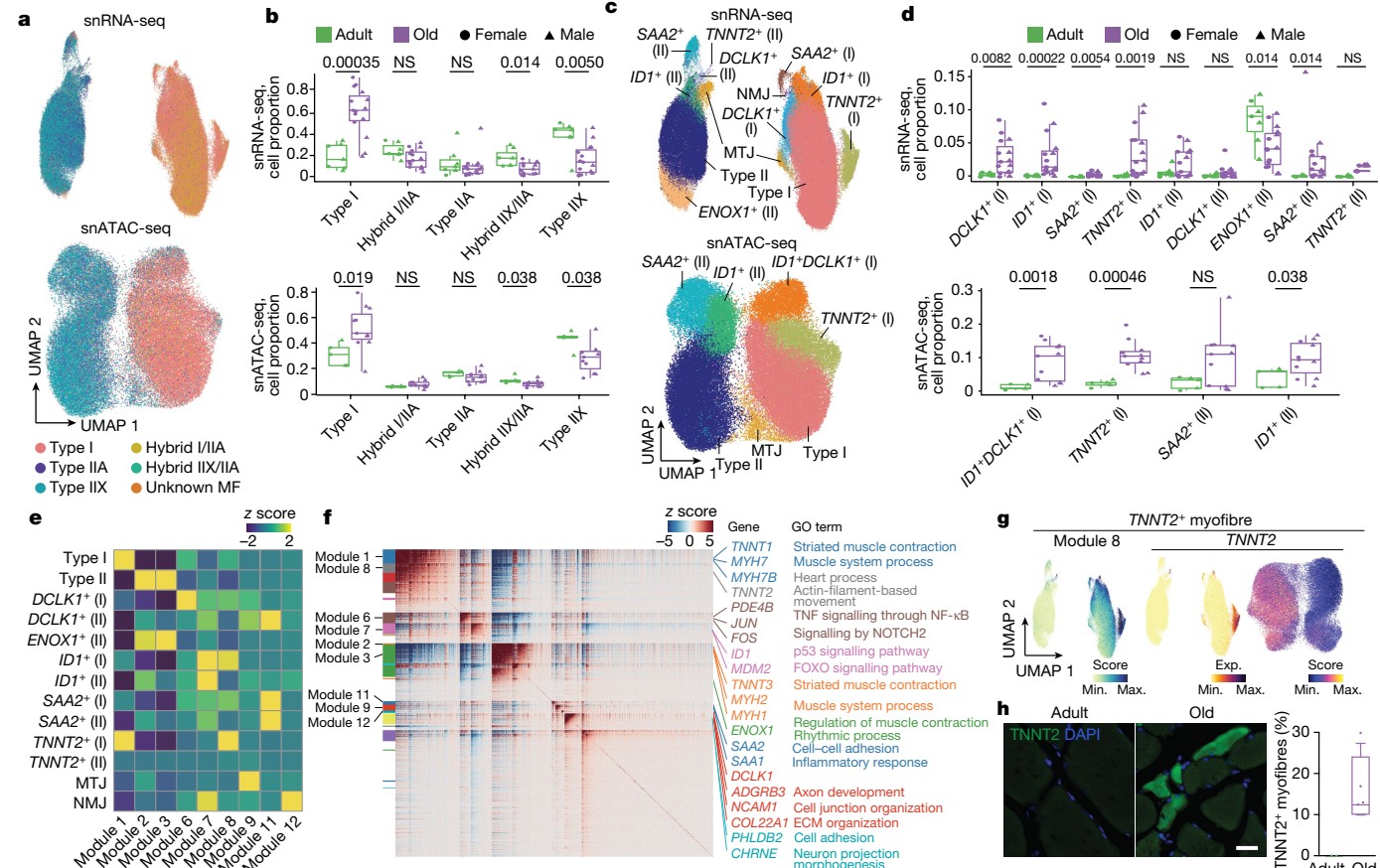

**Fig. 2 | Emergent myonucleus populations with human muscle ageing.**
**a**, UMAP analysis of myonuclei in both snRNA-seq (top) and snATAC-seq (bottom) coloured according to their myofibre-type-specific classification. Each annotated population is correspondingly coloured in both datasets. **b**, Quantification of the myonucleus proportions in adult (green) and older adult (purple) individuals according to the classified myofibre type in snRNA-seq (top) and snATAC-seq (bottom). NS, not significant. For snRNA-seq, $n = 7$ adult individuals and $n = 15$ older adult individuals; for snATAC-seq, $n = 5$ adult individuals and $n = 11$ older adult individuals. **c**, UMAP analysis of myonucleus subpopulations of snRNA-seq (top) and snATAC-seq (bottom) data. Each annotated population is correspondingly coloured in both datasets. MTJ, myotendinous junction. **d**, As in **b**, quantification of the detected myonucleus subpopulation proportions in adult and older adult individuals. **e**, The scaled

aggregated expression levels ($z$ score) in each myonucleus population for the co-expressing genes in each module. **f**, The scaled gene expression level ($z$ score) across co-expression modules. Selected enriched genes and their associated pathways (coloured according to module) are highlighted on the right. GO, Gene Ontology. **g**, UMAP analysis of the aggregated expression (exp.) level for module 8 (left), and *TNNT2* gene expression (middle) and its gene score (right). **h**, Representative images (left) and corresponding quantification (right) of immunofluorescence analysis of *TNNT2*⁺ myofibres (TNNT2, green; nuclei, DAPI, blue) in adult (sample P9) and older adult (sample P29) individuals. Scale bar, 10 μm. $n = 2$ adult individuals and $n = 4$ older adult individuals. For the box plots, the centre line shows the median, the box limits show the upper and lower quartiles, and the whiskers show 1.5× the interquartile range. For **b** and **d**, $P$ values were calculated using two-tailed Mann–Whitney $U$-tests.

was abrupt (Fig. 3a). This difference agrees with the greater sensitivity of type II myofibres to ageing, which results in their preferential loss. By contrast, type I myofibres persist in aged muscle and accumulate progressive damage that further boosts muscle dysfunction over time.

Further analysis of the pseudotime showed ten major clusters of transcriptional variation, most of which reflected the progressive or abrupt course of degeneration in type I or type II myofibres, respectively (Fig. 3c and Supplementary Table 6). For example, the trajectories of the inflammasome (*NFKB1*, *TXNIP*), autophagy (*NBR1*, *ATG7*) and oxidative stress response (*SOD2*, *NFE2L2*) genes increased steadily in type I myonuclei but more sharply in type II myonuclei (cluster 1) (Fig. 3d). Pro-atrophic Notch signalling[28] (*HES1*, *NOTCH2*) increased with a similar trend in both type I and type II myofibres (clusters 1, 2 and 10) with ageing. IL-6 signalling (*IL6ST*, *SOCS3*) was more clearly upregulated along the ageing trajectory in type I myonuclei (cluster 3). Moreover, both myofibre types showed an increased denervation signature (cholinergic synapse; *ITPR1*, *GNG12*) (cluster 10). Consistent with the expression of *DCLK1* in end-stage myonuclei, we also detected myonuclei with the RegMyon repair signature[19] (that is, *MYF6*, *DCLK1*, *MYOG*, *RUNX1*;

Supplementary Table 3) at the end of the trajectory associated with ageing, which emerged progressively in type I myofibres and more abruptly in type II (Extended Data Fig. 6e). This repair program probably arises in response to daily wear-and-tear microdamage in myofibres, which can be fixed by (1) MuSCs[9] or (2) intrinsic myonuclear self-repair mechanisms[29]. However, the chronic presence of this repair signature in aged myonuclei may indicate the persistence of myofibre damage and unsuccessful repair. Consistently, we detected a higher presence of FLNC⁺ scars in aged muscle, which are indicative of ongoing myofibre self-repair[29] (Extended Data Fig. 6f). These findings indicate that aged muscle is not able to cope with daily mild myofibre lesions.

As the largest human tissue, skeletal muscle is the main contributor to whole-body energy expenditure. Mitochondria are crucial for maintaining skeletal myofibre homeostasis and matching energy production through oxidative phosphorylation and fatty acid degradation[30]. The muscle ability to produce energy to sustain contraction substantially reduces with ageing, and defective mitochondria contribute to this phenomenon[30], which we confirmed by succinate dehydrogenase (SDH) activity analysis of myofibres (Extended Data Fig. 6g). Oxidative

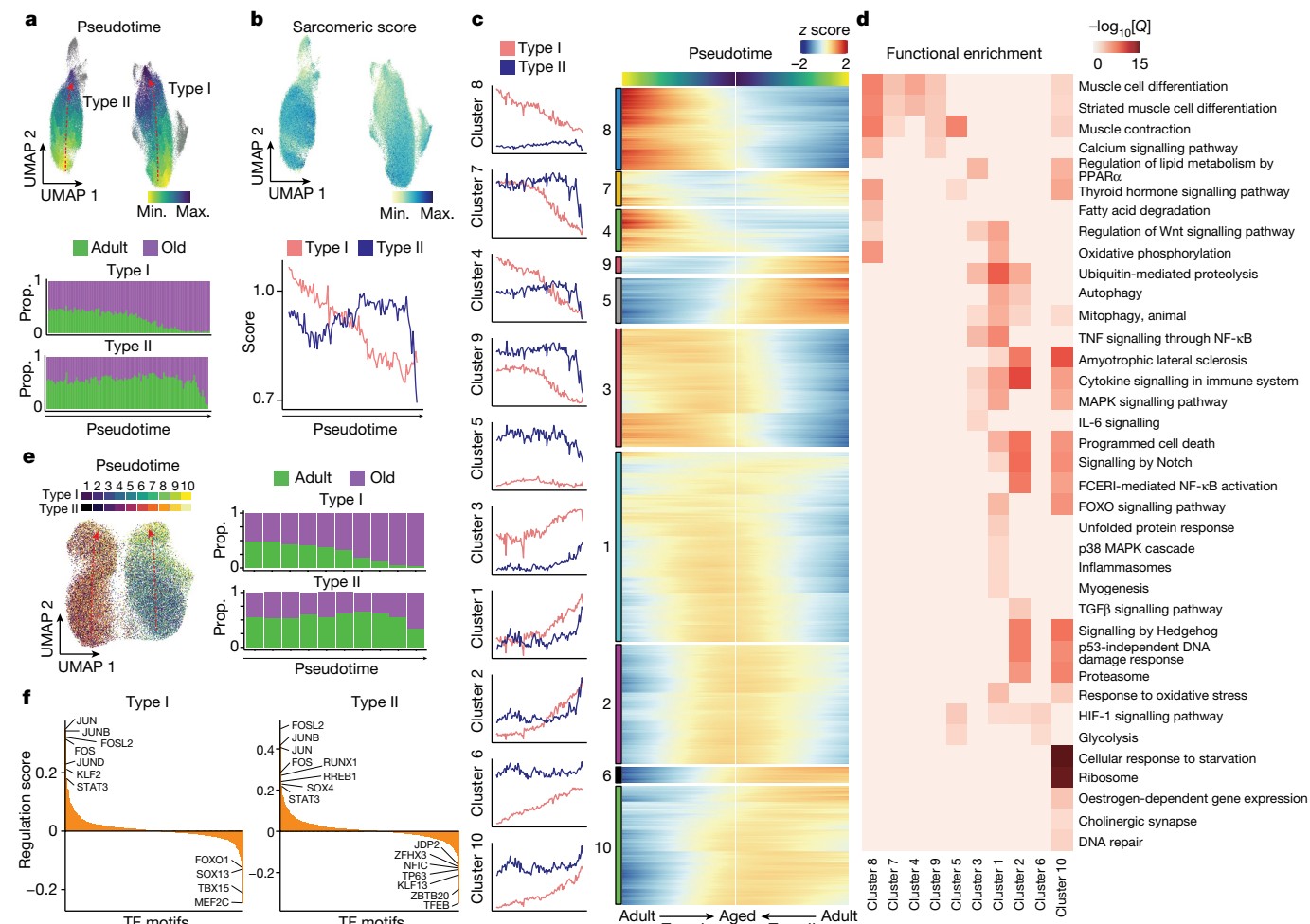

**Fig. 3 | Myonucleus ageing trajectories and GRN. a**, UMAP analysis of the ageing trajectory (pseudotime) (top) for type I and type II myonucleus populations in the snRNA-seq dataset. Dots are coloured by the projected pseudotime. The proportion (prop.) of adult (green) and older adult (purple) myonuclei in snRNA-seq data aligned along the type I (middle) and type II (bottom) myonucleus ageing trajectory (divided into 100 bins). **b**, UMAP analysis of the sarcomeric score for the myonuclei (top) and a line chart showing the average sarcomeric score for type I (red) and type II (blue) myonuclei along the ageing trajectory (bottom). The gene list for sarcomeric score is provided in Supplementary Table 3. **c**, The module score for the gene clusters along the ageing trajectory for type I (red) and type II (blue) myonuclei (left). Right, the corresponding gene expression level (z score). The gene list for each gene cluster is provided in Supplementary Table 6. **d**, Functional enrichment analysis of each gene cluster obtained from **c**. Pathway significance ($-\log_{10}[Q]$) is depicted by the colour scale. A list of genes associated with each pathway is provided in Supplementary Table 6. **e**, UMAP analysis of the ageing trajectory for type I and type II myonuclei in the snATAC-seq dataset, transferred from snRNA-seq data. The ageing trajectory was divided into ten bins (left). The proportion of adult (green) and older adult (purple) myonuclei in snATAC-seq aligned along the type I (top right) and type II (bottom right) pseudotime trajectory. **f**, The mean regulation score ($\log_{10}$-transformed) across all DORCs using the FigR[31] approach per TF for type I and type II myonuclei along the ageing trajectory. The regulation score (y axis) discerns between TF activators (positive score) and repressors (negative score) for the mapped TF motifs.

phosphorylation (*IDH2*, *MDH1*) and fatty acid degradation (*ACADM*, *ACAT1*) were downregulated in the type I myonuclei ageing trajectory (cluster 8) (Fig. 3d and Extended Data Fig. 6h). Unexpectedly, glycolysis (*PKM*, *HK1*) was upregulated (cluster 6) in type I myonuclei at the late stage of the ageing trajectory (Extended Data Fig. 6i). This may reflect a compensatory rearrangement in slow myofibres to prevent the loss of muscle capacity and produce energy during sustained contractions, which is substantially impaired in older individuals[30].

## GRNs in ageing myonuclei

To examine the *cis*-regulatory landscape of myonucleus degeneration, we defined the underlying gene regulatory networks (GRNs) using functional inference of gene regulation (FigR)[31]. We first integrated the snRNA-seq and snATAC-seq datasets using canonical correlation analysis (CCA), identifying the most likely paired nuclei using a constrained optimal cell mapping approach within the common CCA

space (Extended Data Fig. 7a). This yielded a consistent progressive or abrupt cell trajectory towards the ageing state for type I or type II myonuclei, respectively, in snATAC-seq (Fig. 3e). Using these paired nuclei, we linked the snATAC-seq peaks to their target genes based on the peak-to-gene accessibility correlation. This identified a significant association of 28,193 unique chromatin accessibility peaks with 10,707 genes (permutation $P < 0.05$) in type I myofibres, and 27,901 peaks with 10,669 genes in type II myofibres. We defined the high density of the peak–gene interaction subset as domains of regulatory chromatin (DORCs) ($n > 6$ significant peak–gene associations; type I, 8,370 peaks, 908 genes; type II, 7,879 peaks, 912 genes) (Extended Data Fig. 7b). The list of DORC-associated genes included several stress TFs (*JUND*, *JUN*, *FOS*, *JUNB*, *EGR1*) in both type I and type II myofibres. Notably, these gene loci opened their chromatin before the increase of gene expression, indicating a priming process and a stepwise transition to overt myonucleus degeneration (Extended Data Fig. 7c). We next computed TF motif enrichments, considering both the expression patterns and

the chromatin accessibility for all DORCs, to generate the regulation score representing the intersection of motif-enriched and RNA-correlated TFs. We distinguished dozens of putative transcriptional activators and repressors in type I or type II myonuclei along the ageing trajectory (Fig. 3f). Among these, we observed that stress-related TFs (FOSL2, JUN, FOS, JUNB, STAT3) become upregulated and drive a coordinated gene expression program along the degeneration trajectory; this was confirmed by TF footprinting analysis (Extended Data Fig. 7d). We also noticed that some of the enriched transcriptional repressors were myofibre-type-specific; for example, type I myofibres were enriched for TBX15, and type II myofibres were enriched for JDP2. These two TFs have been implicated in glycolytic myofibre metabolism[32] and cardiomyocyte protection by inhibiting AP-1 complex activity[33], and their enrichment could account for myofibre-type-specific dysfunction in old age.

## MuSC exhaustion with ageing by premature priming

The ability of MuSCs to transition on injury from their quiescent state to an activated state for tissue repair is substantially reduced with age[9] but the underlying mechanisms are poorly understood. A major confounding factor for assessing MuSCs using single-cell isolation protocols is that tissue dissociation induces stress[34]. To overcome this, we focused on snRNA-seq data (Extended Data Fig. 8a and Supplementary Table 3).

We identified three MuSC subpopulations expressing the TF *PAX7* and one expressing *MYOG*. *PAX7*+ MuSCs were subclassified in different states according to previously defined markers[4,9]: calcitonin receptor (*CALCR*) for MuSCs in deep quiescence (qMuSCs); *MYF5* for MuSCs early primed for activation (epMuSCs); and modest levels of *MKI67* for MuSCs primed for late activation/proliferation (lpMuSCs) (Fig. 4a and Extended Data Fig. 8b,c). epMuSCs were enriched for *FOS*, *JUN* and *EGR*, which have been described to allow a rapid MuSC exit from quiescence after muscle injury[34]. As *MYOG*+ MuSCs were enriched for myogenic differentiation genes (*ACTC1*), we termed them differentiating MuSCs (dMuSCs). All MuSC subtypes except for the scarce lpMuSCs could also be discerned in snATAC-seq.

Wider DEG and functional enrichment analysis showed that qMuSCs were enriched for extracellular matrix (ECM)-remodelling genes (*FBN1*, *VIT*, *COL5A2* and *CALCR*) and hormone nuclear receptors (*ESRRG*, *GHR*). This is consistent with the knowledge that the collagen V/CALCR axis and hormones help to maintain the pool of qMuSCs in mice[9] (Extended Data Fig. 8d). Accordingly, snATAC-seq peaks in qMuSCs showed substantial enrichment in binding motifs for TFs related to growth hormone regulation[35] (PGR, NR3C1) in addition to myogenic functions (NFIC)[36] (Extended Data Fig. 8e). epMuSCs were enriched for inflammation-related genes (cytokine and TNF signalling), cell growth (MYC targets) and autophagy. Moreover, they were enriched for binding motifs of the cofactor SMARCC1, FOS and JUNB, indicating higher readiness for activation, which is similar to in mice[34]. In addition to proliferation-related genes, lpMuSCs expressed genes involved in chromatin organization (*DNMT1*, *HELLS*, *EZH2*). In agreement with their gene expression pattern, dMuSCs had a higher enrichment of binding motifs for MYOG. Enhanced binding for JUNB and MYOG in epMuSCs and dMuSCs, respectively, was confirmed by footprinting analysis (Extended Data Fig. 8f).

Although MuSC heterogeneity persisted with ageing, there was an increase in the proportion of epMuSCs in older muscle (Fig. 4b and Extended Data Fig. 8g). In mice, a break of quiescence induced by changes in the niche accounts for the loss of MuSCs with age[9]. Thus, an increase in epMuSCs may be partly responsible for the loss of MuSCs in older human muscle. We confirmed the prevalence of FOS+ MuSCs transiting to a primed state in older muscle using immunofluorescence (Extended Data Fig. 8h). Pathways related to MuSC stemness, such as FOXO signalling for qMuSCs[9], and proliferative capacity, such as translation for epMuSCs and cell cycle for lpMuSCs[9], were diminished

with ageing (Fig. 4c). All aged MuSC subtypes except for epMuSCs displayed enhanced mitochondrial oxidative phosphorylation. A detailed analysis of qMuSCs from adult and older adult groups showed that downregulation of ECM-related processes (*ITGBL1*) is progressively associated with ageing, whereas upregulation of myogenesis (*MEF2D*) peaked in qMuSCs from adults aged 74–82 years, and inflammatory and stress pathways (TNF/NF-κB and NFAT–JUN–FOS) peaked in qMuSCs from adults aged ≥84 years (Fig. 4c, Extended Data Fig. 8i,j and Supplementary Table 5c). snATAC-seq analysis showed that older qMuSCs are enriched for binding motifs of TFs that regulate advanced myogenic stages, such as differentiation-related (NFYA, NFYB, NFYC)[37] and stress response (ETS2, EGR1)[34] TFs (Fig. 4d). Conversely, motif enrichment of growth-hormone-related TFs (PGR, NR3C1, AR) in qMuSCs was lost with ageing. These findings suggest exhaustion and inability to respond to muscle injury or homeostatic body signals.

## Pro-inflammatory and profibrotic responses

We next performed subclustering of mononucleated cells in the sc/snRNA-seq datasets. These resident cell types are not only crucial for overall skeletal muscle homeostasis but also support the regenerative activities of MuSCs after injury.

Within the vascular compartment, we identified four subtypes of ECs: (1) arterial ECs that express *SEMA3G*; (2) capillary ECs that express *CA4*; (3) venous ECs (venECs) that express *ACKR1*; and (4) a subpopulation of venECs that express *IL6* (*IL6*+ venECs) (Fig. 4e and Extended Data Fig. 9a). We also identified three subtypes of mural cells: (1) SMCs that express *ACTA2* and *MYH11*; (2) pericytes that express *HIGD1B* and *RGS5*; and (3) mural cells that express *CD44*. snATAC-seq analysis confirmed the same cell types (Extended Data Fig. 9b–d). In older muscle, the proportion of capillary ECs and pericytes decreased, while that of arterial ECs and venECs increased (Fig. 4f and Extended Data Fig. 9e). Vascular cell types downregulated genes related to cell junction assembly and transmembrane transporter activity with ageing, and upregulated inflammatory (IL-6 and AP-1 pathways), fibrotic (TGFβ pathway) and autophagy pathways (Fig. 4g). We concluded that ageing alters the skeletal muscle vascular integrity by increasing pro-inflammatory and stress-related signals.

Among the immune cells, we identified different subpopulations of myeloid cells and lymphocytes from sc/snRNA-seq and snATAC-seq data, including *CD14*+ and *FCGR3A*+ (CD16) monocytes that are endowed with distinct responses to different pathogens and stimuli[38]; macrophages (lipid-associated macrophages (LAMs) and *LYVE1*+ macrophages) with yet-to-be characterized distinctive functions in skeletal muscle[3]; mast cells; dendritic cells; B cells (naive and memory); natural killer (NK) cells; NK T cells; *CD4*+ T cells (effector *CCR7*−, naive *CCR7*+ and regulatory *IL2RA*+); *CD8*+ (effector *CCR7*− and naive *CCR7*+); and a group of *CCL20*+ T cells[3] (Fig. 4h and Extended Data Fig. 9f–i). Consistent with the increased inflammatory cell infiltration shown by histological analysis (Extended Data Fig. 3d–f), mast cells, LAMs and monocytes increased in older muscle, while some of the T cell subtypes and dendritic cells decreased (Fig. 4i and Extended Data Fig. 9j). Activated mast cells in skeletal muscle have been associated with cancer-induced muscle atrophy (cachexia)[39]. All of the immune cell subpopulations, except for mast cells, downregulated homeostatic immune functions with ageing, including antigen processing and presentation (MHC pathway, B cell receptor signalling and TCR signalling) (Fig. 4j). Similarly, anti-inflammatory responses were downregulated in some immune cell types (signalling by ERBB4 in lymphoid cells), while pro-inflammatory ones were upregulated in others (IL-6 pathway in myeloid cells, complement activation and signalling by NTRK1 in myeloid cells and lymphocytes). Moreover, older immune cells were enriched for processes associated with phagocytosis (protein processing in the endoplasmic reticulum, clathrin-mediated endocytosis and

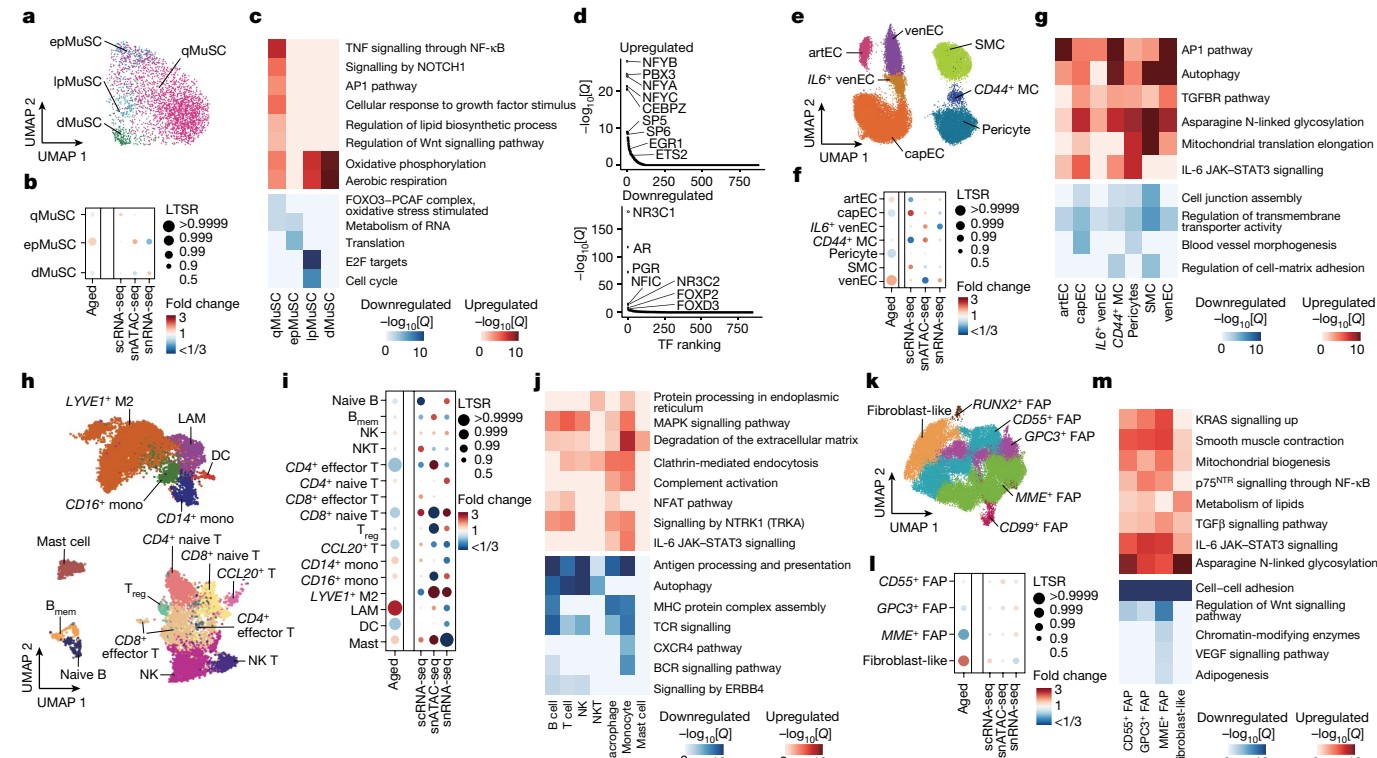

**Fig. 4 | Resident mononucleated populations in the human skeletal muscle with ageing. a**, UMAP analysis of the detected MuSC subpopulations in snRNA-seq data. Dots are coloured according to cell type. **b**, The relative proportional changes of each MuSC subpopulation with ageing (column 1) and each single-cell modality (columns 2–4), considering co-variable factors (ethnicity, omics technology and sequencing batch). The colour scale represents the fold change, and the dot size shows the probability of change (LTSR) calculated using a generalized linear mixed model with a Poisson outcome[14]. **c**, Functional enrichment analysis of DEGs obtained between adult and older adult groups for each MuSC subpopulation. The colour scale represents the significance ($-\log_{10}[Q]$) of the enriched terms for upregulated (red) and downregulated (blue) genes with ageing. **d**, TF motif enrichment for upregulated (top) and downregulated (bottom) peaks in qMuSCs with ageing (older adult versus adult). TFs were plotted according to their rank (*x* axis) and their associated $-\log_{10}[Q]$ (*y* axis). **e**, UMAP analysis as in **a** for vascular cell subpopulations. artEC, arteriole EC; venEC, venule EC; capEC, capillary EC; MC, mural cell subpopulations. **f**, The relative proportional changes as in **b** for vascular cell subpopulations. **g**, Functional enrichment analysis of DEGs (older adult versus adult) as in **c** for vascular cell subpopulations. **h**, UMAP analysis as in **a** for immune cell subpopulations. $B_{mem}$, memory B cells; DC, dendritic cells; M2, M2-like macrophages; mono, monocytes; $T_{reg}$, regulatory T cells. **i**, The relative proportional changes as in **b** for immune cell subpopulations. **j**, Functional enrichment analysis of DEGs (older adult versus adult) as in **c** for immune cell subpopulations. **k**, UMAP analysis as in **a** for stromal cell subpopulations. **l**, The relative proportional changes as in **b** for stromal cell subpopulations. **m**, Functional enrichment analysis of DEGs (older adult versus adult) as in **c** for stromal cell subpopulations.

degradation of the ECM), pointing to a predominantly activated status. Thus, in addition to a general increase in infiltrating immune cells with ageing, there is a switch towards a pro-inflammatory state, consistent with inflammation being a key driver of ageing[40].

Within the stromal cell compartment, we identified various subtypes of fibro-adipogenic cells (FAPs), including *CD55*[+], *CD99*[+], *GPC3*[+], *MME*[+] and *RUNX2*[+], and fibroblast-like cells expressing *THY1* (Fig. 4k and Extended Data Fig. 10a). snATAC-seq analysis confirmed the predominance of fibroblast-like cells, *CD55*[+], *MME*[+] and *GPC3*[+] FAPs in the muscle stroma (Extended Data Fig. 10b–d). Fibroblast-like cells substantially increased with ageing, whereas *MME*[+] FAPs were diminished (Fig. 4l and Extended Data Fig. 10e). *MME*[+] FAPs are a well-known dominant FAP subtype[41] and expressed genes related to adipogenic pathway, whereas *CD99*[+] and *GPC3*[+] FAPs were pro-inflammatory FAPs expressing *CCL2* and *CXCL14* (Extended Data Fig. 10a,f). Fibroblast-like cells and *CD55*[+] FAPs showed higher fibroblast activation traits (epithelial–mesenchymal transition, ECM organization) compared with other FAP subtypes[42] (Extended Data Fig. 10a). *RUNX2*[+] FAPs were enriched for *SOX5* and are involved in migration and collagen production[43]. Older FAP subtypes largely shared an ageing signature characterized by the downregulation of growth factor pathways (VEGF and Wnt) and upregulation of profibrotic (TGFβ signalling) and pro-inflammatory (IL-6 signalling) pathways and asparagine N-linked glycosylation (Fig. 4m). These results point to a shift in the stromal populations (especially *CD55*[+] and fibroblast-like cells) towards an activation state characterized by active ECM remodelling.

Importantly, comparative analysis by age groups (individuals aged 15–46 years; 74–82 years; and ≥84 years) revealed that these muscle-resident cells (vascular, immune and stromal) displayed a peak enrichment of pro-inflammatory pathways (IL-6/AP-1 pathway) in the group aged 74–82 years, and of profibrotic pathways (TGFβ signalling) in the group aged ≥84 years (Extended Data Fig. 10g). These non-myogenic populations, particularly the lymphocytes, also presented a moderate increase in the cell cycle inhibitor genes *CDKN1A* (p21) and *CDKN1B* (p27) (Extended Data Fig. 10h).

## Altered intercellular communication

Cells within a tissue communicate with each other through elaborated circuits[44]. How intercellular cross-talk in the human skeletal muscle niche is affected by ageing is largely unclear. To study this in an integrative manner, we used CellChat[45].

Ligand–receptor interactions involved more dominantly mononucleated cells than myofibres, and the total number of interactions

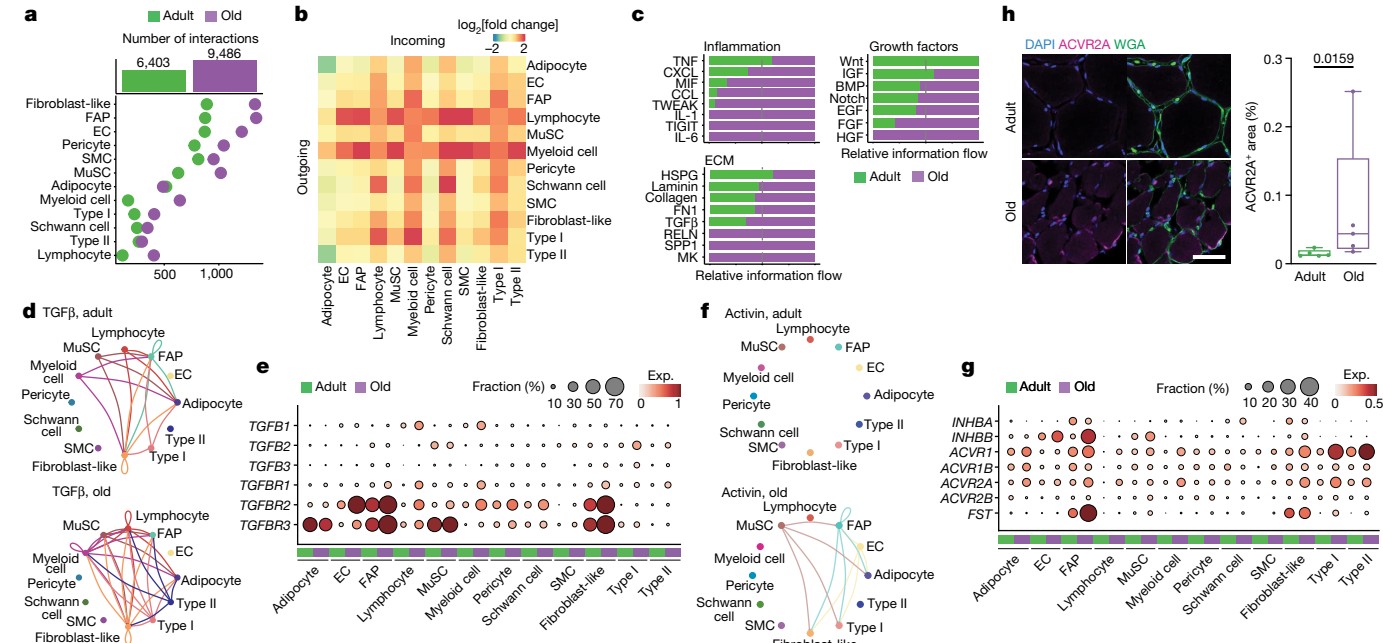

**Fig. 5 | Interactome analysis of skeletal muscle cellular components. a**, The number of predicted interactions (L–R pairs) for each cell type in the adult (green) and older adult (purple) age groups. **b**, The fold change (log₂-transformed, colour scale) with ageing in the number of sent signals (outgoing, horizontal side) and received signals (incoming, vertical side) for each cell type. **c**, The sum of the interaction probability differences (relative information flow) among all pairs for each depicted group of interactions in the adult (green) and older adult (purple) age groups. Interactions are grouped according to the following categories: inflammation (top left), ECM (bottom left) and growth factors (right). **d**, The TGFβ signalling network in adult (top) and older adult (bottom); nodes represent cell types and edges represent the interactions among them. The edge width is proportional to the interaction probability. **e**, The expression levels of the genes associated with TGFβ signalling pathway in adult (green) and older adult (purple) muscles. The colour scale represents the average gene expression, and the dot size shows the percentage of cells expressing a given marker within the subpopulation. **f**, Signalling network, as in **d**, for the activin signalling pathway. **g**, The expression levels as in **e** for the activin signalling pathway and muscle-atrophy-related genes. **h**, Representative images (left) and corresponding quantification (right) of immunofluorescence analysis of ACVR2A⁺ area (ACVR2A, magenta; cell membrane, WGA, green; nuclei, DAPI, blue) in adult (sample P5) and older adult (sample P28) individuals. Scale bar, 50 μm. *n* = 5 individuals for each age group. *P* values were calculated using two-tailed Mann–Whitney *U*-tests. For the box plots, the centre line shows the median, the box limits show the upper and lower quartiles, and the whiskers show 1.5× the interquartile range.

nearly doubled with ageing (Fig. 5a). Interactions involving myeloid and lymphoid cells—and, to a lesser extent, FAPs, fibroblast-like cells and type I myofibres—increased more substantially with ageing compared with those of other cell types (Fig. 5b and Supplementary Table 7). On the basis of these results, we focused on three interaction categories as potential effectors of the muscle-wasting process caused by ageing: inflammation, ECM and growth factors (Fig. 5c).

Although the transient concurrence of immune cells is required for efficient muscle repair, their persistence and the subsequent chronic inflammation is a major driver of dysfunction in aged muscle[40]. Among the inflammation-related communication networks, we observed enhanced secretion of chemokines and cytokines (TNF, CXCL and CCL family members, MIF, IL-1, IL-6) by immune and stromal cells, acting on a variety of cell types including the myofibres (Extended Data Fig. 11a–e). For example, FAPs in older muscle expressed high levels of *CXCL12*, which is a strong chemoattractant for immune cells[3], suggesting the existence of inflammatory–fibrogenic feedback loops. Likewise, increased *IL6R* in myonuclei and *IL6ST* in other cell types with ageing may stimulate myofibre atrophy[8]. *TNF* was reduced with ageing in immune cells, but its receptors (*TNFRSF1A/B*) increased in different cell types.

Excessive ECM deposition, especially of collagen, perturbs skeletal muscle function and is a major hallmark of sarcopenia[8]. Sirius Red staining confirmed extensive fibrosis in older muscle as compared to in adult muscle (Extended Data Fig. 11f). This phenomenon, and the subsequent expansion of the derived interactions with ageing, is consistent with the increase in FAPs and fibroblast-like cells, which are the major ECM

producers in the skeletal muscle[42]. Indeed, we observed an increase in most collagens and fibronectin (*FN1*) in FAPs or fibroblast-like cells, as well as in Schwann cells (Extended Data Fig. 11g). Conversely, there was a downregulation of laminin components in FAPs, fibroblast-like cells and SMCs, concomitant with the downregulation of adhesion molecules (*ITGA7*) in myofibres. This suggests a reduction in the basal lamina causing impaired vascular integrity. Immunofluorescence confirmed the exacerbated reduction of ITGA7 with ageing (Extended Data Fig. 11h). Coinciding with the altered ECM composition, the major profibrotic factor TGFβ increased with ageing, produced mainly by immune cells (*TGFB1*), MuSCs (*TGFB2*) and type I myofibres (*TGFB2*, *TGFB3*), and acting through its receptors (*TGFBR1*, *TGFBR2*, *TGFBR3*) on a variety of cell types, in particular FAPs, fibroblast-like cells, adipocytes and ECs (Fig. 5d,e).

Among the growth factors implicated in muscle mass control, we observed a dysregulation of signalling mediated by activins[46,47], IGF[46], BMP[7], Notch[28] and Wnt[48] factors (Extended Data Fig. 12a). Proatrophic activin signalling[46] was upregulated with ageing, with activin receptors (*ACVR1*, *ACVR1B*, *ACVR2A*, *ACVR2B*) upregulated in myonuclei, and the activin ligand *INHBB* in ECs, FAPs, MuSCs and fibroblast-like cells (Fig. 5f,g). The increased expression of ACVR2A was validated by immunofluorescence analysis (Fig. 5h). Notably, there was an increase in follistatin (*FST*) in FAPs, probably to counteract the proatrophic effects of activin signalling[46]. Higher levels of the Notch ligand *DLL4* in ECs and the *NOTCH2* receptor in older myofibres (Extended Data Fig. 12b) may be related to the recently described EC–myofibre cross-talk in mice[28]. Moreover, *IGF1* increased in FAPs, fibroblast-like and myeloid

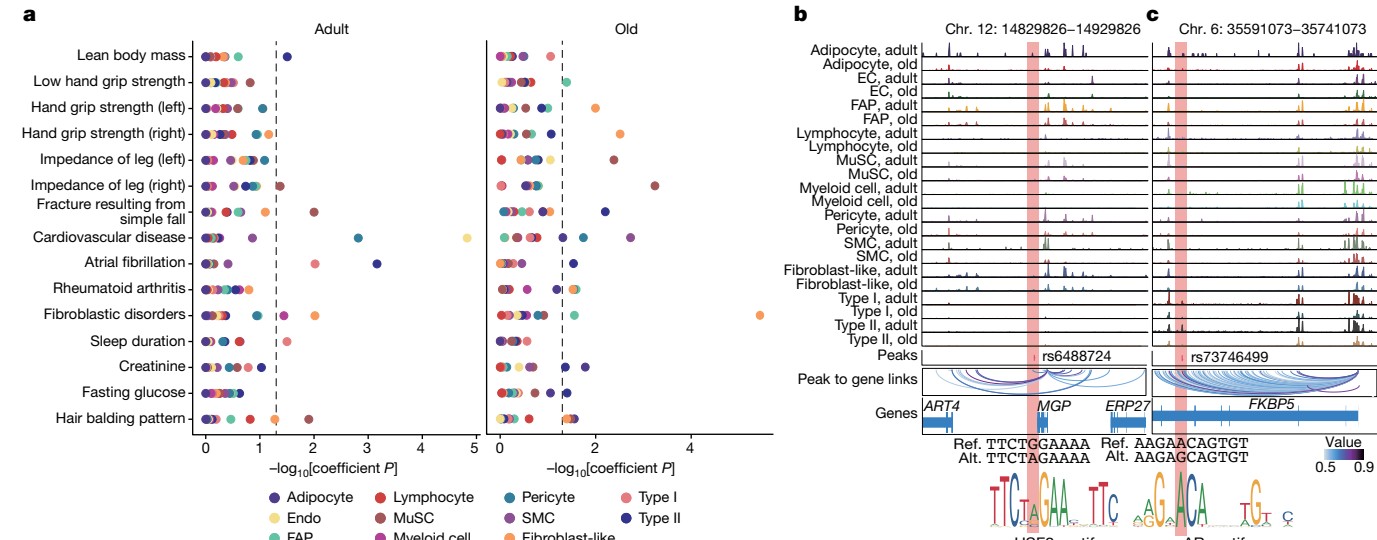

**Fig. 6 | Interpretation of genetic variants related to sarcopenia. a**, Differential enrichment (−log₁₀[$P$]) for complex traits obtained by LDSC analysis of the snATAC-seq peaks mapped within each cell type between adult (left) and older adult (right) age groups. Dots are coloured by cell type. **b,c**, Genome browser tracks (top) showing the normalized aggregate signals associated to the genetic variant rs6488724 at the *MGP* locus (**b**) and rs73746499 at the *FKBP5* locus (**c**) for each cell population (rows) in adult and older adult muscles. Obtained peaks at these loci were linked to corresponding genes. The association between peaks and genes is represented by the colour scale. The reference sequence (ref.) and the altered sequence (alt.) of the genetic variants and the motifs for HSF2 (gain of binding) and AR (loss of binding) are shown at the bottom.

cells but decreased in MuSCs, whereas *IGF2* decreased in ECs and FAPs, suggesting a differential downstream cascade of IGF signalling with ageing. Likewise, *BMP4* decreased in stromal cells, while *BMP5* increased in stromal, pericyte and Schwan cell populations. The shift of BMP ligands and the downregulation of hypertrophy-promoting BMP receptor (*BMPR1A*)[46] in type I and type II myofibres with ageing is probably involved in the loss of muscle mass with ageing (Extended Data Fig. 12c,d). We also observed a reduction with ageing in *WNT9A* expressed mainly by type II myonuclei (Extended Data Fig. 12e) and acting on a variety of cell types, in particular stromal cells. The Wnt pathway amplifiers *LGR4* and *LGR5* were differentially expressed in type II and type I myofibres, respectively, as reported in monkeys[49], and decreased with ageing. Considering that WNT9A regulates NMJ development[48], it is conceivable that alterations in this pathway result in abnormalities of NMJ myonuclei and muscle mass with ageing.

## Linking inherited risk variants to cell types

Recent genome-wide association studies (GWASs) have revealed susceptibility loci associated with muscle weakness[50]. Correlations between candidate loci and susceptibility to sarcopenia in those reports reinforced the direct and indirect functional links of skeletal muscle with other body systems. Our integrated dataset provides a valuable opportunity for interpreting the functional impact of these risk variants at the cellular level. By aggregating fragments from all nuclei across cell types and age groups, we generated a union peak set containing 636,363 peaks, from which we identified 93,565 peaks enriched in individual cell types from all of the tested individuals (Extended Data Fig. 13a). Adult and older adult individuals showed similarities and differences in the openness of these peaks, highlighting that epigenetic alterations are probably an important driver of muscle ageing and sarcopenia. To determine whether cell-type-specific accessible regions in the snATAC-seq data were enriched in GWAS variants for muscle strength and other phenotypes related to muscular diseases or metabolic function[50], we performed linkage disequilibrium (LD) score regression (LDSC) analysis (Fig. 6a and Supplementary Table 8a). For example, whereas lean body mass was enriched in type II myonuclei as

expected, muscle-strength-related traits were unexpectedly enriched in aged fibroblast-like cells and FAPs but not in myofibres, supporting the idea that genetic variations can promote sarcopenia by altering intercellular communication networks. Impedance of leg was highly related to MuSCs in older people, and fracture resulting from a simple fall was associated with adult MuSCs and older type II myonuclei. Moreover, we noticed that sleep duration, creatinine and fasting glucose were related to myofibres, pointing to a potential role of these cell populations in body-level circadian rhythm regulation and metabolic regulation.

As a proof of principle, we prioritized variants taken from low hand grip strength traits[50] and lean body mass traits[51] using a multitiered approach[52] (Extended Data Fig. 13b). We overlapped lead variants and variants with high association (LD $r^2 > 0.8$) with cell-type-specific peaks, identifying 3,158 candidate variants (Extended Data Fig. 13c and Supplementary Table 8b–f). Among others, we found rs1862574 in the *GDNF* locus in myofibres, which may affect muscle innervation. We also observed rs3008232 in *TRIM63* (MuRF1), rs1281155 in *ANGPL2* and rs571800667 in *FOXO1*, which are critical drivers of muscle atrophy[24,53]. We next used the deltaSVM[54] framework to predict the impact of regulatory variants on the binding of TFs. We noticed that, in one of the potential causal variants (rs6488724), the overlapping peak is located in the promoter of *MGP* (Fig. 6b), which is involved in myogenesis[55]. This single-nucleotide polymorphism (SNP) creates a G-to-A mutation that increases the binding affinity of HSF2, which participates in the transcriptional regulation of sarcomeric chaperones to maintain the contractile apparatus[56]. We also identified rs73746499, located in the intronic region of *FKBP5* loci, in myofibre[57] (Fig. 6c). This SNP creates an A-to-G mutation that disrupts the binding affinity of androgen receptor, one of the key TFs for maintaining muscle mass[58]. Notably, chromatin accessibility at the *FKBP5* locus substantially decreased with ageing, consistent with the decrease in *Fkbp5* expression in sarcopenic mice[57].

## Discussion

Our reference atlas for human skeletal muscle ageing provides a compelling series of integrated cellular and molecular explanations for increased sarcopenia and frailty development in older individuals

(Extended Data Fig. 13d). Further exploration using our open and interactive online portal, the Human Muscle Ageing Cell Atlas (HMA) (https://db.cngb.org/cdcp/hlma/), will generate additional insights.

Ageing leads to considerable alterations in the composition of myofibres and the characteristics of myonuclei. These changes include the loss and gain of specific myonucleus types, the emergence of new subtypes, and the alteration of gene programs and GRNs in a general or myofibre-type-specific manner. For example, we observed an overall activation of inflammatory and catabolic programs, impaired expression of contractile protein genes, altered myonuclear identity, upregulation of repair and innervation gene signatures in type I and II myonuclei, and the emergence of myonucleus subtypes associated with denervation. These phenomena may represent compensatory mechanisms, could be causal factors contributing to sarcopenia, or both. Notably, type I myonuclei undergo metabolic reprogramming towards a more glycolytic phenotype, probably counterbalancing the loss of oxidative capacity in resilient type I myofibres. By contrast, type II myonuclei exhibit increased glycogen depletion and protein catabolism processes, explaining their higher susceptibility to atrophy.

Quiescent MuSCs are substantially reduced in aged muscle, whereas resident non-myogenic cells are increased. Importantly, the remaining MuSCs undergo chronic activation of inflammatory and stress pathways, which could explain their failure to proliferate and differentiate[9]. These changes translate into MuSCs being more primed for activation, which may partly account for their exhaustion at an advanced age[9]. The alteration in the activity of TFs involved in the stress response and muscle maintenance probably contributes to the disruption of MuSC homeostasis. In stromal cells, ageing causes clear alterations in vascular integrity, with increased pro-inflammatory and chemoattractant signals, whereas immune cells increase in numbers and turn on inflammatory programs. Furthermore, during ageing, the heterogeneous population of FAPs switches from a proregenerative profile to a profibrotic one, accompanied by a higher presence of mature adipocytes. These changes may predispose the skeletal muscle to cellular senescence in the presence of overt damage, such as trauma[12]. In turn, this pro-inflammatory muscle state may also contribute to systemic inflammation (inflammageing)[40] and accelerate the overall body decline in older individuals. We conclude that the perturbed relationship of muscle cells with mononuclear cells in the niche and the imbalance between pro-fibrotic and pro-regenerative signals acts as a major cause of muscle dysfunction in old age. Comparison with GWAS datasets also enabled us to identify the potential relationship between genome architecture in different cell types and heritable susceptibility to sarcopenia.

Future expansions of HMA will include a larger cohort size and muscle samples from different origins, single-cell multiomics and high-definition spatially resolved technologies[59]. This may reveal differences in ethnic and sex groups unnoticed in the current study. Together, it may provide a window of opportunity for slowing down or even blocking sarcopenia, frailty and disability in older people, promoting healthier body ageing over a longer time and enhancing longevity. In addition to the ageing field, this atlas will be an important reference for future studies in patients with neuromuscular diseases.

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

# Methods

## Muscle biopsy and ethical clearance

Samples were taken during orthopaedic surgery with informed consent from the 18 patients in the European cohort and the 13 patients in the Asian cohort; for one individual below 18 years, the informed consent was obtained from the legally acceptable representative. The study was performed following the Declaration of Helsinki. Ethical approval was granted for the European cohort by the Research Ethics Committee of Hospital Arnau de Vilanova (CEIm 28/2019), and for the Asian cohort by the Institutional Ethics Committee of the First Affiliated Hospital/School of Clinical Medicine of Guangdong Pharmaceutical University, Guangzhou (China) (2020-ICE-90). Exclusion criteria were myopathy, haemiplegia or haemiparesis, rheumatoid arthritis or other autoimmune connective tissue disorders, inability to consent, prior hospital admission in the previous month or major surgery in the previous 3 months. For the European cohort, the individuals' medical and functional states were assessed according to the Barthel index[10] and Charlson Index[11]. The Barthel index estimates the grade of dependency of the individual ranging from 0 (totally dependent) to 100 (independent). The Charlson Index indicates the grade of comorbidities associated with the individual and ranged from 0 (without comorbidity) to 6 (the individual with a higher number of comorbidities) in our samples. A list of detailed information for the individuals is provided in Supplementary Table 1.

## Animal experiment

C57Bl/6 (wild type) mice were bred and raised until 8–12 weeks of age at the animal facility of the Barcelona Biomedical Research Park (PRBB). They were housed in standard cages under a 12 h–12 h light–dark cycle and given unrestricted access to a standard chow diet. All experiments adhered to the 'three Rs' principle—replacement, reduction and refinement—outlined in Directive 63/2010 and its implementation in Member States. Procedures were approved by the PRBB Animal Research Ethics Committee (PRBB-CEEA) and the local government (Generalitat de Catalunya), following European Directive 2010/63/EU and Spanish regulations RD 53/2013. Both male and female mice were used for experiments and were maintained according to the Jackson Laboratory guidelines and protocols. Mice were randomly allocated to experimental or treatment groups. No blinding was used. No statistical methods were used to predetermine the sample size. Muscle injury was induced by intramuscular injection of CTX (Latoxan, L8102, 10 µM) and mice were euthanized at 7 days after injury as previously described[12].

## Muscle sample processing

Muscle samples were obtained in all cases by selecting a macroscopically healthy area of muscle, without signs of contusion or haematoma. A small portion of muscle was removed by blunt dissection following the course of the myofibres and avoiding the use of electrocautery. The samples were immediately processed into three groups and stored next to the operating room as follows: (1) fixed with paraformaldehyde before being mounted in OCT compound as described previously (for immunochemistry and immunofluorescence)[60]; (2) immediately frozen in liquid nitrogen (for snRNA-seq and snATAC-seq); and (3) tissue-digested (for scRNA-seq).

## Single-cell preparation from skeletal muscle

Before the experiment, the post-operative muscle was immediately transferred in prechilled Dulbecco's modified Eagle's medium (DMEM, Corning, 10-017-CVR). For single-cell isolation, adipose and tendon tissues were removed using forceps, the remained muscle chunks were mechanically shredded on ice in a 10 cm plate. Next, prechilled DMEM medium was added to the plate for collecting muscle tissues and transferred into a 50 ml tube. After standing for 3 min, the supernatant containing the remaining adipose tissues was discarded. The remained muscle tissues were transferred to a 15 ml tube for digestion in 5 ml tissue digestion buffer (0.2 mg ml$^{-1}$ liberase (Roche, 5401119001), 0.4 µM CaCl$_2$ (Thermo Fisher Scientific, J63122AE), 5 µM MgCl$_2$ (Thermo Fisher Scientific, R0971), 0.2% BSA (Genview, FA016), 0.025% trypsin-EDTA (Thermo Fisher Scientific, 25300120). The muscles were digested in a shaking metal bath at 1000 rpm, 37 °C for 1 h, and mixed by inversion every 10 min. After all tissue pieces were digested, 3 ml of fetal bovine serum (FBS, Cellcook, CM1002L) was added to the mixture to terminate the digestion. The cell suspension was filtered through a 100 µm strainer, and centrifuged at 700$g$ for 10 min at 4 °C to pellet the cells. The cell pellet was then resuspended in 10 ml wash buffer (DMEM medium supplemented with 10% FBS) and filtered through a 40 µm strainer, then centrifuged at 700$g$ for 10 min at 4 °C to pellet the cells. The resultant single-cell suspensions were washed twice with prechilled PBS supplemented with 0.04% BSA and were used as input for scRNA-seq library construction.

## Single-nucleus extraction from skeletal muscle

Single-nucleus isolation was performed as previously described[6]. In brief, tissues were thawed, minced and transferred to a 2 ml Dounce homogenizer (Sigma-Aldrich, D8938) with 1 ml of homogenization buffer A containing 250 mM sucrose (Sigma-Aldrich, S8501), 10 mg ml$^{-1}$ BSA, 5 mM MgCl$_2$, 0.12 U µl$^{-1}$ RNasin (Promega, N2115) and 1× cOmplete Protease Inhibitor Cocktail (Roche, 11697498001). Frozen tissues were kept in an ice box and homogenized by 25–50 strokes of the loose pestle (pestle A), after which the mixture was filtered using a 100 µm cell strainer into a 1.5 ml tube. The mixture was then transferred to a clean 1 ml Dounce homogenizer with 750 µl of buffer A containing 1% Igepal (Sigma-Aldrich, CA630), and the tissue was further homogenized by 25 strokes of the tight pestle (pestle B). The mixture was then filtered through a 40 µm strainer into a 1.5 ml tube and centrifuged at 500$g$ for 5 min at 4 °C to pellet the nuclei. The pellet was resuspended in 1 ml of buffer B containing 320 mM sucrose, 10 mg ml$^{-1}$ BSA, 3 mM CaCl$_2$, 2 mM magnesium acetate, 0.1 mM EDTA (Thermo Fisher Scientific, 15575020), 10 mM Tris-HCl (Invitrogen, AM9856), 1 mM DTT (Invitrogen, 707265ML), 1× Complete Protease Inhibitor Cocktail and 0.12 U µl$^{-1}$ RNasin. This was followed by centrifugation at 500$g$ for 5 min at 4 °C to pellet the nuclei. The nuclei were then washed twice with prechilled PBS supplemented with 0.04% BSA and finally resuspended in PBS at a concentration of 1,000 nuclei per µl for library preparation.

## Library preparation and sequencing

**sc/snRNA-seq library preparation.** scRNA-seq libraries were prepared using the DNBelab C Series Single-Cell Library Prep Set (MGI, 1000021082)[49]. In brief, the single-cell/nucleus suspensions were converted to barcoded scRNA-seq libraries through droplet encapsulation, emulsion breakage, mRNA-captured bead collection, reverse transcription, cDNA amplification and purification. Indexed sequencing libraries were constructed according to the manufacturer's instructions. Library concentrations were quantified using the Qubit ssDNA Assay Kit (Thermo Fisher Scientific, Q10212). Libraries were sequenced using the DIPSEQ T1 sequencer.

**snATAC-seq library preparation.** snATAC-seq libraries were prepared using the DNBelab C Series Single-Cell ATAC Library Prep Set (MGI, 1000021878)[49]. In brief, nuclei were extracted from tissue using the same protocol described above. After Tn5 tagmentation, transposed single-nucleus suspensions were converted to barcoded snATAC-seq libraries through droplet encapsulation, pre-amplification, emulsion breakage, captured bead collection, DNA amplification and purification. Indexed libraries were prepared according to the manufacturer's instructions. Concentrations were measured with a Qubit ssDNA Assay Kit. Libraries were sequenced by a BGISEQ-2000 sequencer.

## sc/snRNA-seq raw data processing, clustering and cell type annotation

**Raw data processing.** Raw sequencing reads were filtered, demultiplexed, and aligned to hg38 reference genome using a custom workflow (https://github.com/MGI-tech-bioinformatics/DNBelab_C_Series_HT_scRNA-analysis-software)[49]. For scRNA-seq, reads aligned to gene exons were counted. For snRNA-seq, reads aligned to gene loci, including both exons and introns, were counted. Doublets were identified and filtered by DoubletFinder (v.2.0.3)[61]. Ambient RNA for snRNA-seq was reduced using SoupX (v.1.4.8)[62] with the default settings.

**Integration, clustering and cell type annotation.** The resulting count matrix for cells/nuclei was filtered by the number of unique molecular identifiers (UMIs) > 1,000, gene > 500 and mitochondria content < 5%. Global clustering was performed using Scanpy (v.1.8.1)[63] in Python (v.3.7). Filtered data were normalized to total counts and log-transformed. The top 3,000 highly variable genes were selected, and the number of UMIs and the percentage of mitochondrial genes were regressed out. Each gene was scaled with the default options, followed by dimensionality reduction using principal component analysis. Batch effects between snRNA-seq and scRNA-seq were corrected using Harmony[64]. Next, the batch-effect-corrected top 30 principal components were used for generating the neighbourhood graph with the number of neighbours set at 10. The cell clustering was further performed with the Louvain algorithm and annotated by canonical markers, putative scRNA-seq- and snRNA-seq-derived myofibre fragments were removed from the analysis. For satellite cell, immune cell, vascular cell and stromal cell reclustering, cells/nuclei were subset from the global clustering object and processed according to the same procedure as described above. For the reclustering of myonuclei, data were processed in Seurat (v.4.0.2)[65], and only snRNA-seq data were retained for further analysis. In brief, myonuclei data were subjected to SCTransform-based normalization, anchor identification between samples, integration, Louvain clustering and projection onto the UMAP space. Clustering results were further annotated by highly expressed genes.

**Analysis of cell type composition variation in ageing.** A generalized linear mixed model with a Poisson outcome[14] was used to model the effect of age on cell-type-specific counts as previously reported, accounting for the possible biological (sex, ethnicity) and technical (omics, sequencing batch) covariates. The effect of each biological/technical factor on cell type composition was estimated by the interaction term with the cell type. The fold change is relative to the grand mean and adjusted. The statistical significance of the fold change estimation was measured by the LTSR, which is the probability that the estimated direction of the effect is true. As an alternative method, the proportion for each population was estimated over the total number of nuclei/cells for a given dataset (Supplementary Table 9).

**Transcriptional and epigenetic heterogeneity analysis.** Transcriptional heterogeneity analysis was performed as previously described[66]. In brief, snRNA-seq data for each cell type in each age group were downsampled to 300 nuclei. For cell types with fewer than 300 nuclei, all nuclei were included for analysis. The resultant gene × cell matrix was further downsampled to make an equal number of UMI counts and cells between adult/older adult groups in each cell type. Next, all genes were ranked into ten blocks on the basis of the average expression value, and the 10% genes with the lowest coefficient of variation in each block were used to calculate the Euclidean distance between each cell. This Euclidean distance was used to measure transcriptional heterogeneity for each cell. For epigenetic heterogeneity, we adapted the same analysis method as transcriptional noise but using the rounded gene score matrix as input.

**Myonucleus classification.** Myonuclei were classified on the basis of previous markers associated with the described pure myofibre types (type I, type IIA and type IIX) and the hybrid myofibres (hybrid I/IIA, hybrid IIA/IIX). A module score was calculated for each myofibre type based on the expression of the following markers[67]: type I (*TNNT1, MYH7, MYH7B, TNNC1, TNNI1,* and *ATP2A2*), type II (*TNNT3, MYH1, MYH2, TNNC2, TNNI2, ATP2A1*), type IIA (*MYH2, ANKRD2, NDUFA8, MYOM3, CASQ2, HSPB6, RDH11, AIMP1*) and type IIX (*MYH1, MYLK2, ACTN3, MYBPC2, PCYOX1, CAPZA1, CD38, PDLIM7, COBL, TMEM159, HNRNPA1, TFRC*). On the basis of these scores, myonuclei were first classified as type I, type II or hybrid I/IIa; thereafter, type II myonuclei were further classified as type IIA, type IIX or hybrid IIA/IIX. A residual amount of myonuclei remained unclassified due to the lower expression of these genes.

**Differential gene expression and functional enrichment analysis.** Seurat was used to compute the DEGs for each population and subpopulations between samples in the younger and older cohorts with the thresholds set at $\log_2$[fold change] > 0.25 and $Q$ < 0.05 (Supplementary Table 10). For myofibre subpopulations, the thresholds were set at: $\log_2$[fold change] > 1 and $Q$ < 0.05. The obtained DEGs for each comparison were used as input in Metascape online tool[68] to perform functional enrichment analysis, with a $Q$ value threshold set at 0.05 (Supplementary Table 11). Heatmap results were plotted using pheatmap (v.1.0.12) in R.

**Identification of coexpressing gene modules.** Hotspot (v.1.1.1)[16] was used to compute coexpressing gene modules among myofibre populations. The normalized expression matrix for the top 5,000 variable genes, the RegMyon-[19], sarcomeric-[67] and atrophy-related[24] genes (Supplementary Table 3) were used as input. In brief, the $k$-nearest neighbour graph was created using the create_knn_graph function with the parameters: n_neighbors = 30, and then genes with significant correlation ($Q$ < 0.05) were retained for further analysis. The modules were identified using the create_modules function with the parameters min_gene_threshold = 10 and fdr_threshold = 0.05.

**Pseudotime analysis.** For the myofibre degeneration trajectory, *DCLK1*+ (type I), *ID1*+ (type I), *ID1*+ (type II), *ENOX1*+ (type II) and other unperturbed myonuclei were selected for pseudotime analysis using Monocle3[69]. After trajectory construction, myonuclei were ordered by pseudotime, and the corresponding gene expression matrixes were aggregated into 100 bins. The top 4,000 variable genes in type I or type II myonucleus trajectory were selected and visualized by $k$-means clustering heat map ordered by the pseudotime.

**Cell–cell interaction analysis.** CellChat (v.1.1.0)[45] detected ligand–receptor interactions on integrated sc/snRNA-seq data according to the standard procedures. The expression matrix and the cell type information were imported to CellChat. Specialized myonuclei, mast cells and erythrocyte clusters were removed from the analyses due to the insufficient number of cells/nuclei or the disproportionate number of cells/nuclei between the younger and older cohorts. The overall communication probability among the cell clusters was calculated using the computeCommunProb function with a trim set at 0.1.

## snATAC-seq data processing

**Raw data processing, clustering and cell type annotation.** Raw sequencing reads were filtered, demultiplexed and aligned to the hg38 reference genome using PISA (https://github.com/shiquan/PISA)[70]. Fragment files for each library were generated for downstream analysis. The transcription start site enrichment score, number of fragments and doublet score for each nucleus were calculated using ArchR[71]. Nuclei with transcription start site enrichment scores < 8 and number of fragments < 1,000 were removed from the analysis. Doublets were

filtered out using the filterDoublets function with the settings filterRatio = 2. We next performed latent-semantic-indexing-based dimensionality reduction on the 500 bp tiles across the genome using the addIterativeLSI function of ArchR. Anchors between the scATAC-seq and scRNA-seq/snRNA-seq datasets were identified and used to transfer cell type labels identified from the scRNA-seq/snRNA-seq data. For co-embedding of snRNA-seq/scRNA-seq and snATAC-seq data, an anchor-based integration approach was applied based on the sequencing techniques. Then, data were further subjected to batch correction by Harmony among samples. Pearson's correlation between snRNA/scRNA-seq and snATAC-seq was performed based on the integrated assay.

**Motif enrichment analysis.** Before motif enrichment, a reproducible peak set was created in ArchR[71] using the addReproduciblePeakSet function based on cell types/subtypes. Differentially enriched peaks were identified using the getMarkerFeatures function with the thresholds set at $\log_2$[fold change] > 0.5 and $Q < 0.1$. The motif presence in the peak set was determined with the addMotifAnnotations function using CisBP motif database (v.2)[72].

**TF occupancy.** TF occupancy was evaluated by footprinting analysis implemented in ArchR[71]. In brief, putative binding sites of selectively enriched motifs were first inferred using the addMotifAnnotations function. Next, footprintings for the putative TF-binding sites were calculated using the getFootprints function, in which the Tn5 insertion bias was taken into account. The results were further plotted using the plotFootprints function.

### GRN analysis

Construction of the GRNs was performed using FigR[31]. In brief, we first sampled an equal number of nuclei (20,000) in snRNA-seq and snATAC-seq analysis of myofibre and performed data integration using scOptMatch implemented in FigR. For creating the co-embedding map in these two independent datasets, we first input the variable features taken from the snRNA-seq and snATAC-seq datasets to perform CCA using the RunCCA function in Seurat. After integration, pairs of ATAC–RNA cells were identified by geodesic distance-based pairing using the pairCells function, and unpaired cells were removed from the analysis. Significant ($P < 0.05$) peak-to-gene associations were then identified among the cell pairs in type I or type II myonuclei. The DORCs were defined as peak-gene associations ≥ 6. For inference of the GRNs, the smoothed DORC score, RNA counts, snATAC-seq peak counts and the significant peak-to-gene associations were fed into runFigRGRN function, generating the GRNs. Next, the activators and repressors were identified by ranking the TFs by average regulation score.

### GWAS analysis

**Association of GWAS traits with skeletal muscle cell types.** To identify trait/disease-relevant cell types, we performed LDSR analysis[73], a method for partitioning heritability from GWAS summary statistics. In brief, differentially accessible peaks for each adult/older adult cell type were identified ($\log_2$[fold change] > 1 and $Q < 0.01$). The LDSC analysis was performed according to the standard workflow (https://github.com/bulik/ldsc/wiki). The summary statistics file for each trait was downloaded from the GWAS catalogue database[74] or published studies[50,51] (Supplementary Table 8a).

**Fine mapping of non-coding variants and predicting the effect of TF binding.** Lead SNPs were taken from low-hand-grip strength and lean-body-mass traits[50,51,75]. FUMA, a web-based platform for GWAS analysis[76], was used to identify high-correlation SNPs with an LD $r^2 \geq 0.8$ with lead SNPs. High-correlation SNPs within ±50 bp of the differentially accessible peaks were identified for further analysis. The peak-to-gene associations were determined using addPeak2GeneLinks function in the

ArchR package in the integrative object. To identify SNPs that affect TF binding, we used two approaches, (1) gkm-SVM[54] and (2) SNP2TFBS[77]. For gkm-SVM, TF models were used from https://github.com/ren-lab/deltaSVM/tree/master/gkmsvm_models, and effective alleles were identified using the gkmExplain function[78]. For SNP2TFBS tools, the analysis was performed in the SNP2TFBS web interface (https://ccg.epfl.ch/snp2tfbs/) following the tutorial.

### Histology and immunofluorescence

Cryostat sections (10 μm thickness) were collected from muscles and stained with haematoxylin and eosin (Sigma-Aldrich, HHS80 and 45235) to assess tissue morphology or SA-β-gal (AppliChem, A1007,0001) for senescence cells with a modified staining protocol as described previously[12,79]. Histochemical SDH staining was assayed by placing the slides in a solution containing sodium succinate (Sigma-Aldrich, S2378) as a substrate and nitro-blue tetrazolium (Sigma-Aldrich, N6876) for visualization of the reaction for 1 h at 37 °C. The intensity and pattern of staining were evaluated using light microscopy[80]. Muscle collagen content was quantified after Sirius Red (Sigma-Aldrich, 365548) staining as previously described[81]. For immunofluorescence, the sections were air-dried, fixed, washed on PBS, permeabilized with Triton X-100 0.5% (Sigma-Aldrich, 11332481001) and incubated with primary antibodies (diluted as indicated below) after blocking with a high-protein-containing solution (BSA at 5%) (Sigma-Aldrich, A7906-100G) in PBS overnight at 4 °C. Subsequently, the slides were washed with PBS and incubated for 1 h at room temperature with the appropriate secondary antibodies diluted at 1:500; DAPI (Thermo Fisher Scientific, 62248) at 1:1,000 for nuclei; and WGA (Thermo Fisher Scientific, W11261) at 1:200 for cell/myofibre membrane. After washing, the tissue sections were mounted with Mowiol (Sigma-Aldrich, 81381) or Fluoromount-G (SoutherBiotech, 0100-01). Quantitative results for histology and immunofluorescence are listed in Supplementary Table 12. Primary antibodies were as follows: PAX7 (DSHB, PAX7, 1:50), PDGFRa (eBioscience, 17-1401-81, 1:100), perilipin-1 (Cell Signalling, 9349, 1:100), filamin C (MyBiosource, MBS2026155, 1:100), TNNT2 (Bioss, BS-10648R, 1:100), CD11b (eBioscience, 14-0112-85, 1:100), CD3 (Invitrogen, 14-0038-82, 1:100), CD19 (eBioscience, 14-0199-82, 1:100), NCAM1 (Cell Sciences, MON9006-1, 1:100), MYH7 (MyHC type I) (DSHB, A4.840, 1:10), MyHC type IIA/IIX (DSHB, SC-71, 1:70), laminin-647 (Novus Biologicals, NB300-144AF647, 1:200), FOS (Cell Signalling, 2250S, 1:200), ACVR2A (R&D, AF340, 1:100), ITGA7 (BioCell Scientific, 10007, 1:100), dystrophin (Sigma-Aldrich, D8168, 1:100). Secondary antibodies were as follows: goat anti-mouse IgM (DyLight 550, Invitrogen, SA5-10151), goat anti-mouse IgG1 (Alexa Fluor 488, Invitrogen, A21121), goat anti-mouse IgG (Alexa Fluor 488, Invitrogen, A11001), goat anti-mouse IgG (Alexa Fluor 568, Invitrogen, A11004), goat anti-rabbit IgG (Alexa Fluor Plus 488, Invitrogen, A32731TR), goat anti-rabbit IgG (Alexa Fluor Plus 647, Invitrogen, A32733TR), donkey anti-goat IgG (Alexa Fluor Plus 647, Invitrogen, A32849TR), goat anti-rat IgG (Alexa Fluor 568, Invitrogen, A11077).

### Digital image acquisition and processing

Immunohistochemistry images were acquired using an upright microscope (Leica DMR6000B) equipped with a DFC300FX camera, and, for immunofluorescence pictures, using a Hamamatsu ORCA-ER camera. Images were acquired using HCX PL Fluotar objectives (×10/0.30 NA, ×20/0.50 NA and ×40/0.75 NA) and LAS AF software (Leica, v.4.0). Immunofluorescence pictures were also obtained using the Nikon Ti2 fluorescence microscope with NIS Elements software (v.4.11.0), and a confocal microscope (Zeiss 980 Airyscan2) with ZenBlue software (v.3.5) and a ×20 air objective. The acquired images were composed, edited and analysed using Fiji (ImageJ, v.2.14.0/1,54f). To reduce background, brightness and contrast adjustments were applied to the entire image. Myofibre size was assessed using the MyoSight tool[34], with a manual correction applied after automated outlining, and the

cross-sectional area (CSA) was determined using Fiji. Signals of SA-β-gal, PAX7, PDGFRα, perilipin, CD11B, CD3, CD19, TNNT2, NCAM1, filamin C, SDH and FOS staining were manually counted in Fiji. The area of ACVR2A, Sirius Red and ITGA7 staining was calculated by normalizing the positive-signal area to the total imaged area in Fiji.

## Statistical analysis

The sample size of each experimental group or number of independent experiments is described in the corresponding figure legend. The calculation method for *P* values is explained in the figure legends. The number of replicates for each experiment is presented in the figure legends. For Pearson's correlation, statistical significance for positive or negative correlation (represented as the *R* value) was set at $P < 0.05$ and shading represents the 95% confidence interval along the correlation line (Supplementary Table 5). For the box plots, the central line shows the median, the box limits indicate the upper and lower quartiles, and the whiskers indicate 1.5× the interquartile range. Python, R or Prism (v.10) were used for statistical analyses.

## Reporting summary

Further information on research design is available in the Nature Portfolio Reporting Summary linked to this article.

## Data availability

All raw data have been deposited to CNGB Nucleotide Sequence Archive (CNP0004394, CNP0004395, CNP0004494 and CNP0004495). All processed data are available at the Human Muscle Ageing Cell Atlas database (https://db.cngb.org/cdcp/hlma/). The data deposited and made public is compliant with the regulations of the Ministry of Science and Technology of the People's Republic of China. Source data are provided with this paper.

## Code availability

All data were analysed using standard programs and packages, as described in the Methods. Custom code supporting the current study was created for the processing of the sequencing data[49] (https://github.com/MGI-tech-bioinformatics/DNBelab_C_Series_HT_scRNA-analysis-software) and analysis of the generated data[82] (https://github.com/123anjuan/HMA).

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

**Acknowledgements** We thank all of the members of our teams for technical help and discussions; M. Raya for project coordination; A. Navarro for technical assistance; C. García-Domínguez for help in patient characterization; and V. A. Raker for editing. The members of Y.L. and M.A.E.'s team thank the staff at the China National GeneBank, H. Li and D. Zhu (Bioland Laboratory, Guangzhou, China) and M. Cao (Chinese University of Hong Kong, Hongkong, China) for their support. Work in the P.M.-C. laboratory was supported partly by Milky Way Research Foundation and UPGRADE-H2020-825825 at MELIS (recipient of a Maria de Maeztu Program for Units of Excellence to UPF (MDM-2014-0370)) and Altos Labs. Work in Y.L. and M.A.E.'s team was supported by the National Key Research and Development Program of China (2022YFC3400400), the National Natural Science Foundation of China (32200688, U20A2015, 32211530050), Guangdong Basic and Applied Basic Research Foundation (2021B1515120075, 2021A1515110180), Key Laboratory of Guangdong Higher Education Institutes (2021KSYS009), and Guangzhou Key Laboratory of Biological Targeting Diagnosis and Therapy (202201020379). Freshage Research Group was supported by the Instituto de Salud Carlos III CB16/10/00435 (CIBERFES); PID2022-142470OB-I00 (M.C.G-C) from the Spanish Ministry of Innovation and Science; PROMETEO: CIPROM/2022/56 (M.C.G.-C) from Conselleria de Educación, Universidades, y Empleo de la Generalitat Valenciana; EU Funded H2020-DIABFRAIL-LATAM: 825546 (M.C.G.-C), Red EXERNET-RED DE EJERCICIO FISICO Y SALUD: RED2022-134800-T (M.C.G-C) from the Spanish Ministry of Innovation and Science; Ramón Areces Foundation and Soria Melguizo Foundation (J.V.). I.R.-P. was supported by an FPI predoctoral fellowship (Ministry of Science, Spain). Y.S. was supported by the Guangdong Basic and Applied Basic Research Foundation (2020A1414040036). The schematic of Extended Data Fig. 13d was drawn using BioRender under the academic license terms (HI266XMRUN).

**Author contributions** P.M.-C. and M.A.E. supervised the study. Y.L., I.R.-P., J.I., E.P., A.L.S., P.M.-C. and M.A.E. conceptualized the study and wrote the manuscript. Y.L., I.R.-P., J.A., A.L.S., J.S., P.G., V.L. and E.A. performed the experiments. Y.L., I.R.-P., J.A. and J.L. analysed the data. E.G.-D., J.V., J.D.-F, M.C.G.-C. and Y.S. collected the muscle biopsies. J.Z., Y.Y. and C.L. provided technical support. L.L. and X.X. gave relevant advice. I.R.-P., J.I. and Y.L. drew the schematic. J.L. constructed the HMA website. J.V., J.D., M.C.G.-C. and Y.S. contributed equally.

**Competing interests** Y.L., P.G., J.Z., Y.Y., C.L., L.L., X.X. and M.A.E. are employees of BGI Group. I.R.-P., J.I., E.P., A.L.S. and P.M.-C. are employees of Altos Labs. The other authors declare no competing interests.

**Additional information**
**Correspondence and requests for materials** should be addressed to Pura Muñoz-Cánoves or Miguel A. Esteban.

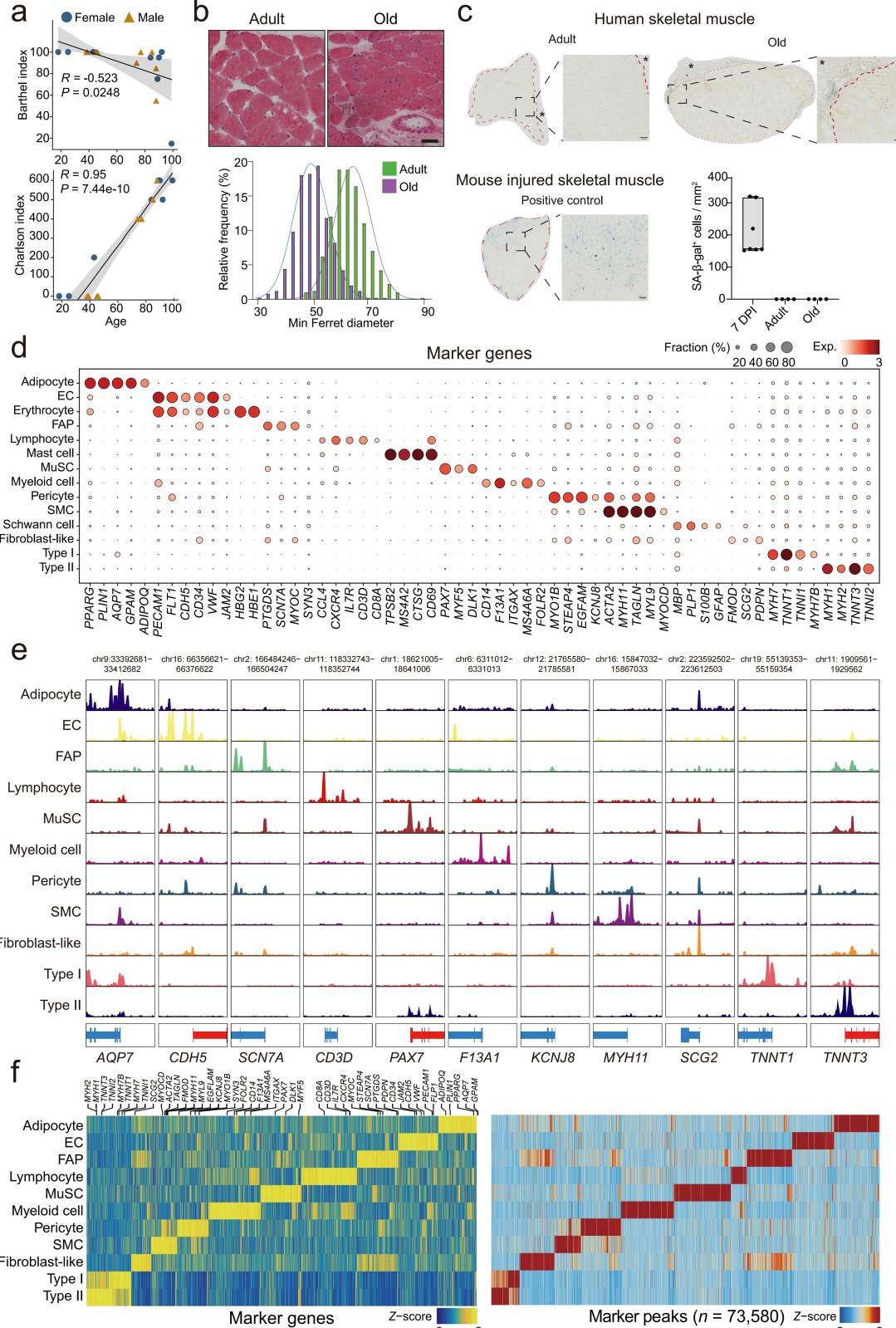

**Extended Data Fig. 1** | See next page for caption.

**Extended Data Fig. 1 | Functional characterization and major cell type identification in skeletal muscle across ageing. a**, Correlation plots of the Barthel index[10] (autonomy vs dependency assessment, top panel) and the Charlson index[11] (patient´s comorbidities, bottom panel) with the age of the individuals in both men (triangles) and women (circles). The shape and colour represent the sex group. *R*, Pearson correlation coefficient; *P*, two-sided *P*-value; centre represents the average value; shade indicates 95% confidence interval. **b**, Samples used in this study were subjected to H&E staining to assess their tissular integrity. One representative image from the tissue of each age group is depicted (top panel; adult, sample P13; old, sample P28). Scale bar, 50 μm. The myofibre size (Min Feret diameter) distribution was quantified between muscles from the two age groups (bottom panel). The colour represents the age group. *n* = 4 for each age group. **c**. Representative images of SA-β-gal staining of adult (sample P26), old (sample P28) uninjured human muscle (top panel) or of injured mouse muscle at 7 days post-cardiotoxin injury (7 DPI) as a positive control (bottom left panel). In each muscle section (overview and insets), muscle tissue is designated with the red dashed line, and the surrounding tissue is marked with an asterisk. Scale bars, 25 μm. Boxplot showing the SA-β-gal+ cells quantification within the myofibre tissue where boxes represent the upper/lower quartiles, the line depicts the median and whiskers the 1.5 interquartile range. *n* = 4 individuals from each age group for human muscle, *n* = 7 for injured mouse muscle. **d**, Dot plot showing the expression level for representative marker genes in the indicated cell types. Colour scale represents the average gene expression, and dot size, the percentage of cells expressing a given marker within the subpopulation. **e**, Normalized genome browser track profiles in the proximal promoters of the indicated marker genes (columns) for the indicated cell types (rows) measured in snATAC-seq. **f**, Heatmap of marker genes (gene score, left panel) and marker peaks (differentially accessible peaks, right panel) for the indicated cell types measured in snATAC-seq. Colour scale represents the relative score (*Z*-score) of each marker gene/peak.

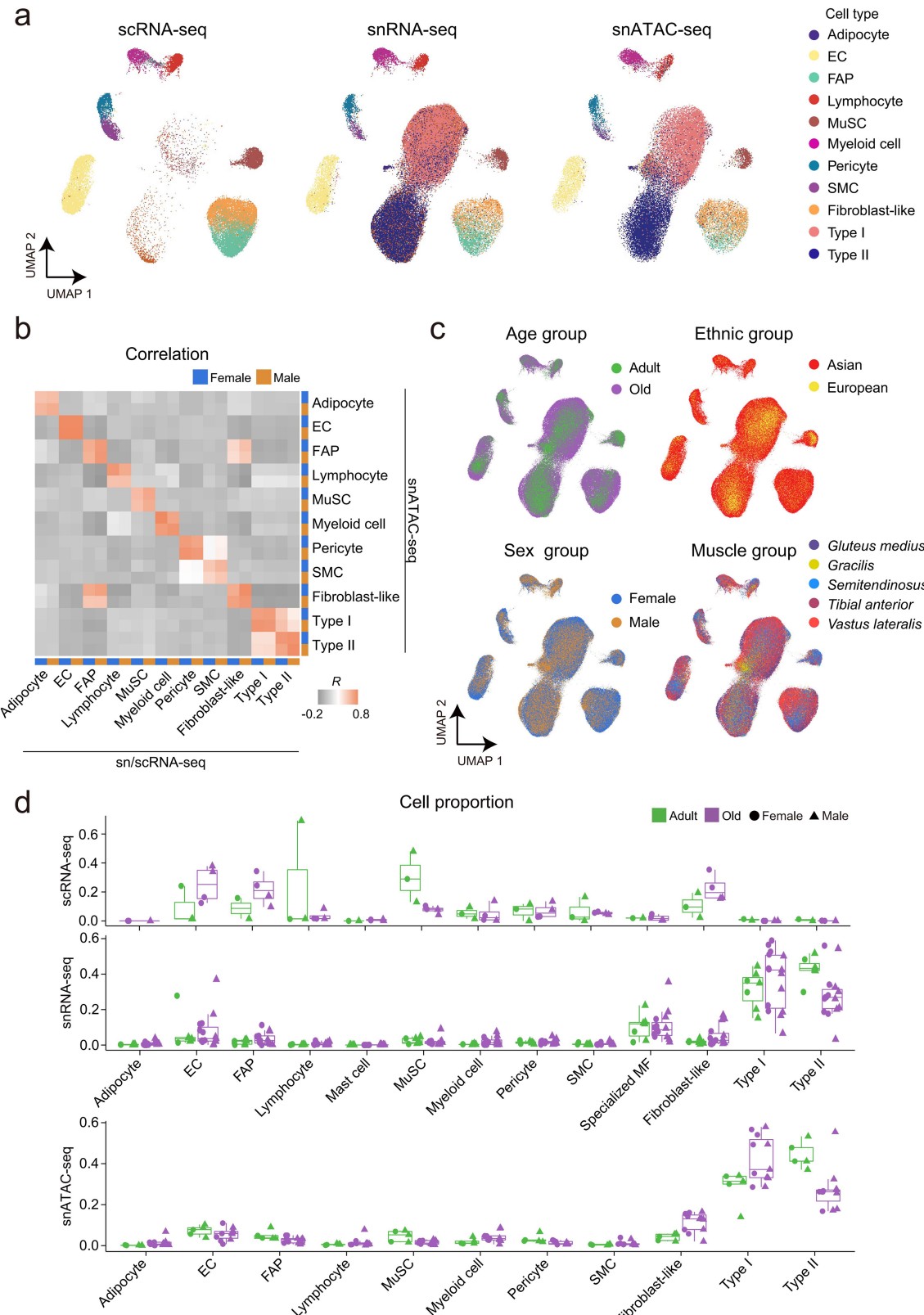

**Extended Data Fig. 2 | Integration of multimodal single-cell data. a**, UMAP plots of the cells/nuclei from the integrated scRNA-seq, snRNA-seq and snATAC-seq datasets split by technologies. Dots are coloured by the cell types. **b**, Correlation heatmap for cell-type assignment between the sc/snRNA-seq and snATAC-seq datasets. Colour scale represents the computed Pearson correlation coefficient (*R*). **c**, UMAP plots of the cells/nuclei from the integrated scRNA-seq, snRNA-seq and snATAC-seq datasets grouped by sampling factors (age, ethnic, sex and muscle group). Dots are coloured by the sampling factors within each group comparison. **d**, Cell proportion analysis for the percentage of each cell population in the scRNA-seq (top), snRNA-seq (middle) and snATAC-seq (bottom) for each individual. The colour represents the age group and the shape represent the sex group. For scRNA-seq, *n* = 3 adult individuals and *n* = 4 old individuals; for snRNA-seq, *n* = 7 adult individuals and *n* = 15 old individuals; for snATAC-seq, *n* = 5 adult individuals and *n* = 11 old individuals. For boxplots, boxes represent the upper/lower quartiles, the line depicts the median and whiskers the 1.5 interquartile range.

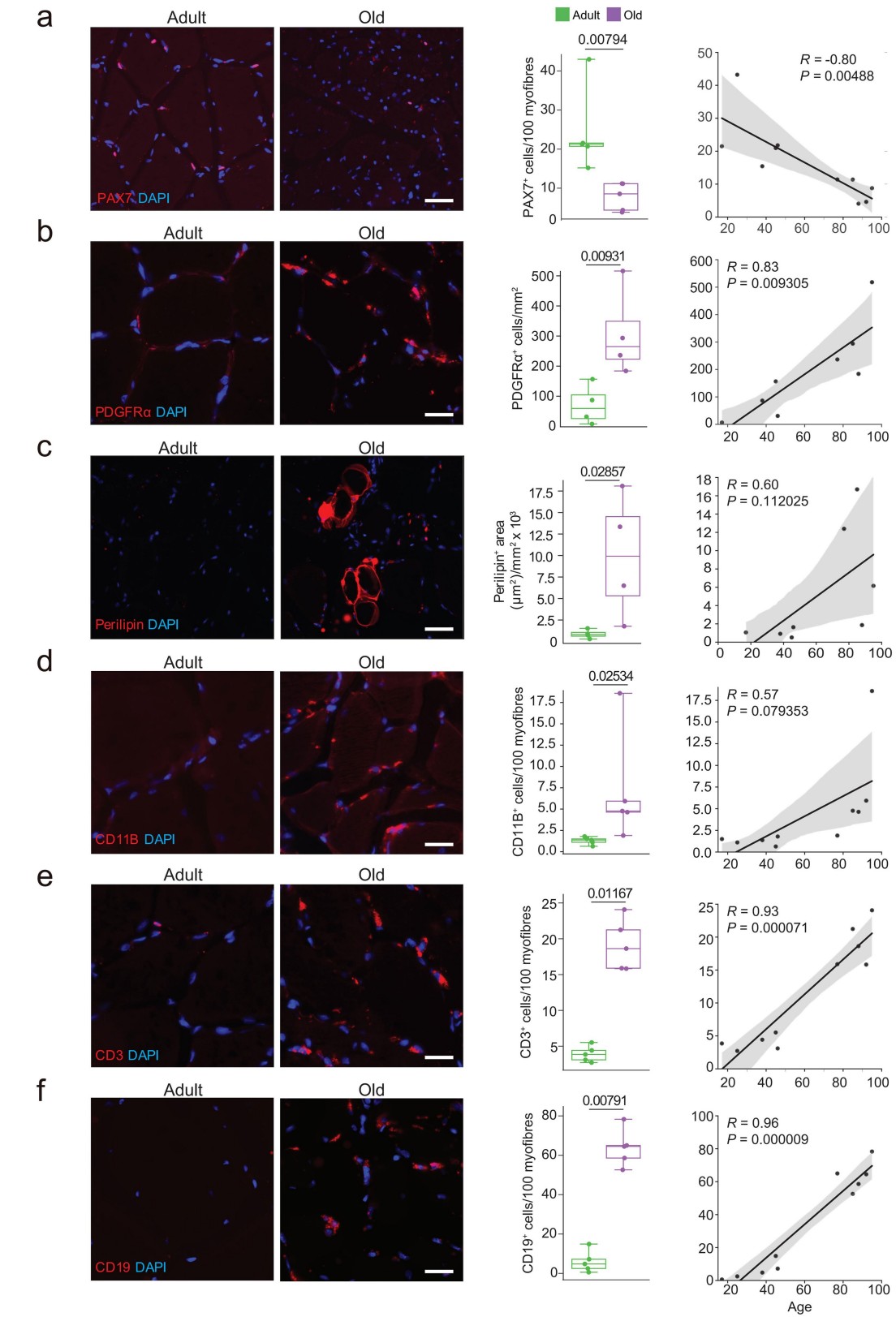

**Extended Data Fig. 3** | See next page for caption.

**Extended Data Fig. 3 | Cell proportion changes of human skeletal muscle with ageing.** Representative images of immunofluorescence staining from adult (left) and old (right) muscle tissue sections for specific cell types (red), their respective quantifications by age groups, and correlation plots with the age of the individuals (right panels). **a**, MuSCs labelled with anti-PAX7 (adult, sample P5; old, sample P28); **b**, FAPs and fibroblast-like cells labelled with anti-PDGFRα (adult, sample P5; old, sample P28); **c**, Adipocytes labelled with anti-Perilipin (adult, sample P26; old, sample P28); **d**, macrophages labelled with anti-CD11B (adult, sample P9; old, sample P29) **e**, T cells labelled with anti-CD3 (adult, sample P9; old, sample P16); **f**, B cells labelled with anti-CD19 (adult, sample P5; old, sample P23). Scale bars, 50 μm in (**a, c**), and 25 μm in (**b, d-f**). Nuclei were counterstained with DAPI (blue). For PDGFRα and perilipin, $n = 4$ individuals for each group; for PAX7, CD11B, CD3 and CD19, $n = 5$ individuals for each group. For all boxplots, $P$ values were calculated by a two-tailed Mann-Whitney $U$-test. For boxplots, boxes represent the upper/lower quartiles, the line depicts the median and whiskers the 1.5 interquartile range. For correlation plots, $R$, Pearson correlation coefficient; $P$, two-sided $P$-value; centre represents the average value; shade indicates 95% confidence interval.

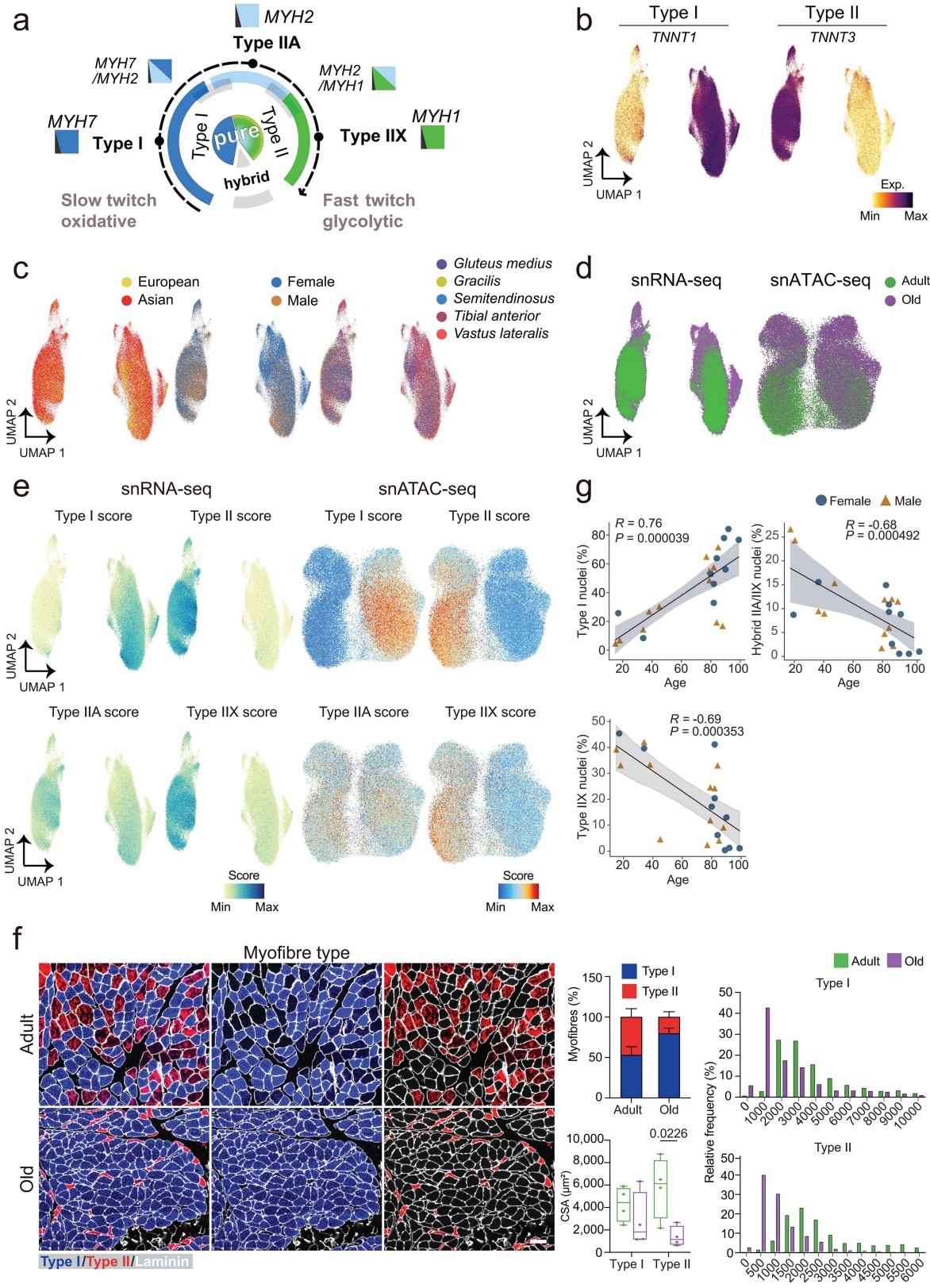

**Extended Data Fig. 4** | See next page for caption.

**Extended Data Fig. 4 | Myofibre type classification and proportion assessment with ageing. a**, Schematic depicting the definition of myofibre types according to myosin (MYH) genes expression. **b**, UMAP plots of myonucleus populations coloured according to *TNNT1* (type I myonuclei) and *TNNT3* (type II myonuclei) gene expression (colour scale). **c**, UMAP plots of myonucleus populations coloured and grouped by sampling factors (ethnic, sex and muscle group) in the snRNA-seq dataset. **d**, UMAP plots of myonucleus populations belonging to adult (green) and old (purple) muscles in both snRNA-seq (left panel) and snATAC-seq (right panel) datasets. **e**, UMAP plots of myonucleus type score for each myofibre type from the snRNA-seq dataset (left panels) and the snATAC-seq dataset (right panels). Myonucleus type scores were calculated based on the gene expression (snRNA-seq) or gene score (snATAC-seq) of distinct *MYH* genes among other myofibre-type specific genes (Supplementary Table 3). Dots (myonuclei) are coloured according to their computed score.

**f**, Representative images of immunofluorescence for myofibre typing from adult (left top, sample P13) and old (left bottom, sample P29) muscle tissue sections of type I (blue) and type II (red) myofibres delimited by laminin (white). Scale bars, 100 µm. Quantification graphs of average myofibre size (middle bottom), myofibre type proportion (middle top) and myofibre size distribution for type I (right top) and type II myofibres (right bottom). Myofibre size was measured as cross-sectional area (CSA). $n$ = 4 individuals of each age group. For boxplots, boxes represent the upper/lower quartiles, the line depicts the median and whiskers the 1.5 interquartile range. $P$ value was calculated by a two-tailed unpaired Student's t-test. For barplots, data are presented as mean values +/− SEM. **g**, Correlation plots of the proportion of indicated myofibre types with ageing. The shape and colours represent the sex group. $R$, Pearson correlation coefficient; $P$, two-sided $P$ value; centre represents the average value; shade indicates 95th confidence interval.

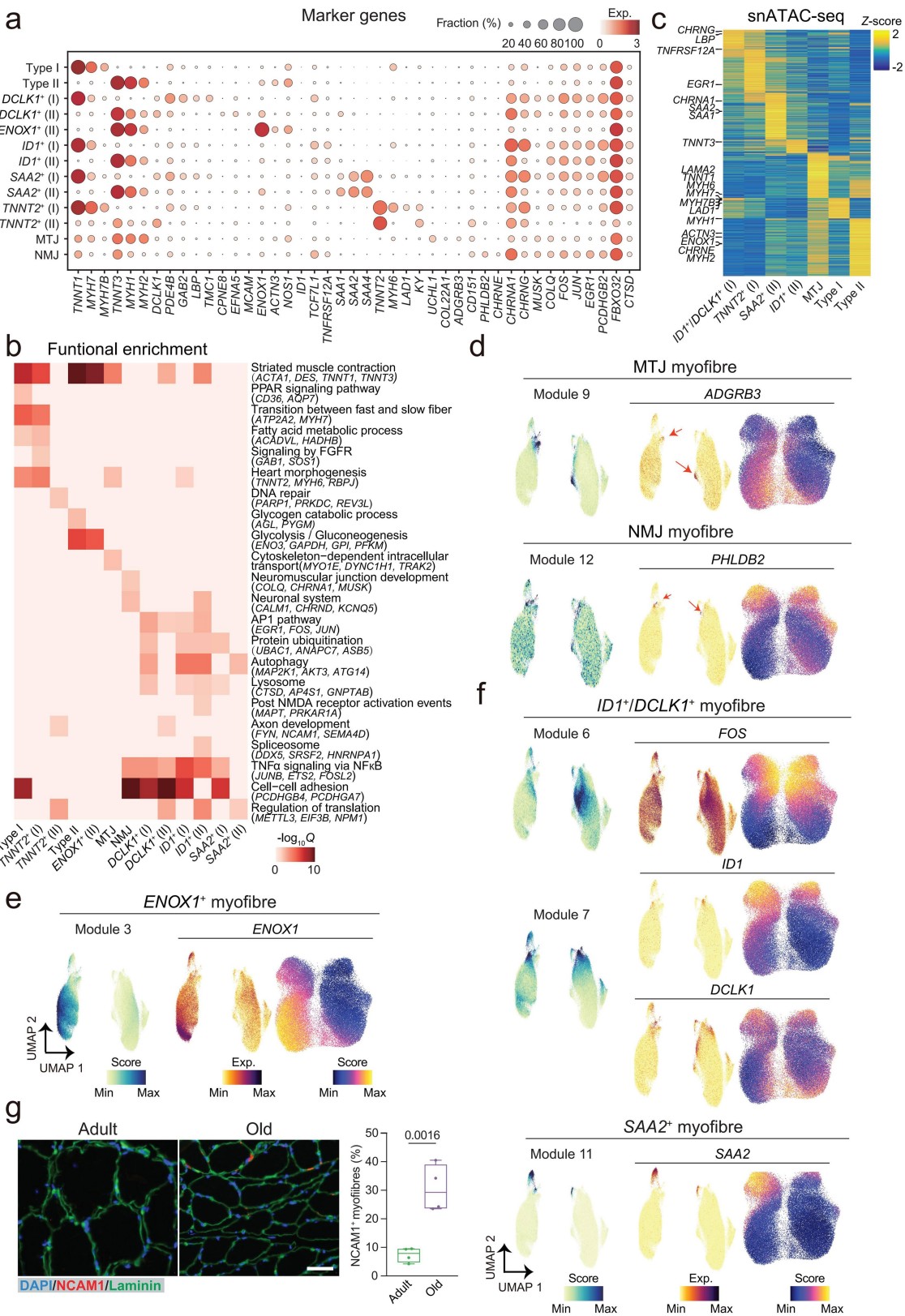

**Extended Data Fig. 5** | See next page for caption.

**Extended Data Fig. 5 | Markers for defining specialized myonucleus subtypes. a**, Dot plot showing the expression levels for representative marker genes in all myonucleus subpopulations. Colour scale represents the average gene expression, and dot size, the percentage of cells expressing a given marker within the subpopulation. **b**, Heatmap depicting enriched pathways from DEGs obtained in each myonucleus population. Colour scale represents the significance (-log$_{10}$ $Q$) of the enriched terms. **c**, Heatmap showing the gene score for DEGs (row) in each myonucleus population (columns) identified in snATAC-seq. Colour scale represents the scaled gene score ($Z$-score) for each DEG. **d-f**, UMAP plots highlighting the aggregated gene module expression (left panel); gene expression (middle panel) and gene score (right panel) for the following myonucleus subpopulations: **d**, module 9 with *ADGRB3* gene for MTJ population; and module 12 with *PHLDB2* gene for NMJ population; **e**, module 3 with *ENOX1* gene for *ENOX1*$^+$ population; **f**, module 6 and 7 with *FOS, ID1* and *DCLK1* genes for *ID1*$^+$/*DCLK1*$^+$ population; and module 11 with *SAA2* genes for *SAA2*$^+$ population. Dots (myonuclei) are coloured according to their computed score or gene expression value. **g**, Representative images (left panel) and the corresponding quantification (right panel) of immunofluorescence for denervated myofibres (NCAM1, red; myofibre membrane, laminin, green; nuclei, DAPI, blue) in adult (sample P9) and old (sample P28) individuals. $n = 4$ individuals of each age group. Scale bar, 10 µm. For boxplots, boxes represent the upper/lower quartiles, the line depicts the median and whiskers the 1.5 interquartile range. $P$ value was calculated by a two-tailed Student's t-test.

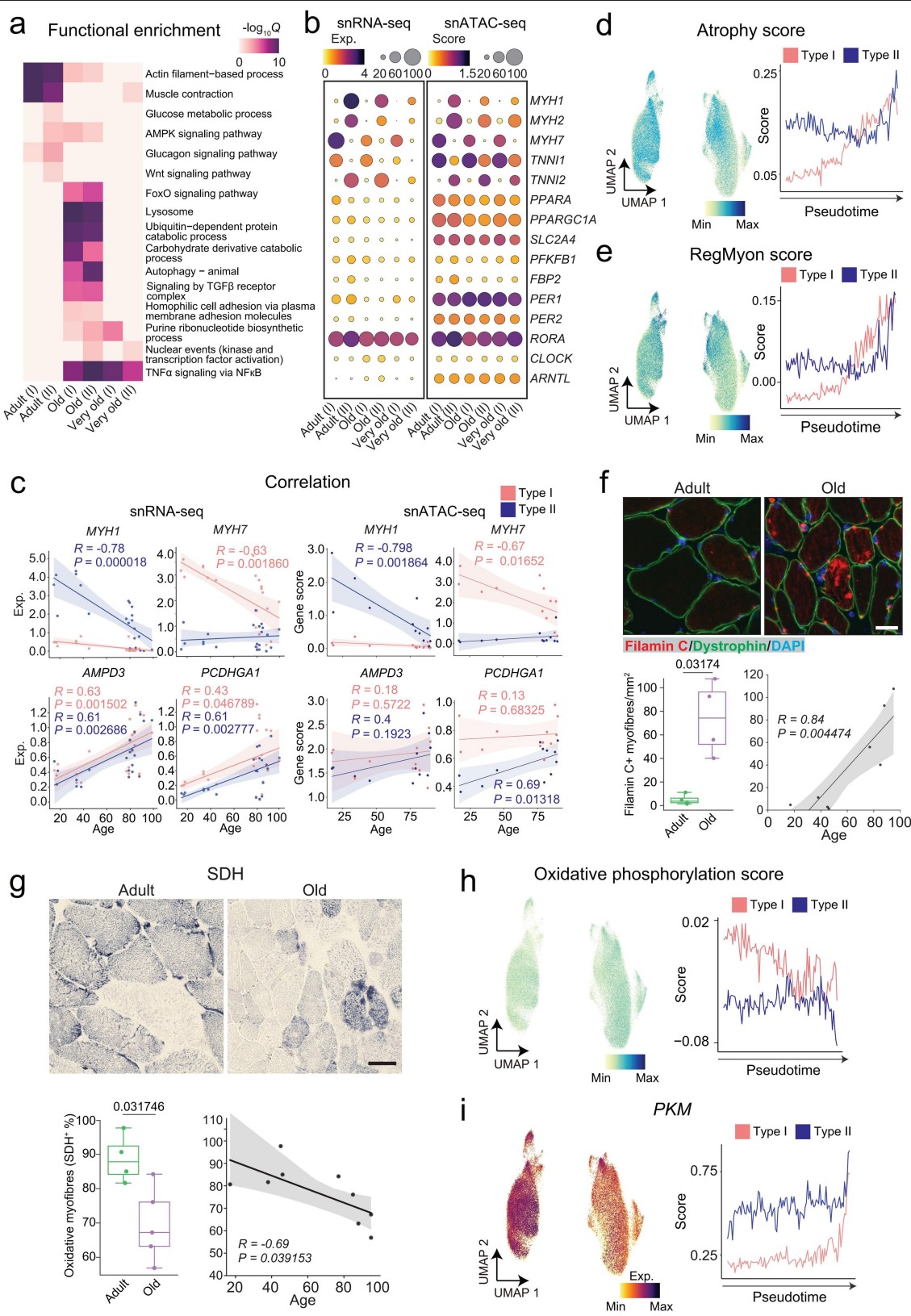

**Extended Data Fig. 6** | See next page for caption.

**Extended Data Fig. 6 | Functional processes driving type I and type II myofibre ageing. a**, Heatmap depicting enriched pathways of DEGs for both myonucleus types in adult (≤ 46 years), old (74-82 years) and very old (≥ 84 years) age groups. Pathway significance (-$\log_{10} Q$) is represented by the colour scale. **b**, Dot plots showing the gene expression (snRNA-seq, left panel) or gene score (snATAC-seq, right panel) for selected genes (rows) in adult, old and very old type I (I) and type II (II) myonuclei. Colour scale represents the average gene expression, and dot size, the percentage of cells expressing a given marker within the subpopulation. **c**, Correlation plots of the expression levels for the given genes (*MYH1, MYH7, AMPD3, PCDHGA1*) with the individual's age, for type I (red) and type II (blue) myonuclei in snRNA-seq (left panels) and chromatin accessibility levels for the corresponding genes in snATAC-seq (right panels). *R*, Pearson correlation coefficient; *P*, two-sided *P*-value; centre represents the average value; shade indicates 95% confidence interval. **d-e**, UMAP plots (left) or line charts (right) showing the (**d**) atrophy score (atrophy-related genes[24]; left panels) and (**e**) RegMyon score (regenerating myonucleus signature[19]; right panels) for type I (red) and type II myonuclei (blue) along the ageing trajectory. Gene lists for atrophy score and RegMyon score can be found in Supplementary Table 3. Dots (myonuclei) are coloured according to their computed score. **f**, Representative images (top panel), quantification comparing the age groups (bottom left panel) and correlation with the age of the individuals (bottom right panel) of immunofluorescence for filamin-C+ myofibres (filamin-C, red; myofibre membrane, dystrophin, green; nuclei, DAPI, blue) (left panel) in adult

(sample P13) and old (sample P28) individuals; boxplot showing the quantification for filamin-C+ myofibres (middle panel); correlation plot of the proportion of filamin-C+ myofibres. *n* = 4 individuals of each age group. Scale bar, 25 μm. For boxplots, boxes represent the upper/lower quartiles, the line depicts the median and whiskers the 1.5 interquartile range. *P* value was calculated by two-tailed Mann-Whitney *U*-test. For the correlation plot, *R*, Pearson correlation coefficient; *P*, two-sided *P*-value; centre represents the average value; shade indicates a 95% confidence interval. **g**, Representative images (top panel), quantification comparing the age groups (bottom left panel) and correlation with the age of the individuals (bottom right panel) for succinate dehydrogenase (SDH) activity staining (blue) in adult (sample P13) and old (sample P28) individuals. *n* = 4 for the adult group and *n* = 5 for old group. Scale bar, 50 μm. For boxplots, boxes represent the upper/lower quartiles, the line depicts the median and whiskers the 1.5 interquartile range. *P* value was calculated by two-tailed Mann-Whitney *U*-test. For the correlation plot, *R*, Pearson correlation coefficient; *P*, two-sided *P* value; centre represents the average value; shade indicates 95% confidence interval. **h-i**, UMAP plots (left) or line charts (right) showing the oxidative phosphorylation scores (**h**) and the gene expression level of *PKM* (**i**) as a representative gene for glycolysis pathway for type I (red) and type II myonuclei (blue) along the ageing trajectory. Gene lists for oxidative phosphorylation scores can be found in Supplementary Table 3. Dots (myonuclei) are coloured according to their computed score or gene expression value.

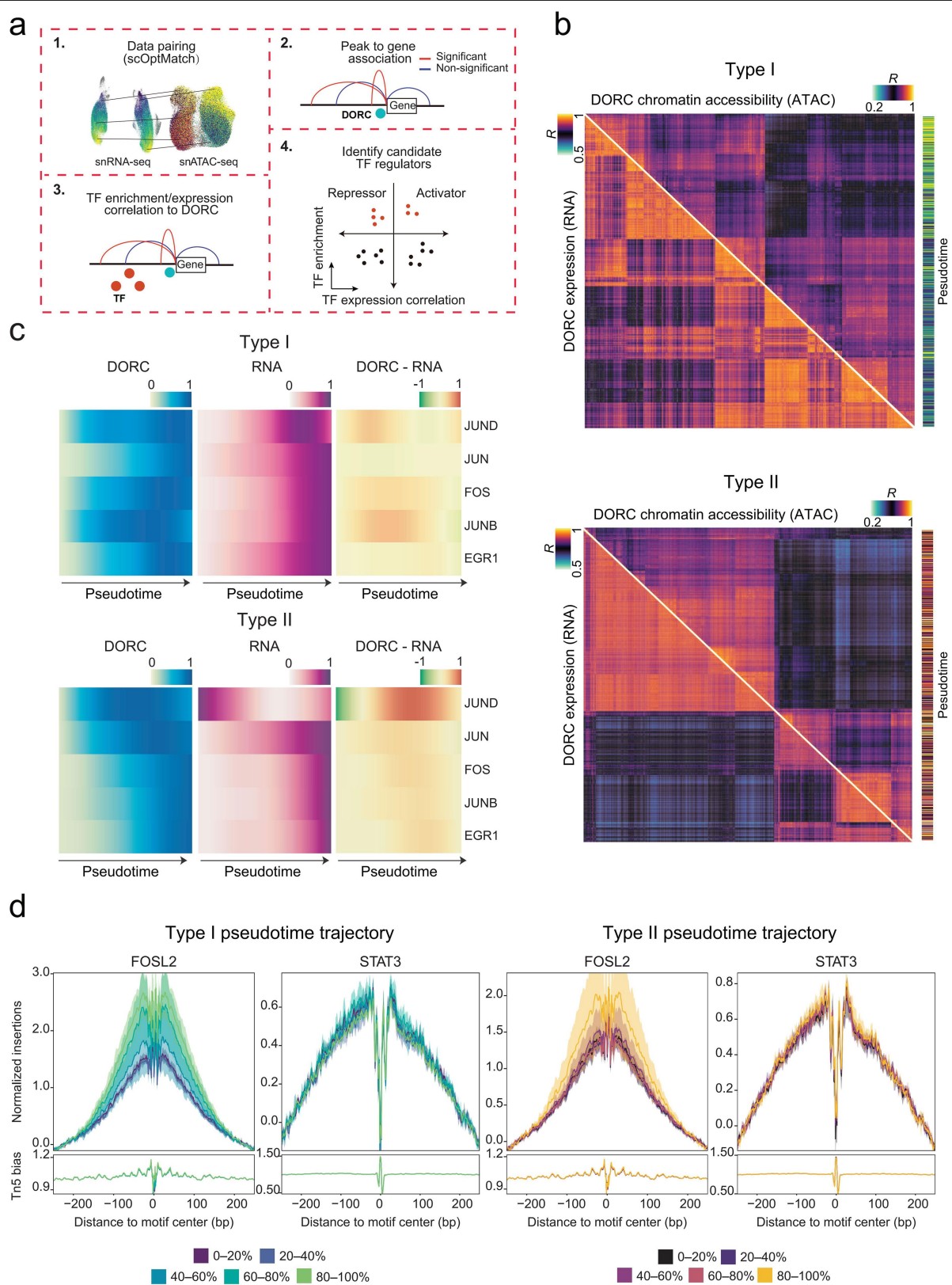

**Extended Data Fig. 7** | See next page for caption.

**Extended Data Fig. 7 | Gene regulatory network for ageing myofiber.**
**a**, Pipeline scheme for constructing gene-regulatory network: (1) nuclei of snRNA-seq and snATAC-seq datasets were paired using scOptMatch[31]; (2) identification of significant peak-to-gene associations for defining the domains of regulatory chromatin (DORC); (3) TF binding motif enrichment was evaluated in DORC regions; and (4) identification of potential TF regulators (repressors or activators) based on the correlation of TF expression and the motif enrichment. **b**, Correlation heatmap for the DORC expression (RNA) and chromatin accessibility (ATAC) for type I (top panel) and type II (bottom panel) myonuclei. Colour scale represents the computed Pearson correlation coefficient ($R$). **c**, Heatmap showing the smoothed normalized DORC accessibility (left panel), RNA expression (middle panel), and residual (DORC-RNA, right panel) levels for JUND, JUN, FOS, JUNB and EGR1 along the ageing trajectory for type I (top panel) and type II (bottom panel) myofibre. **d**, Tn5 bias-adjusted TF footprints for FOSL2 and STAT3 along the type I (left panel) and type II (right panel) ageing trajectory. The type I and type II myonuclei were classified into five proportions according to the pseudotime in the ageing trajectory. In each graph, the Tn5 bias-subtracted normalized insertions (y-axis) from the motif centre to 200 bp at each side (x-axis) were shown.

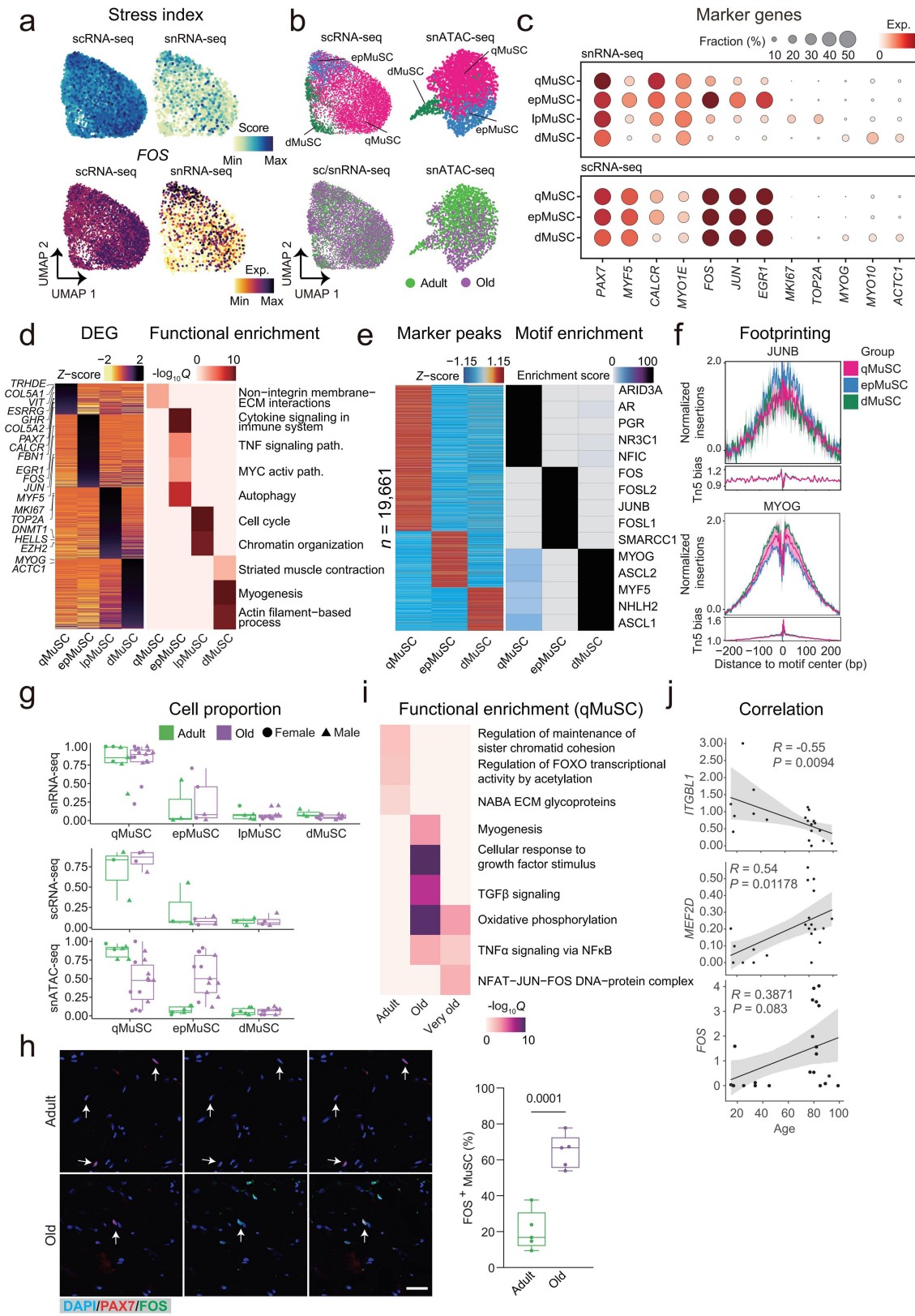

**Extended Data Fig. 8** | See next page for caption.

**Extended Data Fig. 8 | Multimodal human muscle stem cell atlas in ageing.** **a**, UMAP plots showing the level of stress index (top panels) and *FOS* (bottom panels) expression as representative genes contributing to this index for both scRNA-seq (left panels) and snRNA-seq (right panels) datasets. The gene list for the stress index can be found in Supplementary Table 3. Dots (cells/nuclei) are coloured according to their computed score or gene expression value. **b**, UMAP plots of the detected MuSC subpopulations (top panels) and age groups (bottom panels) in sc/snRNA-seq (left panels) and snATAC-seq (right panels). Dots (cells/nuclei) are coloured by subpopulation (top panels) or age group (bottom panels). **c**, Dot plots showing the expression level for representative marker genes (columns) for MuSC subpopulations (rows) by snRNA-seq (top panel) and scRNA-seq (bottom panel). Colour scale represents the average gene expression, and dot size, the percentage of cells expressing a given marker within the subpopulation. **d**, Heatmaps showing the expression levels for DEGs (left panel, colour scale by *Z*-score) and the corresponding functional enrichment analysis (right panel, colour scale by -$\log_{10} Q$) for each MuSC subpopulation in the snRNA-seq dataset. **e**, Heatmaps showing the marker peaks (left panel, colour scaled by *Z*-score) and the corresponding enriched TF motifs (right panel, colour scaled by -$\log_{10} Q$) for each MuSC subpopulation in the snATAC-seq dataset. **f**, Tn5 bias-adjusted TF footprints for JUNB (top panel) and MYOG (bottom panel) in each MuSC subpopulation detected by snATAC-seq.

In each graph, the Tn5 bias-subtracted normalized insertions (y-axis) from the motif centre to 200 bp at each side (x-axis) were shown. **g**, Cell proportion analysis for the percentage of each MuSC subpopulation in the snRNA-seq (top), scRNA-seq (middle) and snATAC-seq (bottom) for each individual. The colours represent the age group and the shapes represent the sex group. For scRNA-seq, $n$ = 3 adult individuals and $n$ = 4 old individuals; for snRNA-seq, $n$ = 7 adult individuals and $n$ = 15 old individuals; for snATAC-seq, $n$ = 5 adult individuals and $n$ = 11 old individuals. **h**, Representative images (left panel) and corresponding quantification (right panel) of immunofluorescence for primed MuSCs (PAX7, red; FOS, green; nuclei, DAPI, blue) in adult (sample P26, top left) and old (sample P28, bottom left) individuals. $n$ = 4 individuals of each age group. Scale bar, 25 μm. For boxplots, boxes represent the upper/lower quartiles, the line depicts the median and whiskers the 1.5 interquartile range. *P* value was calculated by a two-tailed Student's t-test. **i**, Heatmap depicting enriched pathways of the qMuSCs population in adult (≤ 46 years), old (74-82 years) and very old (≥ 84 years) age groups. Colour scale represents the significance (-$\log_{10} Q$) of the enriched terms. **j**, Correlation plots of the expression levels for the given genes (*ITGBL1*, *MEF2D* and *FOS*) with individuals' age for qMuSCs. *R*, Pearson correlation coefficient; *P*, two-sided *P*-value; centre represents the average value; shade indicates 95% confidence interval.

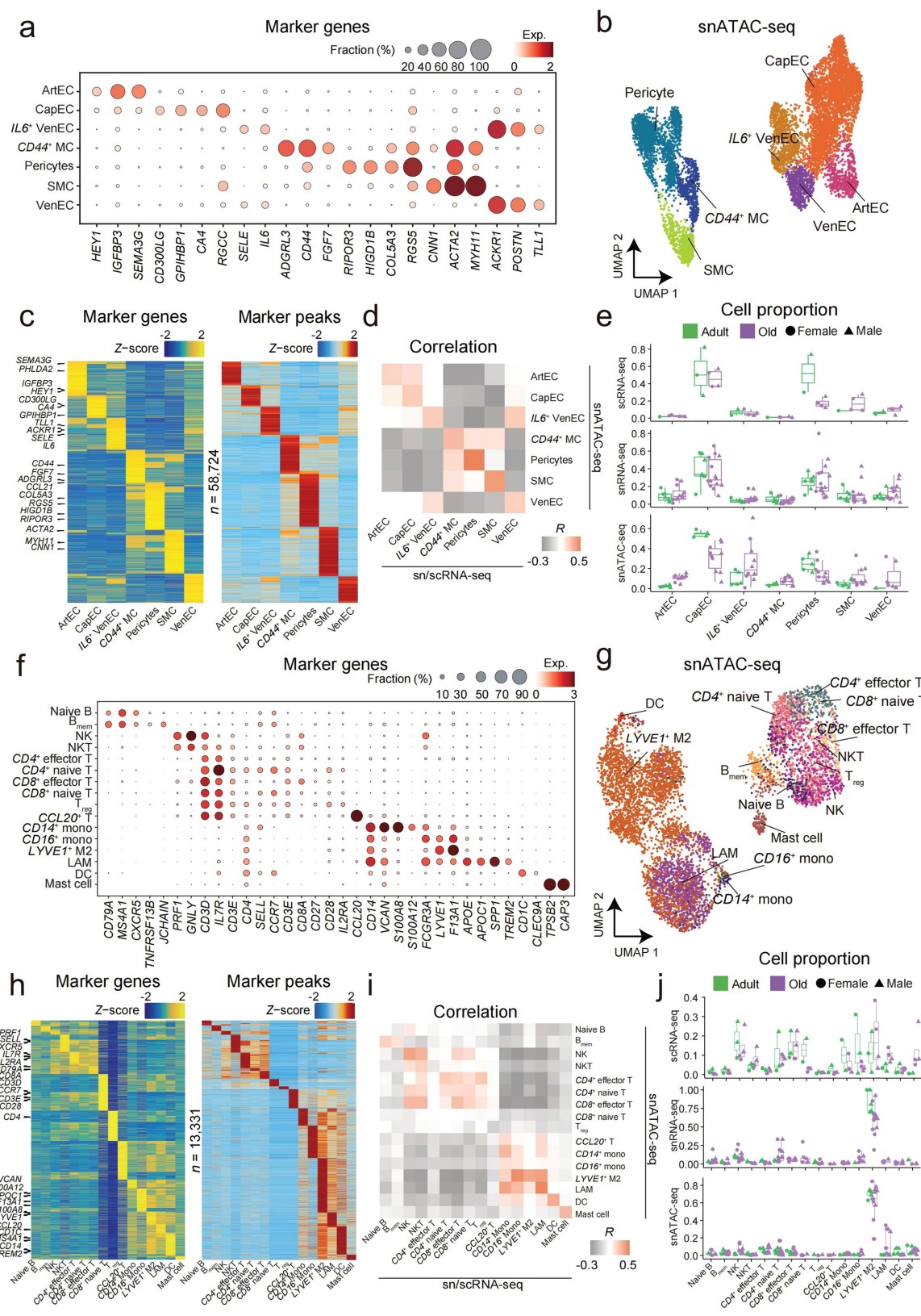

**Extended Data Fig. 9** | See next page for caption.

**Extended Data Fig. 9 | Vascular and immune compartment atlas of the ageing human skeletal muscle. a**, Dot plot showing the expression levels for representative marker genes (columns) for each vascular cell subpopulation (rows). Colour scale represents the average gene expression, and dot size, the percentage of cells expressing a given marker within the subpopulation. **b**, UMAP plot of the detected vascular cell subpopulations by snATAC-seq. Dots were coloured by the cell types. **c**, Heatmaps of the marker genes (gene score, left panel) and marker peaks (differentially accessible peaks, right panel) for each vascular subpopulation measured in snATAC-seq. Colour scale represents the relative score (Z-score) of each marker gene/peak. **d**, Heatmap showing the cell-type assignment correlation for vascular cell subpopulations between the sc/snRNA-seq and snATAC-seq datasets. Colour scale represents the computed Pearson correlation coefficient (R). **e**, Cell proportion analysis with the percentage of each vascular cell subpopulation for the scRNA-seq (top panel), snRNA-seq (middle panel) and snATAC-seq (bottom panel) for each individual. The colours represent the age group and the shapes represent the sex group. For scRNA-seq, $n = 3$ adult individuals and $n = 4$ old individuals; for snRNA-seq, $n = 7$ adult individuals and $n = 15$ old individuals; for snATAC-seq, $n = 5$ adult individuals and $n = 11$ old individuals. For boxplots, boxes represent the upper/lower quartiles, the line depicts the median and whiskers the 1.5 interquartile range. **f**, as in (**a**), for immune cell subpopulations. **g**, as in (**b**), for immune cell subpopulations. **h**, as in (**c**), for immune cell subpopulations. **i**, as in (**d**), for immune cell subpopulations. **j**, as in (**e**), for immune cell subpopulations. For scRNA-seq, $n = 3$ adult individuals and $n = 4$ old individuals; for snRNA-seq, $n = 5$ adult individuals and $n = 13$ old individuals; for snATAC-seq, $n = 4$ adult individuals and $n = 10$ old individuals.

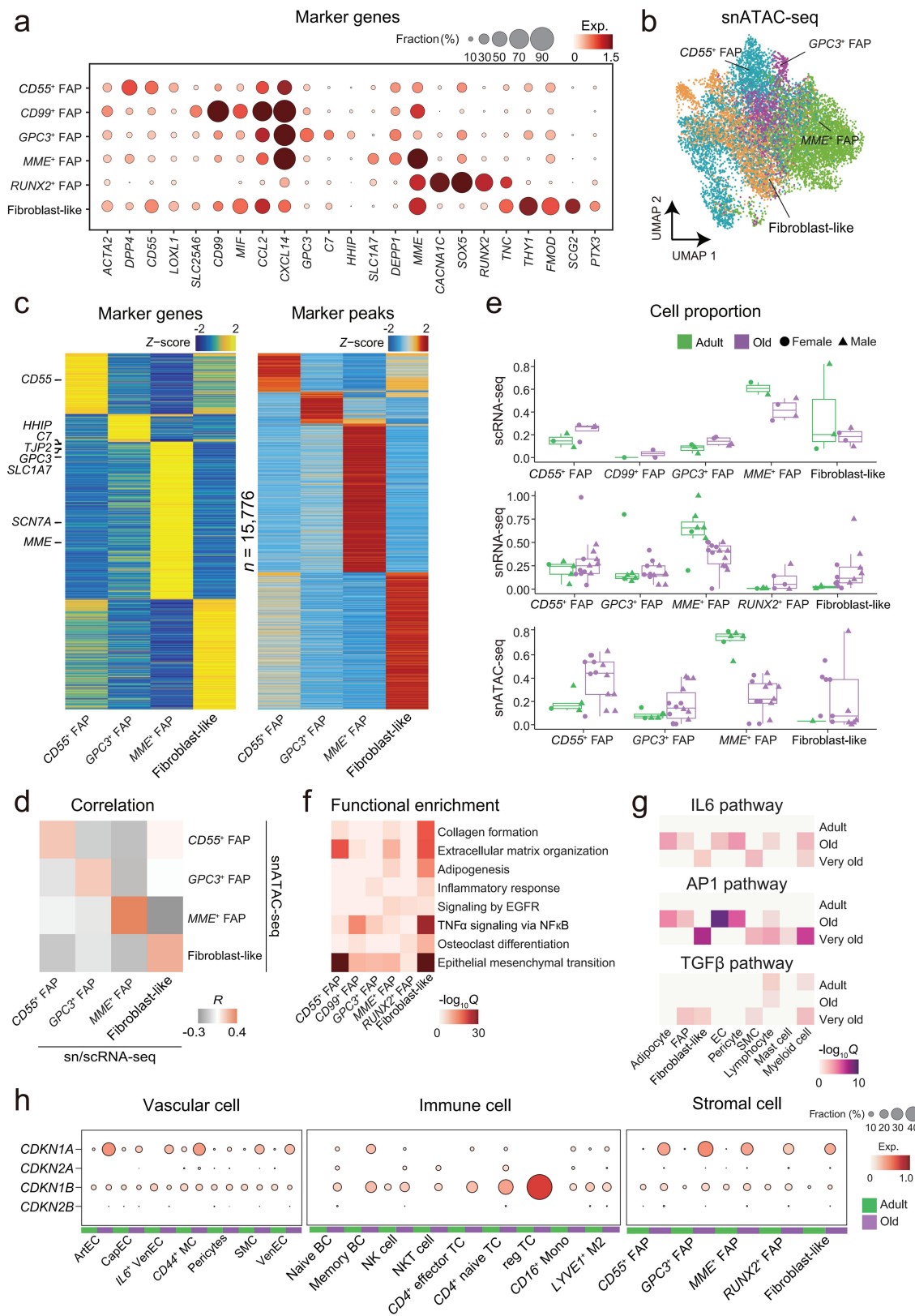

**Extended Data Fig. 10 |** See next page for caption.

**Extended Data Fig. 10 | Stromal compartment atlas of the ageing human skeletal muscle. a**, Dot plot showing the expression levels for representative marker genes for each stromal subpopulation (row). Colour scale represents the average gene expression, and dot size, the percentage of cells expressing a given marker within the subpopulation. **b**, UMAP plot of the detected stromal subpopulations in snATAC-seq. Dots were coloured by the cell types. **c**, Heatmaps of the marker genes (gene score, left panel) and marker peaks (differentially accessible peaks, right panel) for each identified population (columns). Colour scale represents the relative score (Z-score) for each marker gene/peak. **d**, Heatmap showing the cell-type assignment correlation for stromal cell subpopulations. Colour scale represents the Pearson correlation coefficient (R). **e**, Cell proportion analysis with the percentage of each stromal cell subpopulation in the scRNA-seq (top), snRNA-seq (middle) and snATAC-seq (bottom) for each individual. The colours represent the age group and the shapes represent the sex group. For scRNA-seq, $n = 3$ adult individuals and $n = 4$ old individuals; for snRNA-seq, $n = 7$ adult individuals and $n = 14$ old individuals; for snATAC-seq, $n = 5$ adult individuals and $n = 11$ old individuals. For boxplots, boxes represent the upper/lower quartiles, the line depicts the median and whiskers the 1.5 interquartile range. **f**, Heatmap depicting significantly enriched pathways for DEGs in each stromal cell subpopulation. Colour scale represents the significance ($-\log_{10} Q$) of the enriched terms. **g**, Heatmap depicting the enriched pro-inflammatory (IL6/AP1) and pro-fibrotic (TGFβ) pathways of the resident mononucleated cell subpopulations in adult ($\leq 46$ years), old (74-82 years) and very old ($\geq 84$ years) age groups. Colour scale represents the significance ($-\log_{10}Q$) of the enriched terms. **h**, Dot plots showing the expression levels for the indicated cell-cycle inhibitor genes. Colour scale represents the average gene expression, and dot size, the percentage of cells expressing a given gene within the subpopulation.

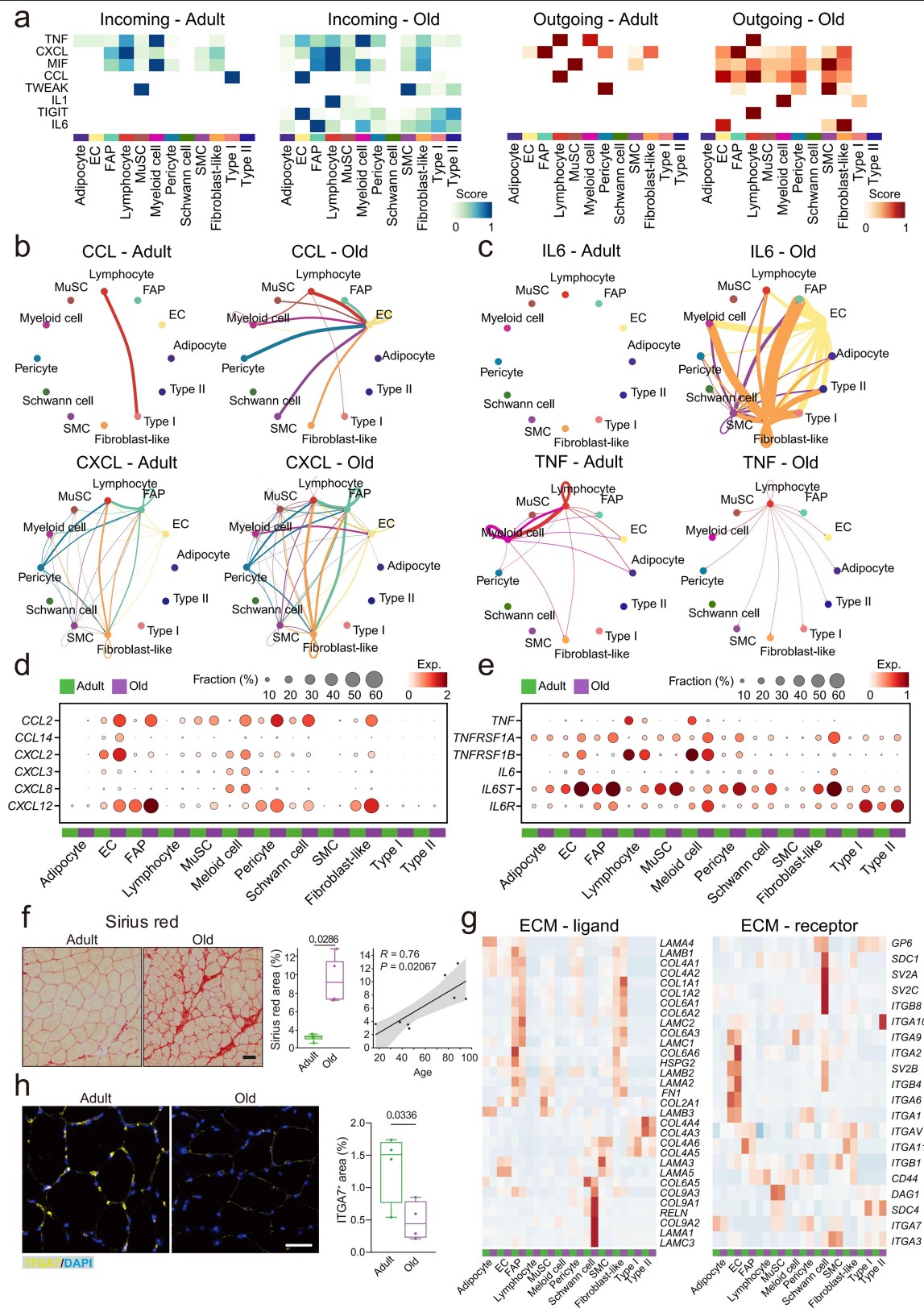

**Extended Data Fig. 11** | See next page for caption.

**Extended Data Fig. 11 | Inflammatory and cell-ECM interaction networks of skeletal muscle with ageing. a**, Heatmaps of the incoming (white-blue scale, left panel) and outgoing (white-red scale, right panel) signals for adult (left side) and old (right side) for indicated signalling pathways. Colour scale represents the relative interaction intensity of each cell type for the represented pathways. **b-c**, (**b**) Chemokines (CCL, top panel; and CXCL, bottom panel) and (**c**) inflammatory factors (IL6, top panel; and TNF, bottom panel) signalling networks in adult (left panels) and old (right panels) groups. The nodes represent cell types and edges the interactions among them. Edge width is proportional to the interaction probability. Nodes and edges are coloured according to the cell type. **d-e**, Dot plot showing the expression level for the representative (**d**) chemokines and (**e**) inflammatory factors in adult (green) and old (purple) skeletal muscle cell populations. Colour scale represents the average gene expression, and dot size, the percentage of cells expressing a given gene within the subpopulation. **f**, Representative images (left panel) of adult (sample P26) and old (sample P28) muscles of Sirius red staining, quantification comparing the age groups (middle panel) and correlation with the age of the individuals (right panel). $n$ = 4 individuals of each age group. Scale bar, 50 μm. For boxplots, boxes represent the upper/lower quartiles, the line depicts the median and whiskers the 1.5 interquartile range. $P$ value was calculated by a two-tailed Mann-Whitney $U$-test. For the correlation plot, $R$, Pearson correlation coefficient; $P$, two-sided $P$ value; centre represents the average value; shade indicates a 95% confidence interval. **g**, Heatmaps of the relative gene expression ($Z$-score) of ECM-ligands (left panel) and ECM-receptors (right panel) in each major cell population in both adult (green) and old (purple) individuals. **h**, Representative images (left panel) and corresponding quantification (right panel) of immunofluorescence for ITGA7$^+$ area (ITGA7, yellow; nuclei, DAPI, blue) in adult (sample P5) and old (sample P29) individuals. Scale bar, 50 μm. $n$ = 4 individuals of each age group. For boxplots, boxes represent the upper/lower quartiles, the line depicts the median and whiskers the 1.5 interquartile range. $P$ value was calculated by a two-tailed Student's t-test.

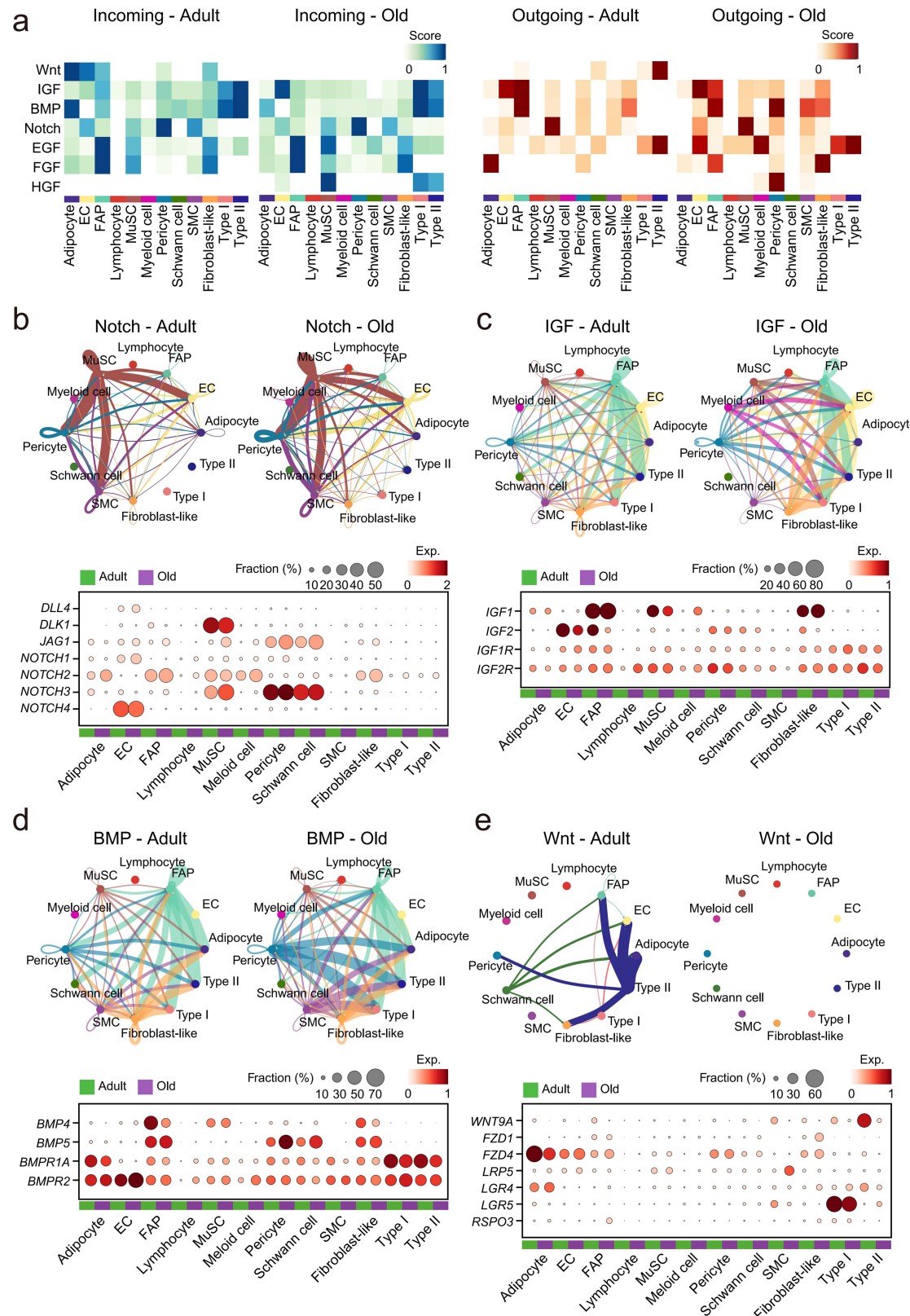

**Extended Data Fig. 12 | Perturbed Notch, IGF, BMP and Wnt signalling networks in aged muscles. a**, Heatmaps of the incoming (white-blue scale, left panel) and outgoing (white-red scale, right panel) signals for adult (left side) and old (right side) for indicated signalling pathways. Colour scale represents the relative interaction intensity of each cell type for the represented pathways. **b-e**, Notch (**b**), IGF (**c**), BMP (**d**), and Wnt (**e**) signalling networks (top panels) and the expression levels for the corresponding representative genes in each

signalling network (bottom panels). Signalling networks are depicted for adult (left top panels) and old (right top panels). The nodes represent cell types and edges the interactions among them. Edge width is proportional to the interaction probability. Nodes and edges are coloured according to the cell type. For dot plots, the colour scale represents the average gene expression and dot size the percentage of cells expressing a given gene within the cell type in adult (green) and old (purple) individuals.

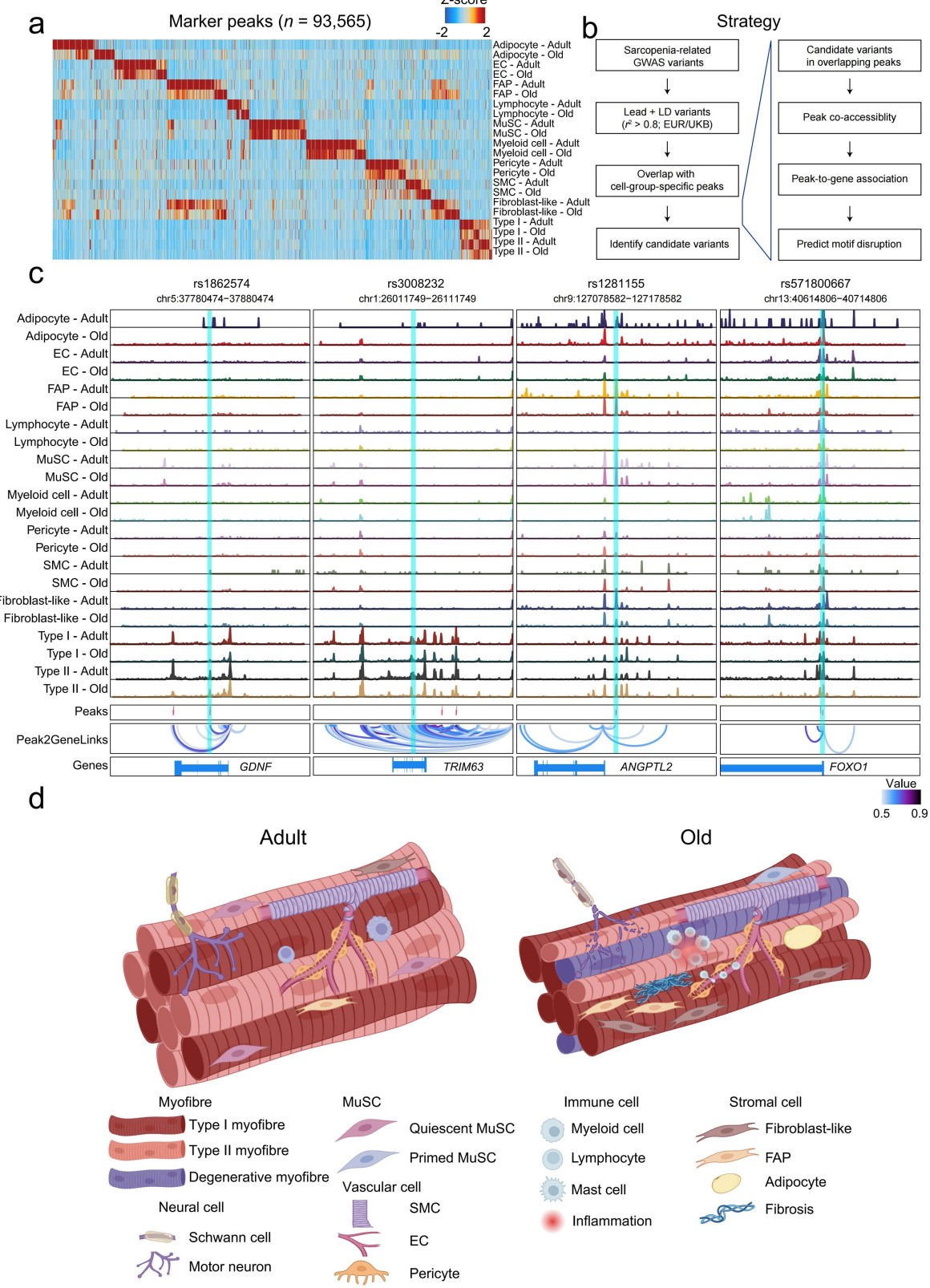

**Extended Data Fig. 13** | See next page for caption.

**Extended Data Fig. 13 | GWAS analysis of ageing muscle and spatial distribution of the human skeletal muscle changes with ageing. a**, Heatmap of the ageing-related marker peaks in each cell type identified by snATAC-seq. Colour scale represents the relative score (*Z*-score) of each marker peak. **b**, Pipeline scheme for identification of candidate sarcopenia-related genetic variants. Briefly, genetic variants with the high linkage disequilibrium (LD, $r^2 \geq 0.8$) to reported lead variants were overlapped to cell-type defined marker peaks to identify candidate variants. Next, the potential effects of candidate variants were assessed by the following steps: 1) identification of the co-accessible peaks; 2) construction of peak to gene linkage 3) predicting the effect of genetic variants on TF binding. **c**, Genome browser tracks for each cell population in adult and old muscles showing the peaks associated with the identified SNPs rs1862574, rs3008232, rs1281155 and rs571800667 within the locus of *GDNF*, *TRIM63*, *ANGPTL2* and *FOXO1*, respectively. Chromatin tracks show the linkages between the variant-containing peaks and the associated gene promoter. **d**, (**Adult**) Human skeletal muscles are mainly composed of syncytial multinucleated type II and type I myofibres, each of which is innervated by single motorneuron endplates whose axons are surrounded by Schwann cells.

Muscle resident mononuclear cell populations, such as MuSCs, stromal cells (FAPs and fibroblast-like cells), immune cells (myeloid cells and lymphocytes) and vascular cells (ECs, SMCs and pericytes), contribute to muscle homeostasis. (**Old**) Human skeletal muscle ageing is characterized by a reduction in mass, increased inflammation and fibrosis. Myofibre numbers are reduced, particularly type II myofibres. In addition, myofibres tend to lose their innervation and change their characteristics, and new myonuclear populations arise with abnormal regenerative, degenerative and denervation traits. The MuSC population is reduced and enters a primed state, leading to exhaustion of the stem cell pool. Intramuscular adipocytes accumulate among muscle fibres, and stromal cells increase, with a shift towards a pro-fibrotic profile. In the vasculature, the proportion of pericytes and capillary ECs decrease, and vascular integrity is impaired. Immune cells (myeloid cells, lymphocytes and mast cells) infiltrate the muscle and acquire a pro-inflammatory profile. Collectively, these alterations and the associated abnormalities in signalling networks impact muscle characteristics and functions. The schematic was created with BioRender.com.

# Reporting Summary

## Statistics

For all statistical analyses, confirm that the following items are present in the figure legend, table legend, main text, or Methods section.

| n/a | Confirmed | |
|---|---|---|
| ☐ | ☒ | The exact sample size (*n*) for each experimental group/condition, given as a discrete number and unit of measurement |
| ☒ | ☐ | A statement on whether measurements were taken from distinct samples or whether the same sample was measured repeatedly |
| ☐ | ☒ | The statistical test(s) used AND whether they are one- or two-sided<br>*Only common tests should be described solely by name; describe more complex techniques in the Methods section.* |
| ☐ | ☒ | A description of all covariates tested |
| ☒ | ☐ | A description of any assumptions or corrections, such as tests of normality and adjustment for multiple comparisons |
| ☐ | ☒ | A full description of the statistical parameters including central tendency (e.g. means) or other basic estimates (e.g. regression coefficient) AND variation (e.g. standard deviation) or associated estimates of uncertainty (e.g. confidence intervals) |
| ☐ | ☒ | For null hypothesis testing, the test statistic (e.g. *F*, *t*, *r*) with confidence intervals, effect sizes, degrees of freedom and *P* value noted<br>*Give P values as exact values whenever suitable.* |
| ☒ | ☐ | For Bayesian analysis, information on the choice of priors and Markov chain Monte Carlo settings |
| ☒ | ☐ | For hierarchical and complex designs, identification of the appropriate level for tests and full reporting of outcomes |
| ☐ | ☒ | Estimates of effect sizes (e.g. Cohen's *d*, Pearson's *r*), indicating how they were calculated |

*Our web collection on statistics for biologists contains articles on many of the points above.*

## Software and code

Policy information about availability of computer code

| Data collection | DIPSEQ T1 sequencer for droplet-based sn/scRNA-seq. BGISEQ-2000 sequencer for snATAC-seq. Zeiss 980 AiryScan2, Leica DMR6000B or Nikon Ti2 for fluorescence microscope imaging. |
|---|---|
| Data analysis | Sequencing data were analyzed using: Cutadapt (v1.14), FastQC (v0.11.2), fastp (v0.21.0), STAR (v2.7.4a), PISA (v0.2), sambamba (v0.7.0), Seurat (v4.0.2), Scanpy (v1.8.1), CellChat (v1.1.0), ArchR (v0.9.5), Python (v3.7), DoubletFinder (v2.0.3), FigR (v0.1.0), SoupX (v1.4.8), pheatmap (v1.0.12), FUMA (v1.5.4), Metascape (v3.0), LDSC (v1.0.1), Hotspot (v1.1.1)<br>Imaging data were analyzed using: LAS AF (v4.0), ZenBlue (v3.5), NIS Elements (v4.11.0), Fiji (ImageJ2) (v2.14.0/1.54f).<br>Custom code supporting this study is available at:<br>https://github.com/MGI-tech-bioinformatics/DNBelab_C_Series_HT_scRNA-analysis-software<br>https://github.com/123anjuan/HMA |

For manuscripts utilizing custom algorithms or software that are central to the research but not yet described in published literature, software must be made available to editors and reviewers. We strongly encourage code deposition in a community repository (e.g. GitHub). See the Nature Portfolio guidelines for submitting code & software for further information.

## Data

Raw data can be found at CNGB Nucleotide Sequence Archive (accession code: CNP0004394, CNP0004395, CNP0004494, CNP0004495) and processed data at Human Muscle Ageing Cell Atlas database (https://db.cngb.org/cdcp/hlma/).
Databases used in this study are: CisBP (http://cisbp.ccbr.utoronto.ca/); SNP2TFBS (https://epd.expasy.org/snp2tfbs/); CellChat (http://www.cellchat.org/); GWAS Catalog (https://www.ebi.ac.uk/gwas/).

## Research involving human participants, their data, or biological material

| | |
|---|---|
| Reporting on sex and gender | Sex was determined based on participants self-reporting. Sex was considered as a covariate in statistical analysis. Details are provided in Supplementary Table 1. |
| Reporting on race, ethnicity, or other socially relevant groupings | Ethnicity or race was not asked to patients, it was determined by the geographic location. |
| Population characteristics | Participants: 18 patients for the European cohort, and 13 patients for the Chinese cohort. All details are provided in Supplementary Table 1. |
| Recruitment | Participants were recruited before orthopedic surgery with informed consent. Before obtaining the informed written consent, the medical team provided both verbal and written information about the research study. For the European cohort, participants were informed about the functional state assessment with the Barthel Index and Charlson Index. Details are provided in Supplementary Table 1. |
| Ethics oversight | The study was performed in accordance with the Declaration of Helsinki. Ethical approval was granted for the European cohort by the Research Ethics Committee of Hospital Arnau de Vilanova (CEIm 28/2019), and for the Chinese cohort by the Institutional Ethics Committee of the First Affiliated Hospital of Guangdong Pharmaceutical University, Guangzhou (China) (2020-ICE-90). |

Note that full information on the approval of the study protocol must also be provided in the manuscript.

# Field-specific reporting

Please select the one below that is the best fit for your research. If you are not sure, read the appropriate sections before making your selection.

☒ Life sciences          ☐ Behavioural & social sciences          ☐ Ecological, evolutionary & environmental sciences

For a reference copy of the document with all sections, see nature.com/documents/nr-reporting-summary-flat.pdf

# Life sciences study design

All studies must disclose on these points even when the disclosure is negative.

| | |
|---|---|
| Sample size | Sample size was not predetermined. 22 individuals (387,444 nuclei/cells ) were analyzed in this study. |
| Data exclusions | We used several criteria to filter low-quality cells or nuclei during data analysis: UMI < 1,000, gene < 500, and mitochondria content > 5% in sn/scRNA-seq, and TSS enrichment scores < 5, number of fragments < 1,000  in snATAC-seq. |
| Replication | The number of individuals used for profiling in replication for each platform is listed below:<br>scRNA-seq: 8 (in total 79,649 cells).<br>snRNA-seq: 22 (in total 212,774 nuclei).<br>snATAC-seq: 17 (in total 95,021 nuclei).<br>For immunostainning or immunohistology, the numbers of replicates are indicated in the corresponding figure legend. |
| Randomization | Not relevant because there was no group allocation. |
| Blinding | There was no blinding performed because there was no group allocation. |

# Reporting for specific materials, systems and methods

We require information from authors about some types of materials, experimental systems and methods used in many studies. Here, indicate whether each material, system or method listed is relevant to your study. If you are not sure if a list item applies to your research, read the appropriate section before selecting a response.

## Materials & experimental systems

| n/a | Involved in the study |
|---|---|
| ☐ | ☒ Antibodies |
| ☒ | ☐ Eukaryotic cell lines |
| ☒ | ☐ Palaeontology and archaeology |
| ☐ | ☒ Animals and other organisms |
| ☒ | ☐ Clinical data |
| ☒ | ☐ Dual use research of concern |
| ☒ | ☐ Plants |

## Methods

| n/a | Involved in the study |
|---|---|
| ☒ | ☐ ChIP-seq |
| ☒ | ☐ Flow cytometry |
| ☒ | ☐ MRI-based neuroimaging |

## Antibodies

| Antibodies used | Primary antibodies:<br>PAX7 (Developmental Studies Hybridoma Bank cat: PAX7, 1:50)<br>PDGFRa(eBioscience cat: 17-1401-81, 1:100)<br>Perilipin (Cell Signalling cat: 9349, 1:100)<br>Filamin C (MyBiosource cat: MBS2026155, 1:100)<br>TNNT2 (Bioss cat:10648R-A488, 1:100)<br>CD11b (eBioscience cat: 14-0112-85, 1:100)<br>CD3 (Invitrogen cat: 14-0038-82, 1:100)<br>CD19 (Invitrogen cat: 14-0199-82, 1:100)<br>NCAM1 (Cell Sciences cat: Mon9006-1, 1:100)<br>MyHC Type I (Developmental Studies Hybridoma Bank cat: A4.840)<br>MyHC Type II (Developmental Studies Hybridoma Bank cat: SC-71, 1:70)<br>Laminin-647 (Novus Biologicals cat: NB300-144AF647, 1:200)<br>cFOS (Cell Signaling cat: #2250S, 1:200)<br>ACVR2A (R&D cat: AF340, 1:100)<br>ITGA7 (Biocell Scientific cat: 10007, 1:100)<br>DYSTROPHIN (Sigma cat: D8168, 1:100)<br><br>Secondary antibodies (Invitrogen, Carlsbad, CA, USA) were coupled to Alexa Fluor 488, 568, 594 or 647 fluorochromes are listed below diluted at 1:500:<br>Goat anti-Mouse IgM Cross-Adsorbed Secondary Antibody, DyLight™ 550 (Invitrogen™ cat: SA5-10151)<br>Goat Anti-Mouse IgG1 Cross-Adsorbed Secondary Antibody Alexa Fluor 488 (Invitrogen™ cat: A21121)<br>Goat anti-Mouse IgG (H+L) Cross-Adsorbed Secondary Antibody, Alexa Fluor™ 488 (Invitrogen™ cat: A11001)<br>Goat anti-Mouse IgG (H+L) Cross-Adsorbed Secondary Antibody, Alexa Fluor™ 568 (Invitrogen™ cat: A11004)<br>Goat anti-Rabbit IgG (H+L) Highly Cross-Adsorbed Secondary Antibody, Alexa Fluor™ Plus 488 (Invitrogen™ cat: A32731TR)<br>Goat anti-Rabbit IgG (H+L) Highly Cross-Adsorbed Secondary Antibody, Alexa Fluor™ Plus 647 (Invitrogen™ cat: A32733TR)<br>Donkey anti-Goat IgG (H+L) Highly Cross-Adsorbed Secondary Antibody, Alexa Fluor™ Plus 647 (Invitrogen™ cat: A32849TR)<br>Goat anti-Rat IgG (H+L) Cross-Adsorbed Secondary Antibody, Alexa Fluor™ 568 (Invitrogen™ cat: A11077) |
|---|---|
| Validation | All antibodies are commonly used in the field and have been validated by the manufacturer. For the primary antibodies, the following information were provided by the supplier:<br>PAX7 (Developmental Studies Hybridoma Bank cat: PAX7, 1:50), reacts with: Amphibian, Avian, Axolotl, Bovine, Canine, Fish, Goat, Human, Mouse, Ovine, Porcine, Quail, Rat, Turtle, Xenopus, Zebrafish; suitable for; Chromatin Immunoprecipitation, FACS, FFPE, Gel Supershift, Immunofluorescence, Immunohistochemistry, Immunoprecipitation, Western Blot; validated in adult skeletal muscle satellite cells.<br>PDGFRa(eBioscience cat: 17-1401-81, 1:100), reacts with: Human, Mouse; suitable for IHC, ICC/IF, Flow cytometry; validated in NIH/3T3 cells.<br>Perilipin (Cell Signalling cat: 9349, 1:100), reacts with: Human, Mouse; suitable for Western Blot, Immunoprecipitation, Immunohistochemistry, Chromatin Immunoprecipitation, CUT&RUN, CUT&Tag, Dot Blot, eCLIP, Immunofluorescence, Flow Cytometry; validated in adipocyte.<br>Filamin C (MyBiosource cat: MBS2026155, 1:100), reacts with: Human; suitable for Western Blot, Immunohistochemistry; validated in Human breast cancer tissue, Human prostate tissue, Human colorectal cancer tissue, Human prostate tissue.<br>TNNT2 (Bioss cat:10648R-A488, 1:100), reacts with: Human, Mouse, Rat; suitable for WB, FCM, IF(IHC-P), IF(IHC-F), IF(ICC); validated in cardiac muscle.<br>CD11b (eBioscience cat: 14-0112-85, 1:100), reacts with Rabbit, Bat, Fish, Mouse, Human; suitable for Western Blot, Immunohistochemistry, Immunocytochemistry, Immunoprecipitation, Flow Cytometry; validated in mouse splenocytes.<br>CD3 (Invitrogen cat: 14-0038-82, 1:100), reacts with: Rat, Human, Mouse; suitable for Western Blot, Immunohistochemistry, Immunocytochemistry, Immunoprecipitation, Flow Cytometry; validated in lymphocytes.<br>CD19 (Invitrogen cat: 14-0199-82, 1:100), reacts with: Human; suitable for Immunohistochemistry, Flow Cytometry; validated in B-lymphocyte. |

NCAM1 (Cell Sciences cat: Mon9006-1, 1:100), reacts with: Human; suitable for IHC-F, IHC-P; validated in small cell carcinomas and carcinoids of the lung.
MyHC Type I (Developmental Studies Hybridoma Bank cat: A4.840), reacts with: Rat, Human, Mouse; suitable for IHC, WB, IF, ICC; validated in skeletal muscle.
MyHC Type II (Developmental Studies Hybridoma Bank cat: SC-71, 1:70), reacts with: Rat, Human, Mouse; suitable for IHC, WB, IF, ICC; validated in skeletal muscle.
Laminin-647 (Novus Biologicals cat: NB300-144AF647, 1:200), reacts with: Rat, Human, Mouse, Rabbit, Fruit Bat, Chinese Hamster; suitable for WB, Flow, ICC/IF, IHC, IHC; validated in epithelial tissue, nerve, fat cells and smooth, striated and cardiac muscle.
cFOS (Cell Signaling cat: #2250S, 1:200), reacts with: Human, Mouse, Rat; suitable for Western Blot, Immunoprecipitation, Immunohistochemistry, Chromatin, Immunoprecipitation, CUT&RUN, CUT&Tag, eCLIP, Immunofluorescence, Flow Cytometry; validated in fibroblasts.
ACVR2A (R&D cat: AF340, 1:100), reacts with: Human; suitable for Western Blot, Immunohistochemistry; validated in Human prostate cancer tissue.
ITGA7 (Biocell Scientific cat: 10007, 1:100), reacts with: Human, Mouse, Rat; suitable for Western Blot, Immunohistochemistry; validated in rodent and human tissues.
DYSTROPHIN (Sigma cat: D8168, 1:100), reacts with: Chicken, Rat, Human, Pig, Rabbit, Mouse; suitable for Western Blot, Immunohistochemistry; validated in mouse muscle.

# Animals and other research organisms

Policy information about studies involving animals; ARRIVE guidelines recommended for reporting animal research, and Sex and Gender in Research

| Laboratory animals | C57Bl/6 (WT) mice were bred and raised till 8-12 weeks of age at the animal facility of the Barcelona Biomedical Research Park (PRBB). They were housed in standard cages with a 12-hour light-dark cycle and given unrestricted access to a standard chow diet. |
|---|---|
| Wild animals | N/A |
| Reporting on sex | Both males and females were used for experiments and were maintained according to the Jackson Laboratories's guidelines and protocols. |
| Field-collected samples | N/A |
| Ethics oversight | All experiments adhered to the 'three Rs' principle—replacement, reduction, and refinement—outlined in Directive 63/2010 and its implementation in Member States. Procedures were approved by the PRBB Animal Research Ethics Committee (PRBB-CEEA) and the local government (Generalitat de Catalunya), following European Directive 2010/63/EU and Spanish regulations RD 53/2013. |

Note that full information on the approval of the study protocol must also be provided in the manuscript.

