## [Peer Review File · Nature]

Manuscript Title: Multimodal cell atlas of the ageing human skeletal muscle

Reviewer Comments & Author Rebuttals

Reviewer Reports on the Initial Version:

Referees' comments:

Referee #1 (Remarks to the Author):

1) I've never seen so many authors listed as "equally contributed." It doesn't seem mathematically possible that all these people contributed the same amount. Also it would seem to hurt those people who really deserve "first author" standing to do this. Just a suggestion... it's the senior authors' call, but this seems quite unusual.

2) As an over-arching point, I think this paper needs to be more focused, with actually fewer figures - but those figures should have more depth, and supported with tables showing genes that are associated with the pathways noted. I really don't think it helps the authors to show 17 Extended Data Figures; given all the work that goes into each figure, it would be a shame if some additional important points are simply lost due to the mass of figures being dumped in this paper. This should be taken as friendly advice; I don't think the ever-expanding proliferation of data in a single paper really helps the scientists deliver a few clear messages and findings.

For example, at least for me, the biggest interest are the particular genes that are differentially regulated in the various nuclei as a function of age, which might point to age-regulated mechanisms that are impinging on muscle function. This sort of finding is really diluted and difficult to induce given the current data presentation. As another example of this, consider the single sentence on page 6, that old myonuclei are enriched in atrophy related genes, with 4 such genes given as an example: Fox, Jun, Fbxo32, AMPD3. Among these, only Fbxo32 is purely an atrophy-associated genes, and the authors point to 3 separate extended figures to make this point: Extended data Fig 5a, 6d, and e. It seems the actual data might also be in Suppl Table 4, but that table is never discussed or mentioned in the entire paper.

Only extended data fig 5a is really getting to this - it's the heat map - and a heat map is really not sufficient, because it's very difficult to go through each column and understand which gene is upregulated; a table for this of all the atrophy-associated genes would be much more helpful. Is this what Supplementary Table 4 is? As noted, there is zero discussion of that table in the figure (or paper), but it's the one that contains genes that purport to be part of the degenerative phenotype (at least by the title of the table). If this table has this data, it needs to be extensively discussed in the paper.

When I finally got to extended data figure 7 d, this seems like the heart of the paper, so I wonder why it's buried in extended figures, and isn't in the main paper. Isn't this the figure that really shows the age-perturbed signaling pathways in the various clusters?

As one more over-arching point, which is listed again below - there's not a single mention of senescence in this paper. Was evidence of senescence not found in aged human muscle? Certainly many of the pathways perturbed and genes found are consistent with the presence of senescent cells, according to prior papers.

3) As the authors note, there have been prior single-cell analyses of skeletal muscle, including in aging, and prior single-cell analysis of skeletal muscle in the non-aged human, so it would seem the "novelty" here would be the comparison of many humans throughout a wide aging range, and the possibility of correlating findings to sarcopenia. The authors agree that this is the point of the study (pg 5 of the manuscript).

4) Given the desire to find differentially-expressed genes associate with sarcopenia, it's not obvious that the methods employed here are the best approaches.

Old-fashioned bulk RNA seq would more comprehensively show sarcopenia-associated genes, since it would allow for a much deeper analysis, whereas single cell RNA seq limits one to the top 5000 or so highest-expressed genes. This lets you determine cell types, and perhaps most obvious mechanisms, but does not allow for a full characterization of genes which are specifically expressed in sarcopenia.

5) Was there really no effect of gender on the findings? (Extended Data Fig. 2a-c) This seems surprising given the females would be undergoing menopause, and sex hormones do effect skeletal muscle. There do seem to be clusters that are relatively dominated by male, just from looking at the figures - or is that a function of the number of males vs females? (for example, look at Extended Data, Figure 2b, male vs female, the top cluster.)

6) The ENOX1 myonuclei specific for type II myofibres, enriched in the young group, is quite interesting, as is the TNNT2+ myonuclei, and the three additional populations enriched in the old group. For these nuclei, were there any differences besides relative abundance as a function of age? In other words, were there co-located transcripts that one could say were significantly differentially expressed in these nuclei, with the understanding that this is a question better left for bulk RNA-seq? - or is this what is shown in Figure 2h? There's just one sentence devoted to 2h, but if this is really a discussion of differential gene expression in young vs old, this should be explained and expanded on quite a bit, as to what genes in particular are relatively perturbed vs TNNT2 in these nuclei. As noted above, a table of the differentially regulated genes as a function of age in these nuclei would help the authors make their point more easily, and these genes could also be annotated as to the pathways they fall in.

7) The authors label TNNT2 expression as pathological, but it is found in the young - it's just found more in the old, consistent with the nuclei as a class being more abundant. Doesn't the finding in the young suggest these nuclei are not necessarily pathological, or do the authors think that loss of innervation is even happening in the young, just at a lesser extent?

8) The authors in page 7 refer to the emergence of new nuclei, but is there an example of an entirely

new type of nuclei that appears in the older muscle? The prior discussions all point to relative enrichments - or did I miss a figure that shows something that purely found in older subjects? Later on that same page they refer to "the emergence of specific end-stage myonuclear populations, expressing degenerative traits", but they don't reference a figure showing this. What figures shows that? Or do the authors mean that myonuclear populations also found in the young develop degenerative traits with age? (that's not the same as claiming there are novel degenerative nuclei forming).

9) As noted above, Extended Data Figure 7 d seems to be the most important and helpful figure, because it shows the pathways that are functionally enriched due to age. I'm confused by the arrows going in opposite directions in Type I vs Type II fibers; why is the data presented like that? In any case,, it seems that many of the pathways previously shown in rodent models to be age-regulated were found here in humans, which is comforting - especially Foxo signaling, IL6, DNA repair, etc etc. For each of these, the particular genes found should be listed in a table.

10)The ACVR2A-TGFbeta axis was found to be upregulated in the oldest individuals. Out of curiosity, was GDF8, or GDF11 or ActivinA, or ActivinB in particular found? It's not so surprising that follistatin was found in these nuclei - follistatin and FLRG/FSTL3 are markers of ACVR2 pathway activation.

11) In this entire paper, there's no mention of senescence, or senescent nuclei. Were there no markers of senescence found? Does human muscle escape senescence, in contrast to rodent muscle?

12) Bottom of page 9: why are junb, jund, jun, fos and EGR1 listed as atrophy-related genes?

13) Of the aged individuals, are there associations that are specific to those with sarcopenia? Throughout the paper, in terms of the figures, it seems like all the studies done are on young vs aged, without discriminating amongst the aged for those with sarcopenia vs those aged individuals who don't have sarcopenia. Can this be done?

14) For the statistical analysis, there's no mention of FDR corrections. According to the paper, significance was set simply at $p < 0.05$. Of course for analyses like this, one has to further correct for multiple comparisons in R. If that wasn't done, all statistical analyses need to be redone.

15) Can the authors point to the particular aged individuals who were found to be sarcopenic, and show if there are any pathological changes specific to them?

Referee #2 (Remarks to the Author):

The manuscript from Lai and colleagues presents a combined single-cell/single-nucleus RNA and ATAC-sequencing atlas human (hindlimb) skeletal muscle samples across various ages to document cellular and molecular features on human muscle ageing. Though prior reports have generated scRNAseq data sets in human skeletal muscles (De Micheli, Skeletal Muscle 2020; Barruet, eLife

2020) those have not been explicitly analyzed by donor age nor been combined with snRNA/ATACseq to provide a higher resolution view of myonuclei though the preprint by Perez (bioRxiv 2021) provides some insights on the later. Therefore this work is a novel extension of the field and provides highly valuable reference data that should be frequently used by the field to study human muscle aging.

In summary, this work is important, performed in a technically proficient manner, and generates numerous novel insights. Considering my favorable view of the work as presented, I will focus my criticism on a few key aspects of the work that could be addressed to improve it for publication.

1) The authors could provide more clear and consistent presentation of human donor age and sex and muscle group in both their generalized analyses and in the presentation of individual samples. Given the wide number of reports documenting sex differences in muscle aging (see Hagg eLife 2021 for a review), examination of sex as a biological variable is be important and is missing in this work. For example, in all LTSR statistical analyses in which age is considered as a biological variable, sex should also be considered and its effect size reported (e.g, Fig 1d, Fig 4gj, etc). Given the variation in myofiber type distribution in different hindlimb muscles, considering muscle of origin especially in the myofiber assessments (Figs 2 and 3) is critical to allow the reader to understand the context of these data. Further, a UMAP plot (as in Fig 1bc and Extended Fig 2) showing the muscle group of origin should be provided.

2) The authors do not provide any orthogonal confirmation of the pathway enrichment (Fig 4), cell communication (Fig 5) and TF regulatory mechanics (Fig 6) and are expected to provide at least a modest confirmation of some of these insights with new experiments.

3) The primary value in this work is the breadth of these data and the robustness of their analyses. It would be most impact of these data could be made available in a data viewer to allow others to interrogate them more interactively and easily.

4) The authors should at least mention their sample collection, storage, and processing strategy in the beginning of the main text as it is critical and should not just be left in the Methods.

5) On line 89-92 of the text, the authors use the terms “noise” and “instability” incorrectly and should replace them. The observations that they are describing are simply cell population variation or heterogeneity not “noise” or "instability", which instead refer to dynamic measures and typically require time-series data on a consistent population of measured cells. These data are snapshots collected across donors of various ages and do not meet that criteria and thus other terms should be used.

6) In extended Fig 7, the p-values should be reported in the color bars not just Min to Max.

7) In extended Fig 10, the ordering of cell populations should always be Quiescent, early Primed, Late Primed, Differentiated which is the logical progression.

8) For all individual samples with histology or other data shown, identify the sample ID, sex, age, and

muscle group in some manner. These include Fig 2i, 3g, Ext fig 1b, Ext fig 3 and so forth.

9) In Fig 2h the basis of the differential expression testing is not clear. Is it TNNT2+ myonuclei vs all other myonuclei or TNNT2+ myonuclei in young-adult versus old? Also it is not clear which sc/sn data set this analysis is using.

10) It is unclear what the x axis in Fig 1a (right) is as it is not labelled.

Referee #3 (Remarks to the Author):

In this manuscript, the Muñoz-Cánoves and Esteban laboratories provide a comprehensive and impressive atlas of the ageing human skeletal muscle.

Profiling nuclei and cells (387,000 cells) obtained from muscle biopsies of 25 Caucasoid and Chinese individuals (13 males and 12 females), the authors have analyzed transcriptomes and chromatin accessibility at single-cell level resolution, confirming some of the findings with imaging approaches. The individuals were divided in two groups, young/adults (15-to 45-years old, n=8) and old/very old (74- to 99-years old, n=17). Both scRNA-seq and snRNA-seq approaches were employed to define the transcriptomes of myofiber nuclei, muscle stem cells (MuSCs), and non-myogenic muscle resident cells, and snATAC-seq was employed to probe chromatin accessibility. The integration of scRNA-seq/snRNA-seq and snATAC-seq combined with pseudotime ordering allowed the authors to obtain an unprecedented view of the aging human skeletal muscle.

SUMMARY

Myonuclei

Myonuclei composition was changed during aging, with decrease of type II (glycolytic) and relative increase of type I (oxidative) myonuclei. Previously undescribed myonuclei (ENOX1+ and TNNT2+ myonuclei) and three myonuclear populations enriched in the old group are reported. Old myonuclei were enriched in stress, inflammation, and atrophy-related transcripts, and dysregulated for genes of the circadian machinery. Chromatin accessibility was increased in old nuclei at stress and atrophy-related genes before their expression, indicating priming process and perhaps suggesting a predictive role of snATAC-seq on the aging process. Binding sites for transcriptional activators known to mediate the stress response (FOS, JUN, STAT3) were over-represented at regulatory regions of stress and atrophy-related genes. Repair signatures emerged in old myonuclei.

Muscle stem cells

Four cell subpopulations were identified: quiescent (qMuSCs), early primed (epMuSCs), late primed (lpMuSCs), and differentiating (dMuSCs). Old muscles displayed increased epMuSCs percentage indicating break of quiescence associated with decreased number of quiescent MuSCs, as observed in both old mice and elderly. Old epMuSCs revealed increased expression of DNA damage genes and, along with lpMuSCs, reduced DNA repair transcripts. In aged ep/lp MuSCs and dMuSCs, cell cycle-

related pathways were strongly downregulated, perhaps indicating that break of quiescence is uncoupled from proliferation in old MuSCs, a phenomenon that may well explain reduced number of MuSCs in the elderly. Old MuSCs were enriched for in binding motifs for transcription factors regulating myogenic differentiation (NFYA,B,C) and inflammation-related genes.

Non-myogenic muscle resident cells

In old muscles, the proportion of capillary endothelial cells (ECs) and pericytes decreased while that of arterial and vein ECs increased. Immune cells downregulated homeostatic immune and anti-inflammatory functions while increasing pro-inflammatory transcripts. A macrophage population (LAMM ϕ , Figure 4j) was prominently enriched in old muscles. Tenocytes increased in aged muscles, FAPs were increased and snATAC-seq suggested a phenotypic switch of FAPS from pro-regenerative to pro-fibrotic phenotypes.

Altered cell communications

Interactome analysis revealed modified signaling interactions between distinct cell types in aged muscles with special emphasis on cells involved in ESC, inflammation, and cytokine/growth factors signaling. More specifically, IGF, TGF β , BMP, Notch and WNT signalling was dysregulated in old muscles.

Cell-type assignment of causal variant at inherited risk loci for sarcopenia

snATAC-seq datasets were correlated to GWAS variants for muscle strength and other phenotypes related to muscular disease or metabolic function. Lean body was enriched, perhaps as expected, in the snATAC-se landscape of type II myonuclei. However, and surprisingly, muscle strength-related traits were enriched in old tenocytes and FAPs. This important finding suggests that non-myogenic muscle resident cells, and associated microenvironment, may be important genetic contributors to sarcopenia. Interestingly, sleep duration, creatinine and fasting glucose were related to myofibers, suggesting a link between metabolism and circadian rhythms and the myofiber.

COMMENTS

The unique combination of human samples from different populations and sexes, depth of molecular analysis conducted at single-cell level, identification of not previously detected cell subsets, and description of gene programs and chromatin accessibility landscapes in young and old muscles make this an important and authoritative study constituting a rich resource for the scientific communities interested in human muscle biology and human aging.

SUGGESTIONS

1. Figure 1d, Figure 4j,m. To better clarify that the indicated data relate to old samples, "Age" may be replaced by "Aged".

2. Extended Data Figure 2C. Can sn/scRNA-seq and snATAC-seq datasets be quantitatively correlated?
3. Extended Data Figures 7, Figure 9, 10, 11, 12, 13, 14,17. Heatmaps of functional enrichment (panels a-e, “Min-Max” Extended Figure 7), DORC heatmap (Extended Figure 9 panel b); functional enrichment (Extended Figure 10 panel g); markers (Extended Figure 11 panel c), and all other heatmaps (Extended Figures 12,13,14,17)should be quantified.
4. Figure 2a. Do the UMAPs represent both young and old snRNA-seq and snATAC-seq samples? If not, can the authors present them?
5. Figure 4e. Figure legends refer to left (old) and right (adult) panels. However, there are not left and right panels and Figure 4e illustrates TF binding site enrichment of up- (top) and down- (bottom)regulated genes.
6. Page 16, lines 451, 457, 461. Figure 6 should be Figure 5
7. scATAC-seq footprinting analysis of few selected regulatory regions in myonuclei, MuSCs, and non-myogenic muscle resident cells would establish whether the enriched DNA binding motifs are indeed occupied by their cognate transcription factors.
8. Extended Data Figure 3. The p-values of Pax7 and perilipin staining are both 0.02857. The authors may want to make sure that this is not an inadvertent duplication.

Author Rebuttals to Initial Comments:

Point-by-point response to the reviewers' comments

We thank all three reviewers for their efforts in evaluating our work and their overall positive assessment. We found that the points raised were very constructive and have now performed new experiments and analyses and/or modified the descriptions in the text to address them. All major changes to answer these points are highlighted in red in the revised text. In addition, we have made minor changes throughout the manuscript to enhance the readability (which are not highlighted); for instance, we divided the '*General and myofibre type-specific wasting mechanisms in ageing myonuclei*' into two sections to avoid an overly long section. We hope the reviewers will be satisfied with the revised manuscript and our detailed replies below.

Specific reviewers' comments:

REVIEWER 1:

"1) I've never seen so many authors listed as "equally contributed." It doesn't seem mathematically possible that all these people contributed the same amount. Also it would seem to hurt those people who really deserve "first author" standing to do this. Just a suggestion... it's the senior authors' call, but this seems quite unusual."

Our reply: We appreciate the reviewer's concern about the authorship hierarchy of this manuscript. While it might be certain that the completion and success of this study required efforts that cannot be exactly quantified as equal, it has been a *tour de force* possible only through an international collaboration involving different teams and expertise. Each co-first author provided extensive and essential intellectual and technical contributions to the overall development of the study, including, among other things, endless optimisation steps for tissue preparation and bioinformatic pipelines that are not shown in the manuscript. Without all this, this work would not have been possible. We hope that the reviewer will understand this, and we appreciate the honest opinion and suggestion.

"2) As an over-arching point, I think this paper needs to be more focused, with actually fewer figures - but those figures should have more depth, and supported with tables showing genes that are associated with the pathways noted. I really don't think it helps the authors to show 17 Extended Data Fig.s; given all the work that goes into each figure, it would be a shame if some additional important points are simply lost due to the mass of figures being dumped in this paper. This should be taken as friendly advice; I don't think the ever-expanding proliferation of data in a single paper really helps the scientists deliver a few clear messages and findings.

For example, at least for me, the biggest interest are the particular genes that are differentially regulated in the various nuclei as a function of age, which might point to age-regulated mechanisms that are impinging on muscle function. This sort of finding is really diluted and difficult to induce given the current data presentation.

As another example of this, consider the single sentence on page 6, that old myonuclei are enriched in atrophy related genes, with 4 such genes given as an example: Fox, Jun, Fbxo32, AMPD3. Among these, only Fbxo32 is purely an atrophy-associated genes, and the authors point to 3 separate extended figures to make this point: Extended data Fig 5a, 6d, and e. It seems the actual data might also be in Suppl Table 4, but that table is never discussed or mentioned in the entire paper.

Only extended data fig 5a is really getting to this - it's the heat map - and a heat map is really not sufficient, because it's very difficult to go through each column and understand which gene is upregulated; a table for this of all the atrophy-associated genes would be much more helpful.

Is this what Supplementary Table 4 is? As noted, there is zero discussion of that table in the figure (or paper), but it's the one that contains genes that purport to be part of the degenerative phenotype (at least by the title of the table). If this table has this data, it needs to be extensively discussed in the paper.

When I finally got to Extended Data Fig. 7 d, this seems like the heart of the paper, so I wonder why it's buried in extended figures, and isn't in the main paper. Isn't this the figure that really shows the age-perturbed signaling pathways in the various clusters?"

Our reply: We thank the reviewer for raising these relevant points, as they have helped us improve our manuscript's analysis and overall clarity. The reviewer's points can be summarised as shown below:

- (1) Reduce the number of Extended Data Figures to make the paper more focused and not bury relevant new data.
- (2) Present and discuss more clearly the differentially regulated genes in the various cell types with age, such as atrophy-associated genes.
- (3) Reorganise some of the key panels in the figures to highlight the age-perturbed signalling pathways revealed by pseudotime analysis in myonuclei; provide Supplementary Tables for the gene sets mentioned in those pathways; improve the description of Supplementary Table 4 (NEW Supplementary Table 6).

For (1): We have reduced the number of supplementary figures to 15 and simplified/reorganised others. This has helped us to highlight the most novel findings, which we believe has improved the overall clarity while preserving the most useful supplementary information. The current number of supplementary figures is still substantial, but we believe that in the context of this newly-revised version of the manuscript, the information flows better and is easier to understand. We will be happy to perform additional modifications if the reviewer and/or editor considers it necessary.

For (2): We have now provided supplementary tables (NEW Supplementary Tables 10-11) containing all the enriched differentially expressed genes (DEGs) that appear with ageing and the associated pathways in the various cell types of our manuscript. Specifically, as suggested by the reviewer, we provide a carefully defined atrophy-

associated gene list¹ in **NEW Supplementary Table 3** and have quantified the overall changes in those genes (computed as a score) along the myofibre degeneration trajectory (**NEW Extended Data Fig. 6d**). We include the descriptions for this in the revised manuscript (**see page 8**). In addition, we performed a Pearson's correlation analysis to highlight those genes highly correlated with ageing in myonuclei and MuSCs as the major contributors to muscle wasting. This information is now included in **NEW Supplementary Table 5**, **NEW Extended Data Fig. 6c and 8j** and **pages 8 and 12** of the revised manuscript.

For (3): We agree with the reviewer's comments and have relocated the previous Extended Data Fig. 7d into **NEW Fig. 3c**. Indeed, this figure explains which genes/pathways are associated with the degenerative trajectories in each myofibre type with ageing, and it is thus very relevant. The genes in the shown clusters and pathways are now included in **NEW Supplementary Table 6**. We have also modified the text accordingly (**see pages 9 and 10** in the revised manuscript).

*“As one more over-arching point, which is listed again below - **there's not a single mention of senescence in this paper. Was evidence of senescence not found in aged human muscle? Certainly many of the pathways perturbed and genes found are consistent with the presence of senescent cells, according to prior papers.**”*

Our reply: The reviewer raised another good point. Although senescence has been postulated as a hallmark of ageing² *bona fide* senescent cells are hardly detected in unperturbed human muscles. This was reported in our previous study³ using senescence-associated beta-galactosidase (SA- β -gal) and CDKN2A staining (*see below REBUTTAL Figure 1, upper panel*) and other studies⁴ using γ H2AX staining (*see below REBUTTAL Figure 1, bottom panel*). Consistent with these observations, we performed SA- β -gal staining of sections from some of the same donor muscle samples used in this study but failed to detect the presence of senescent cells (*see below REBUTTAL Figure 2, also added in NEW Extended Data Fig. 1c*). Low numbers of senescent cells might still be present; however, given the limited sequencing depth inherent to droplet-based single-cell sequencing technologies and without relying upon other more robust parameters, we could not differentiate any small population of senescent cells in our omics dataset. Therefore, we conclude that the contribution of senescent cells to our findings is low. We cannot exclude, however, that some of the molecular changes associated with ageing described in our study predispose to the onset of senescence upon damage (*e.g.*, trauma). We have modified the text to accommodate these points in the revised manuscript (**see page 4**).

REBUTTAL Figure 1.

SA-β-gal staining coupled to CDKN2A⁺ immunohistochemistry of uninjured or damaged human muscle samples (upper panel, taken from Moiseeva *et al.* ³), and γ H2AX immunofluorescence of human muscle sections (bottom panel, taken from Dungan *et al.* ⁴).

REBUTTAL Figure 2.

Representative SA-β-gal staining of uninjured human muscle samples from this manuscript ($n = 4$ per group), and injured mouse muscles (7 days post-cardiotoxin injury, used as control). SA-β-gal positive cells were only detected in injured muscles. Scale bar, 100 μ m. This is included in **NEW Extended Data Fig. 1c** in our revised manuscript.

“3) As the authors note, there have been prior single-cell analyses of skeletal muscle, including in aging, and prior single-cell analysis of skeletal muscle in the non-aged human, so it would seem the “novelty” here would be the comparison of many humans throughout a wide aging range, and the possibility of correlating findings to sarcopenia. The authors agree that this is the point of the study (pg 5 of the manuscript).”

Our reply: Yes, there have been other single-cell papers on skeletal muscle (which we reference in our manuscript). Yet our study presents the most comprehensive single-cell/single-nucleus transcriptomic and epigenomic human skeletal muscle ageing atlas to date (as acknowledged by all three reviewers). Importantly, we also redress the lack of studies at this level of molecular detail on human muscle biology since the unique features characteristic of humans are often underrepresented in other previous studies that mostly rely on small mammal models such as rodents, which display a different muscle myofibre composition and physiology. Further, as the reviewer mentions, our study sheds light on the molecular and cellular communication aspects of sarcopenia and frailty in humans.

“4) Given the desire to find differentially-expressed genes associate with sarcopenia, it's not obvious that the methods employed here are the best approaches.

Old-fashioned bulk RNA seq would more comprehensively show sarcopenia-associated genes, since it would allow for a much deeper analysis, whereas single cell RNA seq limits one to the top 5000 or so highest-expressed genes. This lets you determine cell types, and perhaps most obvious mechanisms, but does not allow for a full characterization of genes which are specifically expressed in sarcopenia.”

Our reply: The reviewer raised another good point. Genome-wide transcriptional profiling with bulk RNA-seq allows a comprehensive measurement of the molecular state of the tested samples and has been applied to ageing and sarcopenia studies⁵⁻⁷. An important consideration is, as the reviewer notices, that bulk RNA-seq relies either on:

i) Chunks of tissues comprising heterogeneous mixtures of cells. Here, bulk RNA-seq provides the average transcriptomic profiling of the mixture, which enables genome-wide detection of regulated genes but not of the cell type-specific responses, such as the differential responses of type I and type II myofibres or other cell types upon ageing. However, this approach is not ideal for detecting responses in small cell subpopulations, such as myonuclei subtypes or diverse forms of satellite cells. In addition, sequencing chunks of tissues is often biased by suboptimal sampling, which results in different cell compositions between samples; or

ii) Sorted cell populations with known surface markers. However, cell sorting is biased to specific markers and requires laborious tissue dissociation and flow cytometry-based sorting⁸, which can lead to a perturbed cell state. For instance, stress caused by cell dissociation is known to be a major caveat for the characterisation of satellite cells^{9,10}. We have explained these points more clearly in our revised manuscript (see page 11).

To further corroborate the above-mentioned points, we conducted bulk RNA-seq analysis using muscle tissue samples obtained from a representative subset of the original biopsies used in our study. Additionally, we have re-analyzed several publicly available bulk RNA-seq datasets from previous publications, which involved a larger number of individuals⁵⁻⁷. A summary table depicting the donors from these studies is presented below as **REBUTTAL Table 1**. Comparing the results of PCA analysis between bulk RNA-seq and sc/snRNA-seq, we observed significant transcriptional variability among individuals within the same age group in the bulk RNA-seq studies. This high variability led to a limited number of DEGs (**REBUTTAL Figure 3a-b**). In contrast, sc/snRNA-seq enables the identification of age-related changes specific to cell types, thereby reducing the transcriptional variability observed in bulk RNA-seq studies.

REBUTTAL Figure 3.

a. PCA of bulk RNA-seq experiments for young and old skeletal muscle from this study and the indicated published studies and snRNA-seq. b. Number of DEGs obtained from each dataset.

GEO ID	Study	Adult		Old		
		Age (years)	Healthy	Age (years)	Healthy	Sarcopenic
-	This study (Europe)	18-45	n = 4	77-95	n = 4	-
-	This study (Asia)	15-35	n = 5	80-82	n = 4	-
GSE164471	Tumasian, R. A., 3rd et al [PMID:3795677]	22-52	n = 53	53-83	n = 25	-
GSE111006	Hertfordshire Sarcopenia Study [PMID: 31862890]	-	-	68-77	n = 36	n = 4
GSE111010	Jamaica Sarcopenia Study [PMID: 31862890]	-	-	63-89	n = 30	n = 9
GSE111016	Singapore Sarcopenia Study [PMID: 31862890]	-	-	65-79	n = 20	n = 20
GSE167186	Perez, K. et al [PMID: 36516485]	18-30	n = 19	65-85	n = 29	n = 24

REBUTTAL Table 1. Description of the samples from the newly generated and publicly available datasets of bulk RNA-seq from human skeletal muscle.

“5) Was there really no effect of gender on the findings? (Extended Data Fig. 2a-c) This seems surprising given the females would be undergoing menopause, and sex hormones do effect skeletal muscle. There do seem to be clusters that are relatively dominated by male, just from looking at the figures - or is that a function of the number of males vs females? (for example, look at Extended Data, Figure 2b, male vs female, the top cluster.)”

Our reply: This is also an excellent comment. Sexual dimorphism in biological ageing is common in various tissues and organs¹¹. A similar comment was also raised by reviewer 2 (see point 1).

To address this important question, we re-analysed our dataset, taking into consideration the sex of each sample, as follows:

(1) We performed correlation analyses for cell type identity across data modalities and sexes, observing a high consistency between male and female nuclei/cells (see **NEW Extended Data Fig. 2b**).

(2) To further clarify the effect of sex on nucleus/cell composition, we have now included new panels split by sex (see **NEW Fig. 1a and 2b, d; and Extended Fig. 1a, 2b-d, 4c, 4g, 8g, 9e, 10e and 11e**) and a supplementary table (**NEW Supplementary Table 9**). We observed a high variability among the donors, as expected, but the cell composition changes upon ageing presented in this work are comparable between male and female donors. We have added a note explaining this in the revised manuscript (see **pages 4-6**).

(3) We also performed PCA analysis of pseudobulk scRNA-seq data for type I and type II myofibres from all individuals. We found that the sex difference is also not apparent using this approach (see *below* **REBUTTAL Figure 4**). This analysis is not included in the revised manuscript.

REBUTTAL Figure 4.
PCA for pseudobulk gene expression values of type I/type II myonuclei in snRNA-seq.

Thus, we conclude that, in our study, cellular and molecular changes upon ageing are generally consistent between the sexes. Further exploring the sex-specific differences would require increasing the number of tested individuals, which we will do in future expansions of our atlas. We have added a note for this point in the discussion of our revised manuscript (see page 20).

“6) The ENOX1 myonuclei specific for type II myofibres, enriched in the young group, is quite interesting, as is the TNNT2+ myonuclei, and the three additional populations enriched in the old group. For these nuclei, were there any differences besides relative abundance as a function of age? In other words, were there co-located transcripts that one could say were significantly differentially expressed in these nuclei, with the understanding that this is a question better left for bulk RNA-seq? - or is this what is shown in Figure 2h? There's just one sentence devoted to 2h, but if this is really a discussion of differential gene expression in young vs old, this should be explained and expanded on quite a bit, as to what genes in particular are relatively perturbed vs TNNT2 in these nuclei. As noted above, a table of the differentially regulated genes as a function of age in these nuclei would help the authors make their point more easily, and these genes could also be annotated as to the pathways they fall in.”

Our reply: The reviewer raised important points that have helped improve the clarity of our manuscript. These include:

- (1) Identify the co-located transcripts differentially expressed in the specialised myonuclei, elaborate the descriptions and provide a table for the enriched pathways associated to each myonuclear population.
- (2) Comprehensively address the age-related molecular changes in each myonuclear population.

For (1), to address this, we have provided a comprehensive functional enrichment analysis for the DEGs in each specialised myonuclei subpopulation (see **NEW Extended Data Fig. 5b**). To further determine the co-located transcripts within a myonuclear population, we have also performed Hotspot analysis¹², an algorithm that infers

informative co-expression gene modules in single-cell transcriptomic data (see NEW Fig. 2e-g, and Extended Data Fig. 5d-f). Taken together the hotspot analysis (see NEW Supplementary Table 4) and the functional enrichment analysis from the obtained DEGs of each population (see NEW Supplementary Tables 10-11), we have now elaborated the description for specialised myonuclei in the revised manuscript (see pages 6 and 7).

For (2), we first determined the common DEGs between adult (≤ 46 years), old (74-82 years), and very old (≥ 84 years) type I and type II myonuclei populations and performed a Pearson's correlation analysis to highlight genes highly correlated with ageing in myonuclei. These identified changes in old/very old myonuclei reflect the age-associated increase in the tissue inflammatory status and the dysregulation of general functions that translates into myofibre atrophy (see NEW Extended Data Fig. 6a-c). We have now improved the description for this part in the revised manuscript (see page 8) and provided the supplementary tables containing all the DEGs and the associated pathways and the age-related gene lists in myofibres (see NEW Supplementary Table 5). Second, to study the molecular dynamics of myofibre progress toward degeneration, we constructed the pseudotime trajectory from the healthy myonuclei to the degenerative ones, which not only reflected the progressive or abrupt course of degeneration in type I or type II myofibres respectively but also revealed transcriptional variation during myofibre degeneration. This information is included in the NEW Fig. 3a-d. We have now improved the description for this part in the revised manuscript (see pages 9-10) and provided the supplementary table containing all the gene clusters and the associated enriched pathways (see NEW Supplementary Table 6).

In addition, the reviewer may also point to the intrinsic age-related changes in specialised myonuclear populations. However, these populations are lowly detected in muscles that limits the analysis. Therefore, we have performed DEG and functional enrichment analysis comparing adult and old myonuclei in the most abundant populations (*please see below*, REBUTTAL Figure 5). According to the previous findings, we observed the enrichment of common ageing mechanisms, such as the upregulation of inflammation (TNF α signaling via NF κ B) or the downregulation of sarcomeric genes (muscle contraction). However, this analysis also showed specific functions enriched in each myonuclear population such as greater downregulation of muscle contraction and glycolytic process in the *ENOX1*⁺ population. Moreover, we observed the greater enrichment in upregulated genes of protein catabolic process in *DCLK1*⁺(I) population or the enrichment of TGF β receptor signaling in *ID1*⁺ populations. This analysis points to different age-related responses in the specialized myonuclei populations reinforcing the molecular identity of these populations (see pages 6 and 7). Due to the space limitation, this analysis is not included in the revised manuscript.

Taken together, we believe we have addressed the age-related molecular changes in myonuclei in a comprehensive manner.

REBUTTAL Figure 5.

Heatmap of the functional enrichment analysis for the upregulated (left panel) and downregulated (right panel) genes in each myonuclei subpopulation. Pathway enrichment significance is represented by the colour scale. For each depicted pathway, representative genes are labelled.

“7) The authors label TNNT2 expression as pathological, but it is found in the young - it's just found more in the old, consistent with the nuclei as a class being more abundant. Doesn't the finding in the young suggest these nuclei are not necessarily pathological, or do the authors think that loss of innervation is even happening in the young, just at a lesser extent?”

Our reply: The reviewer raises a relevant point that has helped us clarify our findings. The expression of *TNNT2* has been reported during *de novo* myogenesis in mice but was undetected in unperturbed skeletal muscle^{13–16}. Those reports are consistent with the low abundance of *TNNT2*⁺ myonuclei in adult skeletal muscle in our study (median: 0.08% among all myonuclei), likely representing a rare transient state in normal skeletal muscle for myonuclei renewal during the daily mild myofibre lesions (see **NEW Fig. 2d**), as the reviewer suggests. However, in the old, we found that the proportion of *TNNT2*⁺ myonuclei substantially increased (to 2.45%, a ~30-fold increase compared to the adult group), suggesting an association with impaired muscle function and the development of sarcopenia. In this regard, *TNNT2* expression has also been associated with muscle denervation and several neuromuscular diseases^{17,18}; and ageing^{19,20} in mice. Importantly, knockdown of *TNNT2* in old mice has been associated with decreased denervation markers and improved muscle function¹⁷ whereas overexpression of *TNNT2* leads to accelerated muscle degeneration and motor activity decline¹⁹. We have now explained these findings more clearly in our revised manuscript (see **pages 6 and 7**).

“8) The authors in page 7 refer to the emergence of new nuclei, but is there an example of an entirely new type of nuclei that appears in the older muscle? The prior discussions all point to relative enrichments - or did I miss a figure that shows something that purely found in older subjects? Later on that same page they refer to "the emergence of specific end-stage myonuclear populations, expressing degenerative traits", but they don't reference a figure showing this. What figures shows that? Or do the authors mean that

myonuclear populations also found in the young develop degenerative traits with age? (that's not the same as claiming there are novel degenerative nuclei forming)."

Our reply: This is also a very good point related to our reply to the item above by this reviewer. Based on the snRNA-seq results, *ENOX1*⁺ myonuclei were found across different samples despite its loss with ageing (median: adult, 9.13%, old, 4.21%). However, the rest of the populations appeared mainly in the old donors with a 8.4- to 30.6-fold difference as compared to the adult (e.g., *TNNT2*⁺ (I): adult, 0.08%, old, 2.45%; *ID1*⁺ (I), 0.17%, old, 1.47%; *DCLK1*⁺ (I), adult, 0.27%, old, 2.27%; *SAA2*⁺ (II), adult, 0.10%, old, 0.98%). Indeed, we cannot exclude the possibility that these rare populations increase in adult individuals upon certain circumstances, such as muscle trauma, syndromic or pathological conditions. However, given the tremendous proportional difference, we conclude that apart from *ENOX1*⁺, these rare myonuclear populations are highly associated with ageing. In this regard, we also refer to them as end-stage myonuclei subpopulations. These observations fit well with a model in which normal adult myonuclei populations progressively develop degenerative traits with ageing rather than a *de novo* emergence of age-related myonuclear populations. This was shown in the **previous and NEW Fig. 2d**. We have now provided the quantification of each myonuclei subpopulation in the **NEW Supplementary Table 9** and explained this more clearly in the revised manuscript (see **page 6**).

"9) As noted above, *Extended Data Fig. 7 d* seems to be the most important and helpful figure, because it shows the pathways that are functionally enriched due to age. I'm confused by the arrows going in opposite directions in Type I vs Type II fibers; why is the data presented like that? In any case, it seems that many of the pathways previously shown in rodent models to be age-regulated were found here in humans, which is comforting - especially Foxo signaling, IL6, DNA repair, etc etc. *For each of these, the particular genes found should be listed in a table.*"

Our reply: Following the reviewer's suggestions, we have moved *Extended Data Fig. 7d* to **NEW Fig. 3d** to highlight the aged-associated pathways. The arrows indicate the pseudotime directionality, from the healthy state towards the degenerative state, in type I or type II myofibres as measured by trajectory analysis. The global and myofibre-specific degenerative gene expression gradients along the trajectories were shown in the heatmap for side-by-side comparison. We have revised the panel to present it better by adding the pseudotime colour bar (see **NEW Fig. 3d**) and provided the corresponding table with the associated genes for the enriched pathways in each gene cluster (**NEW Supplementary Table 6**).

"10)The *ACVR2A-TGFbeta* axis was found to be upregulated in the oldest individuals. Out of curiosity, was *GDF8*, or *GDF11* or *ActivinA*, or *ActivinB* in particular found? It's not so surprising that *follistatin* was found in these nuclei - *follistatin* and *FLRG/FSTL3* are markers of *ACVR2* pathway activation."

Our reply: Thanks for these suggestions. Indeed, *GDF11*, activin A and activin B ligands (*INHBA* and *INHBB*) and their receptors (*ACVR1*, *ACVR1B* and *ACVR2A*) are increased

in the old donors. We have included these genes in **NEW Fig. 5g**. *GDF8 (MSTN)* displayed low expression levels in all groups. We also performed immunofluorescence staining to validate the increased expression of *ACVR2A* (**NEW Fig. 5h**). We have modified the manuscript to describe these findings (see **page 16**).

“11) *In this entire paper, there's no mention of senescence, or senescent nuclei. Were there no markers of senescence found? Does human muscle escape senescence, in contrast to rodent muscle?”*

Our reply: Please see this reviewer's response to the above point 2 (**pages 3-4** of the rebuttal letter).

“12) *Bottom of page 9: why are junb, jund, jun, fos and EGR1 listed as atrophy-related genes?”*

Our reply: Thanks for noticing this. We agree with the reviewer that these genes (*JUNB*, *JUND*, *JUN*, *FOS*, and *EGR1*) are not canonical atrophy genes, although several studies have shown their involvement in atrophic conditions like denervation^{21–23}. We have now eliminated them from the list of atrophy-related genes and revised the list according to a previous report¹ (see **NEW Supplementary Table 3**).

“13) *Of the aged individuals, are there associations that are specific to those with sarcopenia? Throughout the paper, in terms of the figures, it seems like all the studies done are on young vs aged, without discriminating amongst the aged for those with sarcopenia vs those aged individuals who don't have sarcopenia. Can this be done?”*

Our reply: The reviewer raised a very relevant point. Barthel^{24,25} and Charlson²⁶ indexes have been used in clinical practice to assess patients' functional status and help diagnose sarcopenia. Our study shows a good correlation between patients with more abnormal scores in those indexes and advanced age, as indicated more clearly in our revised manuscript (see **page 4**). Our bioinformatic analyses also confirmed the association of an individual's age with the progression of locomotor dysfunction in our cohort. Specifically, we defined the association of multiple molecular processes with ageing, and some of these changes are more aggravated in very old patients (>88 years old). For example, the increased fibrosis, *ACVR2A* expression, loss of MuSCs, and loss of Type II myofibres. However, no single transcriptomic or epigenomic change differentiates very old patients with sarcopenic signs from others, reinforcing the idea that sarcopenia is a progressive event driven simultaneously by multiple factors. We have now explained this more clearly in the revised manuscript (see **page 20**).

“14) *For the statistical analysis, there's no mention of FDR corrections. According to the paper, significance was set simply at $p < 0.05$. Of course for analyses like this, one has to further correct for multiple comparisons in R. If that wasn't done, all statistical analyses need to be redone.*”

Our reply: We agree with the reviewer and have now redone all the functional enrichment analyses by applying a well-accepted FDR correction test, the Benjamini-Hochberg procedure²⁷ (see NEW Fig. 3d, 4c, 4g, 4j and 4m; and Extended Data Fig. 5b, 6a, 8d, 8i and 11f-g). The main conclusions of our study remain unaffected.

“15) Can the authors point to the particular aged individuals who were found to be sarcopenic, and show if there are any pathological changes specific to them?”

Our reply: This is an excellent suggestion related to the above item 13 by this reviewer. Individual ‘P28’ displayed the lowest Barthel Index (55) and higher Charlson Index (600). Consistently, this patient presented the highest reduction in type II myofibres, and the MuSCs displayed the highest expression of ACVR2A in our snRNA-seq dataset, pointing to increased atrophy, a key hallmark of sarcopenia progression. See the details for all patients in REBUTTAL Table 2. This information is now included in Supplementary Table 12.

Donor	Age	CI	BI	Type I (%)	Type II (%)	I/II ratio	PAX7 ⁺ FOS ⁺ cells (%)	PAX7 ⁺ cells /100 myofibres	ACVR2A ⁺ area (%)
P26	18	0	100	53.02145	46.97855	1.128631	56.69291339	21.5	0.012512059
P13	38	0	100	39.13009	60.86991	0.642848	55.88235294	15.4	0.011526242
P5	45	0	100	47.56084	52.43916	0.906972	58.82352941	20.9	0.02350688
P9	46	0	100	37.18168	62.81832	0.591892	56.37583893	21.7	0.01153678
P29	77	400	100	72.71293	27.28707	2.664739	69.79166667	11.4	0.014743697
P23	85	500	100	46.09106	53.90894	0.85498	92.30769231	11.3	0.056057943
P28	88	600	55	70.35861	29.64139	2.373661	96.2962963	4.07	0.226086509
P16	95	NA	NA	65.05028	34.94972	1.861253	73.07692308	8.7	0.043990887

REBUTTAL Table 2. Description of the functional and histological parameters associated with muscle function measured in the cohort used in this study.

REVIEWER 2:

“The manuscript from Lai and colleagues presents a combined single-cell/single-nucleus RNA and ATAC-sequencing atlas human (hindlimb) skeletal muscle samples across various ages to document cellular and molecular features on human muscle ageing. Though prior reports have generated scRNAseq data sets in human skeletal muscles (De Micheli, Skeletal Muscle 2020; Barruet, eLife 2020) those have not been explicitly analyzed by donor age nor been combined with snRNA/ATACseq to provide a higher resolution view of myonuclei though the preprint by Perez (bioRxiv 2021) provides some insights on the later. Therefore this work is a novel extension of the field and provides highly valuable reference data that should be frequently used by the field to study human muscle aging.”

In summary, this work is important, performed in a technically proficient manner, and generates numerous novel insights. Considering my favorable view of the work as presented, I will focus my criticism on a few key aspects of the work that could be addressed to improve it for publication.”

Our reply: We thank the reviewer for the positive comments regarding our manuscript and the careful assessment.

“1) The authors could provide more clear and consistent presentation of human donor age and sex and muscle group in both their generalized analyses and in the presentation of individual samples. Given the wide number of reports documenting sex differences in muscle aging (see Hagg eLife 2021 for a review), examination of sex as a biological variable is be important and is missing in this work. For example, in all LTSR statistical analyses in which age is considered as a biological variable, sex should also be considered and its effect size reported (e.g, Fig 1d, Fig 4gj, etc).”

Given the variation in myofiber type distribution in different hindlimb muscles, considering muscle of origin especially in the myofiber assessments (Figs 2 and 3) is critical to allow the reader to understand the context of these data.

Further, a UMAP plot (as in Fig 1bc and Extended Fig 2) showing the muscle group of origin should be provided.”

Our reply: The reviewer raises excellent points. We agree that the donor characteristics (age, sex, and muscle group) should be more clearly and consistently presented throughout the manuscript. We have now provided new figures for the generalized analyses (see **NEW Extended Fig. 2c, and 4c-d**) and provide donor ID in figure legends for individual samples of the histology and immunofluorescence staining (see figure legends for **NEW Fig. 2h and 5h; and NEW Extended Fig. 1b, 3a-f, 4f, 5g, 6f, 6g, 8h, 13a and 13c**), in which the donor information can be further explored in **NEW Supplementary Table 1**.

Regarding the consideration of sex differences, a similar concern was also indicated by reviewer 1 (please also see above point 5 for reviewer 1). We concur that this is an important aspect of human muscle ageing, and of any biological study in general. As stated in our detailed response to reviewer 1, we observed the existence of individual variability, as expected, but, overall, there were no substantial differences between male and female donors. In fact, biological factors (sex and ethnicity) and technical factors (omics modalities, sequencing batch) were considered as variables in all LTSR analyses performed along the manuscript. We have explained this more clearly in the revised manuscript (see **page 5**).

Regarding the consideration of muscle origin: indeed, due to the logistics of biopsy collections, samples were taken from different locomotor muscles, as they corresponded to different types of orthopedic surgery. In the adult group, the muscle origin included *semitendinosus*, *gracilis*, *tibial anterior* and *gluteus medius*, mostly from patients

undergoing surgery after an acute fracture or fracture repair. In the old group, the muscle samples mostly were from *gluteus medius* and *vastus lateralis*, which came from patients undergoing hip surgery. This information was included in the Methods part and Table S1 in the previous submission and in the revised manuscript (see **NEW Supplementary Table 1**). To further clarify the myofiber type distribution across different muscle origins, we have calculated the proportions in muscles from different muscle origins in the snRNA-seq and snATAC-seq dataset. We observed that ageing induced a general decrease in type II myonuclei and a relative increase in type I regardless of muscle origin (see *below* **REBUTTAL Figure 6a**). The same applied to the appearance of end-stage myonuclei subpopulations (see *below* **REBUTTAL Figure 6b**).

■ *Gluteus medius* ■ *Gracilis* ■ *Semitendinosus* ■ *Tibial anterior* ■ *Vastus lateralis*

REBUTTAL Figure 6.

Quantification of the myonuclei population proportions in adult and old individuals coloured by muscle origin in snRNA-seq (upper panel) and snATAC-seq (bottom panel) for (a) myofibre type; (b) myonuclei subpopulations.

Moreover, immunohistochemical myofibre typing of the same muscle type (*gluteus medius*) from adult 'P13' and old 'P29' individuals showed consistent results (see **Extended Data Fig. 4f**). Thus, the changes upon ageing in myonuclei are generally comparable across locomotor muscles. As requested by the reviewer, we have now included UMAP plots for muscle origin (see **NEW Extended Data Fig. 2c and 4c**), which shows no myonuclei population/subpopulation biases. We have added a note explaining this in the revised manuscript (see **pages 5 and 6**).

In the future, it will be important to explore further the sex-specific and muscle origin-specific differences by expanding the number of individuals. We have added a note for this in the discussion of the revised manuscript (see **page 20**).

"2) The authors do not provide any orthogonal confirmation of the pathway enrichment (Fig 4), cell communication (Fig 5) and TF regulatory mechanics (Fig 6) and are expected to provide at least a modest confirmation of some of these insights with new experiments."

Our reply: We thank the reviewer for this suggestion. We have now added new validations to reinforce our findings, as follows:

- (1) Validation of the progressive myofibre type switching upon ageing (**NEW Extended Data Fig. 4f**);
- (2) Increase in the number of biological replicates for TNNT2 immunofluorescence staining (**NEW Fig. 2h**);
- (3) Increase in the number of biological replicates for NCAM1 immunofluorescence staining (**NEW Extended Data Fig. 5g**);
- (4) Confirmation of the increase of early primed MuSCs upon ageing by FOS/PAX7 immunofluorescence staining (**NEW Extended Data Fig. 8h**);
- (5) Confirmation of the increased *ACVR2A* expression upon ageing by immunofluorescence staining (**NEW Fig. 5h**);
- (6) Confirmation of the reduction of integrin component *ITGA7* upon ageing by immunofluorescence staining (**NEW Extended Data Fig. 13c**).

These experiments are now mentioned on **pages 6, 7, 12 and 16** of the revised manuscript.

"3) The primary value in this work is the breadth of these data and the robustness of their analyses. It would be most impact of these data could be made available in a data viewer to allow others to interrogate them more interactively and easily."

Our reply: We thank the reviewer for this relevant suggestion. We have now constructed the Human Muscle ageing cell Atlas (HMA) database. We believe HMA will become a fundamental resource for future human muscle biology and ageing studies. We have

added a note for this in the revised manuscript (see page 3). It can be accessed at <https://db.cngb.org/cdcp/hlma/>.

“4) The authors should at least mention their sample collection, storage, and processing strategy in the beginning of the main text as it is critical and should not just be left in the Methods.”

Our reply: We have now included this in the revised manuscript (see page 4).

“5) On line 89-92 of the text, the authors use the terms “noise” and “instability” incorrectly and should replace them. The observations that they are describing are simply cell population variation or heterogeneity not “noise” or “instability”, which instead refer to dynamic measures and typically require time-series data on a consistent population of measured cells. These data are snapshots collected across donors of various ages and do not meet that criteria and thus other terms should be used.”

Our reply: The reviewer is right. We have changed this term to ‘transcriptional heterogeneity’ or ‘epigenetic heterogeneity’ in the revised manuscript (see NEW Fig. 1e and page 5).

“6) In extended Fig 7, the p-values should be reported in the color bars not just Min to Max.”

Our reply: Thanks for this helpful suggestion. We have now changed this figure and all other similar panels accordingly (see NEW Fig. 3d, 4c, 4g, 4j, and 4m; Extended Data Fig. 5b, 6a, 8d, 8i, 11f and 11g).

“7) In extended Fig 10, the ordering of cell populations should always be Quiescent, early Primed, Late Primed, Differentiated which is the logical progression.”

Our reply: We have now revised this as suggested (see NEW Fig. 4b-c and Extended Data Fig. 8c-g).

“8) For all individual samples with histology or other data shown, identify the sample ID, sex, age, and muscle group in some manner. These include Fig 2i, 3g, Ext fig 1b, Ext fig 3 and so forth.”

Our reply: Following the reviewer’s suggestion, we have included the sample ID in the corresponding figure legends (see figure legends for NEW Fig. 2h and 5h; and NEW Extended Fig. 1b, 3a-f, 4f, 5g, 6f, 6g, 8h, 13a and 13c). All related information can also be found in Supplementary Table 1.

“9) In Fig 2h the basis of the differential expression testing is not clear. Is it TNNT2+ myonuclei vs all other myonuclei or TNNT2+ myonuclei in young-adult versus old? Also it is not clear which sc/sn data set this analysis is using.”

Our reply: Thank you for bringing this to our attention. We realised that the previous panel used was not presented clearly enough. To address this issue, we have made improvements. We have substituted this panel for Hotspot analysis to infer informative co-expression gene modules in myofibre subtypes (see **NEW Fig. 2e-f**), and provided a new panel for DEG analysis for each myonuclei subpopulations (see **NEW Extended Data Fig. 5b**) and improved the descriptions in the revised manuscript (see **pages 6 and 7**). As for the second question, snRNA-seq was used for all the myonuclei analyses. We have labelled this panel better in our revised manuscript (see **NEW Extended Data Fig. 5b**).

“10) It is unclear what the x axis in Fig 1a (right) is as it is not labelled.”

Our reply: Thanks for spotting this. We have now amended it (see **NEW Fig. 1a**).

REVIEWER 3:

“In this manuscript, the Muñoz-Cánoves and Esteban laboratories provide a comprehensive and impressive atlas of the ageing human skeletal muscle.”

Profiling nuclei and cells (387,000 cells) obtained from muscle biopsies of 25 Caucasoid and Chinese individuals (13 males and 12 females), the authors have analyzed transcriptomes and chromatin accessibility at single-cell level resolution, confirming some of the findings with imaging approaches. The individuals were divided in two groups, young/adults (15-to 45-years old, n=8) and old/very old (74- to 99-years old, n=17). Both scRNA-seq and snRNA-seq approaches were employed to define the transcriptomes of myofiber nuclei, muscle stem cells (MuSCs), and non-myogenic muscle resident cells, and snATAC-seq was employed to probe chromatin accessibility. The integration of scRNA-seq/snRNA-seq and snATAC-seq combined with pseudotime ordering allowed the authors to obtain an unprecedented view of the aging human skeletal muscle.

SUMMARY

Myonuclei

Myonuclei composition was changed during aging, with decrease of type II (glycolytic) and relative increase of type I (oxidative) myonuclei. Previously undescribed myonuclei (ENOX1+ and TNNT2+ myonuclei) and three myonuclear populations enriched in the old group are reported. Old myonuclei were enriched in stress, inflammation, and atrophy-related transcripts, and dysregulated for genes of the circadian machinery. Chromatin accessibility was increased in old nuclei at stress and atrophy-related genes before their expression, indicating priming process and perhaps suggesting a predictive role of snATAC-seq on the aging process. Binding sites for transcriptional activators known to mediate the stress response (FOS, JUN, STAT3) were over-represented at regulatory regions of stress and atrophy-related genes. Repair signatures emerged in old myonuclei.

Muscle stem cells

Four cell subpopulations were identified: quiescent (qMuSCs), early primed (epMuSCs), late primed (lpMuSCs), and differentiating (dMuSCs). Old muscles displayed increased epMuSCs percentage indicating break of quiescence associated with decreased number of quiescent MuSCs, as observed in both old mice and elderly. Old epMuSCs revealed increased expression of DNA damage genes and, along with lpMuSCs, reduced DNA repair transcripts. In aged ep/lp MuSCs and dMuSCs, cell cycle-related pathways were strongly downregulated, perhaps indicating that break of quiescence is uncoupled from proliferation in old MuSCs, a phenomenon that may well explain reduced number of MuSCs in the elderly. Old MuSCs were enriched for in binding motifs for transcription factors regulating myogenic differentiation (NFYA,B,C) and inflammation-related genes.

Non-myogenic muscle resident cells

In old muscles, the proportion of capillary endothelial cells (ECs) and pericytes decreased while that of arterial and vein ECs increased. Immune cells downregulated homeostatic immune and anti-inflammatory functions while increasing pro-inflammatory transcripts. A macrophage population (LAMM ϕ , Figure 4j) was prominently enriched in old muscles. Tenocytes increased in aged muscles, FAPs were increased and snATAC-seq suggested a phenotypic switch of FAPS from pro-regenerative to pro-fibrotic phenotypes.

Altered cell communications

Interactome analysis revealed modified signaling interactions between distinct cell types in aged muscles with special emphasis on cells involved in ESC, inflammation, and cytokine/growth factors signaling. More specifically, IGF, TGF β , BMP, Notch and WNT signalling was dysregulated in old muscles.

Cell-type assignment of causal variant at inherited risk loci for sarcopenia

snATAC-seq datasets were correlated to GWAS variants for muscle strength and other phenotypes related to muscular disease or metabolic function. Lean body was enriched, perhaps as expected, in the snATAC-se landscape of type II myonuclei. However, and surprisingly, muscle strength-related traits were enriched in old tenocytes and FAPs. This important finding suggests that non-myogenic muscle resident cells, and associated microenvironment, may be important genetic contributors to sarcopenia. Interestingly, sleep duration, creatinine and fasting glucose were related to myofibers, suggesting a link between metabolism and circadian rhythms and the myofiber.

COMMENTS

The unique combination of human samples from different populations and sexes, depth of molecular analysis conducted at single-cell level, identification of not previously detected cell subsets, and description of gene programs and chromatin accessibility landscapes in young and old muscles make this an important and authoritative study

constituting a rich resource for the scientific communities interested in human muscle biology and human aging.”

Our reply: We thank the reviewer for our manuscript's positive and detailed summary.

“SUGGESTIONS

1. *Figure 1d, Figure 4j,m. To better clarify that the indicated data relate to old samples, “Age” may be replaced by “Aged”.*”

Our reply: Following this reviewer’s suggestion, we have revised this (see NEW Fig. 1d, 4b, 4f, 4i and 4l).

“2. *Extended Data Fig. 2C. Can sn/scRNA-seq and snATAC-seq datasets be quantitatively correlated?*”

Our reply: As suggested, we have added this information to the scale bar of the panel (see NEW Extended Data Fig. 2b).

“3. *Extended Data Fig.s 7, Figure 9, 10, 11, 12, 13, 14,17. Heatmaps of functional enrichment (panels a-e, “Min-Max” Extended Figure 7), DORC heatmap (Extended Figure 9 panel b); functional enrichment (Extended Figure 10 panel g); markers (Extended Figure 11 panel c), and all other heatmaps (Extended Figures 12,13,14,17) should be quantified.*”

Our reply: Following the reviewer’s suggestion, we have now quantified these heatmaps and other panels (see NEW Fig. 1d, 2e-f, 3c-d, 4b-c, 4f-g, 4i-j, 4l-m, 5b, 5e, 5g and 6b-c; and Extended Data Fig. 1d, 1f, 2b, 5a-c, 6a-b, 7b-c, 8c-e, 8i, 9a, 9c-d, 10a, 10c-d, 11a, 11c-d, 11f-g, 12a, 12d-e, 13b, 14a-e, 15a and 15c).

“4. *Figure 2a. Do the UMAPs represent both young and old snRNA-seq and snATAC-seq samples? If not, can the authors present them?*”

Our reply: The reviewer is right. The UMAPs in Figure 2a contained all the data from snRNA-seq and snATAC-seq for both age groups. This is now clearly stated in the corresponding figure legend (see figure legend for NEW Fig. 2a in the revised manuscript).

“5. *Figure 4e. Figure legends refer to left (old) and right (adult) panels. However, there are not left and right panels and Figure 4e illustrates TF binding site enrichment of up-(top) and down-(bottom)regulated genes.*”

Our reply: Thanks for spotting this. We have revised it in the revised manuscript (see figure legend for NEW Fig. 4d).

“6. *Page 16, lines 451, 457, 461. Figure 6 should be Figure 5”*

Our reply: Thanks for spotting this. Please noticed that we have reorganised the information of this part to make it concise in our revised manuscript (see page 17). We have also checked the figure calling throughout the manuscript.

“7. scATAC-seq footprinting analysis of few selected regulatory regions in myonuclei, MuSCs, and non-myogenic muscle resident cells would establish whether the enriched DNA binding motifs are indeed occupied by their cognate transcription factors.”

Our reply: The reviewer raises a good point. We have now performed footprinting analysis for FOSL2 and STAT3 in myonuclei (type I and II) along the pseudotime ageing trajectory and of JUNB and MYOG in MuSC subtypes, which confirmed our findings (see NEW Extended Data Fig. 7d and 8f). We have explained this information in the revised manuscript (see pages 11-12).

“8. Extended Data Fig. 3. The p-values of Pax7 and perilipin staining are both 0.02857. The authors may want to make sure that this is not an inadvertent duplication.”

Our reply: Thanks for realizing this. The reviewer is right. We have now amended it (see NEW Extended Data Fig. 3a and 3c).

Note:

To enhance the manuscript's clarity, we have made a revision regarding the annotation of "tenocytes." Instead, we have opted for a more encompassing term, "fibroblast-like." This decision was made considering that tenocytes are a relatively scarce population of stromal cells specifically found in tendons. It is improbable that we have captured a significant number of these cells in our study. Furthermore, from a transcriptional perspective, tenocytes exhibit numerous similarities with muscle fibroblasts, challenging their differentiation.

References:

1. Taillandier, D. & Polge, C. Skeletal muscle atrogenes: From rodent models to human pathologies. *Biochimie* 166, 251–269 (2019).
2. López-Otín, C., Blasco, M. A., Partridge, L., Serrano, M. & Kroemer, G. The Hallmarks of Aging. *Cell* 153, 1194–1217 (2013).
3. Moiseeva, V. *et al.* Senescence atlas reveals an aged-like inflamed niche that blunts muscle regeneration. *Nature* 613, 169–178 (2023).
4. Dungan, C. M. *et al.* In vivo analysis of γ H2AX+ cells in skeletal muscle from aged and obese humans. *FASEB J.* 34, 7018–7035 (2020).

5. Tumasian, R. A. *et al.* Skeletal muscle transcriptome in healthy aging. *Nat. Commun.* 12, 2014 (2021).
6. Migliavacca, E. *et al.* Mitochondrial oxidative capacity and NAD⁺ biosynthesis are reduced in human sarcopenia across ethnicities. *Nat Commun* 10, 5808 (2019).
7. Perez, K. *et al.* Single nuclei profiling identifies cell specific markers of skeletal muscle aging, frailty, and senescence. *Aging (Albany NY)* 14, 9393–9422 (2022).
8. Brink, S. C. van den *et al.* Single-cell sequencing reveals dissociation-induced gene expression in tissue subpopulations. *Nat. Methods* 14, 935–936 (2017).
9. Machado, L. *et al.* Tissue damage induces a conserved stress response that initiates quiescent muscle stem cell activation. *Cell Stem Cell* 28, 1125-1135.e7 (2021).
10. Machado, L. *et al.* In Situ Fixation Redefines Quiescence and Early Activation of Skeletal Muscle Stem Cells. *Cell Reports* 21, 1982–1993 (2017).
11. Hägg, S. & Jylhävä, J. Sex differences in biological aging with a focus on human studies. *Elife* 10, e63425 (2021).
12. DeTomaso, D. & Yosef, N. Hotspot identifies informative gene modules across modalities of single-cell genomics. *Cell Syst.* 12, 446-456.e9 (2021).
13. García-Prat, L. *et al.* FoxO maintains a genuine muscle stem-cell quiescent state until geriatric age. *Nat Cell Biol* 22, 1307–1318 (2020).
14. Dell’Orso, S. *et al.* Single cell analysis of adult mouse skeletal muscle stem cells in homeostatic and regenerative conditions. *Development* 146, dev174177 (2019).
15. Saggin, L., Gorza, L., Ausoni, S. & Schiaffino, S. Cardiac troponin T in developing, regenerating and denervated rat skeletal muscle. *Development* 110, 547–554 (1990).
16. Padova, M. D., Caretti, G., Zhao, P., Hoffman, E. P. & Sartorelli, V. MyoD Acetylation Influences Temporal Patterns of Skeletal Muscle Gene Expression*. *J Biol Chem* 282, 37650–37659 (2007).
17. Xu, Z. *et al.* Cardiac troponin T and fast skeletal muscle denervation in ageing. *J. Cachexia, Sarcopenia Muscle* 8, 808–823 (2017).
18. Rittoo, D., Jones, A., Lecky, B. & Neithercut, D. Elevation of Cardiac Troponin T, But Not Cardiac Troponin I, in Patients With Neuromuscular Diseases Implications for the Diagnosis of Myocardial Infarction. *J Am Coll Cardiol* 63, 2411–2420 (2014).
19. Zhang, T. *et al.* Cardiac troponin T and autoimmunity in skeletal muscle aging. *Geroscience* 44, 2025–2045 (2022).

20. Lin, I.-H. *et al.* Skeletal muscle in aged mice reveals extensive transformation of muscle gene expression. *Bmc Genet* 19, 55 (2018).
21. Weis, J. Jun, Fos, MyoD1, and Myogenin proteins are increased in skeletal muscle fiber nuclei after denervation. *Acta Neuropathol* 87, 63–70 (1994).
22. Choi, M.-C. *et al.* A Direct HDAC4-MAP Kinase Crosstalk Activates Muscle Atrophy Program. *Mol Cell* 47, 122–132 (2012).
23. Olivé, M. *et al.* Expression of myogenic regulatory factors (MRFs) in human neuromuscular disorders. *Neuropath Appl Neuro* 23, 475–482 (1997).
24. Iranzo, M. A. C. i *et al.* Functional and Clinical Characteristics for Predicting Sarcopenia in Institutionalised Older Adults: Identifying Tools for Clinical Screening. *Int J Environ Res Pu* 17, 4483 (2020).
25. Chiu, A.-F. *et al.* Barthel Index, but not Lawton and Brody instrumental activities of daily living scale associated with Sarcopenia among older men in a veterans' home in southern Taiwan. *Eur Geriatr Med* 11, 737–744 (2020).
26. Gong, G. *et al.* Correlation between the Charlson comorbidity index and skeletal muscle mass/physical performance in hospitalized older people potentially suffering from sarcopenia. *Bmc Geriatr* 19, 367 (2019).
27. Korthauer, K. *et al.* A practical guide to methods controlling false discoveries in computational biology. *Genome Biol* 20, 118 (2019).

Reviewer Reports on the First Revision:

Referees' comments:

** Please note that in my previous correspondence, you were sent Referee #1 comments to the editor in error. Referee #1 (Remarks to the Author) are as follows:

The authors have adequately responded to the points raised.

Referee #3 (Remarks to the Author):

The authors have addressed my comments. I suggest that if scRNA-seq and ATAC-seq were not performed in the same cell, then the title should be modified as multiomics is commonly employed for measurements simultaneously obtained from the same cell.

Referee #4 (Remarks to the Author):

Lai and co-authors present a single-cell/single-nucleus atlas of human limb skeletal muscles to investigate the impact of ageing on cell types, gene expression, and biological pathways. The dataset contains >387,000 single cells or single nuclei from 12 young adults (15-16 yro) and 19 old donors (74-99 yro) and with distinct fitness and frailty level. This dataset includes not only sc or snRNA-seq but also snATAC-seq to link epigenetic state to transcriptional cell types. This is a highly valuable atlas data resource. More important, they conducted extensive analysis of cell population changes during ageing and unveiled cell-specific and multicellular network features (at the transcriptomic and epigenetic levels) associated with these changes. Integration with scATAC-seq data, they could identify key elements of chromatin architecture associated with susceptibility to sarcopenia.

Since this is a revised manuscript, this reviewer took the liberty of reading the responses to previous review comments and found the revision has properly addressed all major questions. It seems the authors did take the comments seriously to improve the manuscript by adding more details in sample processing, better clarity, and some orthogonal validation experiments. In particular, single-cell atlas data are now available at an interactive data portal, which makes this highly valuable resource more accessible. Herein this reviewer provides some additional comments for the authors to consider.

(1) A main question is related to senescence. In the first submission, no senescence was mentioned at all although it is well known as an important mechanism in age-related tissue functional decline. In this revision, the authors checked the SA- β -gal staining and claimed bona fide senescent cells are hardly detected in unperturbed human muscles using senescence-associated beta-galactosidase (SA- β -gal) and CDKN2A staining and other markers like γ H2AX (all in Rebuttal figure 1). However, it appears that the authors have not done expensive senescence marker detection in the samples

analyzed in this work. In addition, senescence cells are known to be rare and large-area tissue staining and imaging is required to draw conclusions. The small region immunostaining may not be sufficient to make any conclusion.

(2) Along the same line of analysis, thanks for the online data portal provided (<https://db.cngb.org/cdcp/hlma/rnaseq/>), this reviewer checked several bona fide senescence markers and did observe significant, although rare, P16+ cells in scRNA-seq data. Other senescence markers are also detectable at even high levels and frequencies. Although these p16+ cells are rare, they are detectable in type I and II myonuclei. More importantly, they are relatively abundant in fibroblasts, muscle stem cells (MuSC), and interestingly a significant portion of lymphocytes. H2AFX, which was discussed in this paper, was not found to be expressed extensively in many non-myonuclei. In particular, these senescence marker expression in MuSCs is higher in aged donor samples, highlight some relevance of senescence in human muscle. Similarly, one of the key SASP factors IL6 is also expressed in fibroblast, MuSC and many types of vascular cells. Therefore, according to snRNA-seq data, there are a significant number of cells positive for senescence markers and they are enriched in cells from aged donors. Thus, it is somewhat confusing to claim senescence is not detectable.

(3) The dataset itself contains both RNA and ATAC, but largely presented as separate datasets with limited efforts to integrate or examine epigenetic mechanisms except some preliminary peak to gene linkage analysis. First of all, could the authors do co-embedding to better utilize both data types to identify stable cell populations. Second, could the authors check the temporal dynamics by examining the timing of chromatin accessibility vs gene expression in addition to the pseudo time trajectory.

(4) In fact, old-fashioned bulk RNA seq may not be able to more comprehensively show sarcopenia-associated genes since it can not resolve different cell types and the differential expression could be attributed to cell type difference or gene module difference. It is the unique value to identify cell type specific alterations between young and old or normal vs sarcopenia that can offer more accurate mechanistic insights. However, to better address this question, could the authors use top DEG to identify major cell populations, and then go back to all the detected genes to perform cell-type-specific differential analysis between young and old.

(5) as mentioned by the authors, sexual dimorphism in biological ageing is common in various tissues and organs. Although the authors found cell composition changes upon ageing in this work are comparable between male and female donors, within each population, there seem to be exist sex-dependent alterations, in particular, in FAP/Fibroblast and Type II cells. Any further analysis within these populations to examine sex as a biological variable.

(6) Although some orthogonal validation experiments have been done, overall, it is rather limited or not so informative. In particular, given that the majority of this work is single-cell sequencing data, any additional data from spatial gene expression or protein mapping may better validate the findings and also identify where these cells and their associated alterations occurs in ageing.

Author Rebuttals to First Revision:

Point-by-point response to the reviewers' comments

We thank the original three reviewers and the new reviewer #4 for their constructive comments that have helped us to improve our manuscript throughout the entire review process. We found that the points raised were very useful and have now performed new experiments and analyses and/or modified the descriptions in the text to address them. All major changes to answer these points are highlighted in red in the revised text. We hope that the reviewers will be satisfied with the revised manuscript and our detailed replies below.

REVIEWER 1:

"The authors have adequately responded to the points raised."

Our reply: We thank the reviewer for thoroughly assessing our manuscript.

REVIEWER 3:

"The authors have addressed my comments. I suggest that if scRNA-seq and ATAC-seq were not performed in the same cell, then the title should be modified as multiomics is commonly employed for measurements simultaneously obtained from the same cell."

Our reply: We thank the reviewer for thoroughly assessing our manuscript and for this very good additional suggestion. We have now modified the title of our manuscript to 'Multimodal cell atlas of the ageing human skeletal muscle' and referred to a multimodal rather than multiomics atlas elsewhere in the text (see for example **page 4**).

REVIEWER 4:

"Lai and co-authors present a single-cell/single-nucleus atlas of human limb skeletal muscles to investigate the impact of ageing on cell types, gene expression, and biological pathways. The dataset contains >387,000 single cells or single nuclei from 12 young adults (15-16 yro) and 19 old donors (74-99 yro) and with distinct fitness and frailty level. This dataset includes not only sc or snRNA-seq but also snATAC-seq to link epigenetic state to transcriptional cell types. This is a highly valuable atlas data resource. More important, they conducted extensive analysis of cell population changes during ageing and unveiled cell-specific and multicellular network features (at the transcriptomic and epigenetic levels) associated with these changes. Integration with scATAC-seq data, they could identify key elements of chromatin architecture associated with susceptibility to sarcopenia."

Our reply: We thank the reviewer for considering that our manuscript is a highly valuable data resource for the research community.

"Since this is a revised manuscript, this reviewer took the liberty of reading the responses to previous review comments and found the revision has properly addressed all major questions. It seems the authors did take the comments seriously to improve the manuscript by adding more details in sample processing, better clarity, and some orthogonal validation experiments. In particular, single-cell atlas data are now available at an interactive data portal, which makes this highly valuable resource more accessible. Herein this reviewer provides some additional comments for the authors to consider."

Our reply: We thank the reviewer for the thorough assessment of our manuscript and for considering that we have carefully addressed the comments from the previous round. For this reviewer's additional comments, our replies are shown below.

"(1) A main question is related to senescence. In the first submission, no senescence was mentioned at all although it is well known as an important mechanism in age-related tissue functional decline. In this revision, the authors checked the SA- β -gal staining and claimed bona fide senescent cells are hardly detected in unperturbed human muscles using senescence-associated beta-galactosidase (SA- β -gal) and CDKN2A staining and other markers like γ H2AX (all in Rebuttal figure 1). However, it appears that the authors have not done expensive senescence marker detection in the samples analyzed in this work. In addition, senescence cells are known to be rare and large-area tissue staining and imaging is required to draw conclusions. The small region immunostaining may not be sufficient to make any conclusion."

Our reply: We appreciate the reviewer's feedback regarding the further evaluation of senescent cells in human skeletal muscle tissue, which has helped us present our findings more clearly.

As suggested, we have now conducted more thorough tests for senescence markers in larger tissue areas. We carried out additional SA- β -gal staining on two separate sections for each sample. This was done for a panel of muscle biopsies from four adult/old donors, and we also included a section of injured murine muscle as a positive control for the staining. We provide a low-magnification view of the analyzed samples, highlighting a selected area. This information is included in NEW Extended Data Figure 1c, and we have explained it more clearly in our revised manuscript (see **page 4**). Moreover, we combined simultaneous SA- β -gal detection (using the fluorescent substrate SPiDER- β -gal) and DNA damage immunostaining with anti- γ H2AX, an additional method for evaluating senescence^{1,2} (see **REBUTTAL Figure 1**). For simplicity, we included only the SA- β -gal staining in our manuscript, but if the reviewer and/or editor consider it necessary, we will be happy to include this other assay in the supplementary information.

These two additional tests reinforce our previous findings: senescent cells are rarely detected by traditional SA- β -gal or SPiDER- β -gal/ γ H2AX analyses of uninjured muscle. Interestingly, however, in some sections from either adult or old donors, we identified rare areas of the

epimysium/perimysium containing isolated SA- β -gal⁺ or SPiDER- β -gal⁺ cells that serve as an internal positive control for staining (NEW Extended Data Figure 1c). Further investigation is needed to determine the impact of these rare connective tissue senescent cells on skeletal muscle ageing.

Moreover, since an aged skeletal muscle environment could facilitate sustained cellular damage, we acknowledge that cells in this environment might exhibit some isolated senescent cell-like characteristics defined *in vitro*, such as expression of cell cycle inhibitor genes¹, as supported by the limited expression (at the RNA level) of *CDKN1A* (p21) and *CDKN2A* (p16) (NEW Extended Data Figure 11h). We have explained this more clearly below in **point 2** and in our revised manuscript (see **page 20**). We hope that the reviewer will be satisfied with these explanations but will be happy to modify the text further or include more analyses if s/he and/or the editor consider it necessary.

REBUTTAL Figure 1.

a. Representative sections of injured mouse skeletal muscle (7-days post-cardiotoxin injury, 7 DPI, top), adult (middle) and old (bottom) human skeletal muscle stained with SPiDER- β -gal (green), anti- γ H2AX (red), anti-laminin (magenta, basement membrane) and DAPI (blue, nucleus). The border of the muscle fibre and the surrounding tissue is designated with the yellow dashed line,

and the SPiDER- β -gal⁺ or γ H2AX⁺ cells in the non-fibre tissue are marked with an arrow. Scale bar, 25 μ m.

b. Boxplot indicating the number of SPiDER- β -gal⁺ cells (left) and γ H2AX⁺ cells (right) per mm² in muscle fibre areas.

"(2) Along the same line of analysis, thanks for the online data portal provided (<https://db.cngb.org/cdcp/hlma/rnaseq/>), this reviewer checked several bona fide senescence markers and did observe significant, although rare, P16+ cells in scRNA-seq data. Other senescence markers are also detectable at even high levels and frequencies. Although these p16+ cells are rare, they are detectable in type I and II myonuclei. More importantly, they are relatively abundant in fibroblasts, muscle stem cells(MuSC), and interestingly a significant portion of lymphocytes. H2AFX, which was discussed in this paper, was not found to be expressed extensive in many non-myonuclei. In particular, these senescence marker expression in MuSCs is higher in aged donor samples, highlight some relevance of senescence in human muscle. Similarly, one of the key SASP factors IL6 is also expressed in fibroblast, MuSC and many types of vascular cells. Therefore, according to snRNA-seq data, there are a significant number of cells positive for senescence markers and they are enriched in cells from aged donors. Thus, it is somewhat confusing to claim senescence is not detectable."

Our reply: We thank the reviewer for carefully assessing known senescent markers in our dataset portal, highlighting the value of the collected data as a resource for the scientific community.

To address this matter, we have now checked the expression levels for a series of cell cycle inhibitor genes (*e.g.*, p21 [*CDKN1A*] and p16 [*CDKN2A*]), which are considered hallmarks defining cellular senescence *in vitro*¹. Mapping these genes in our data, we observed a variable expression pattern. In older individuals, the expression of p21 increased in most cell types (and particularly in non-myogenic subpopulations), whereas p16 expression was quite residual except for lymphocytes (see **REBUTTAL Figure 2**). The results for non-myogenic subpopulations are included in NEW Extended Data Figure 11h, and we have added an explanatory note in our revised manuscript (see **page 15**). Additionally, as the reviewer pointed out, IL6 also increased in old donors in endothelial cells, FAPs and fibroblast-like cells, albeit with low expression (see **REBUTTAL Figure 2**). This information was included previously; in this version, it is presented in the NEW Extended Data Figure 11h and discussed on **page 16**.

Notably, although these transcriptomic markers of cell cycle inhibitors and senescence-associated secretory phenotype can be detected in a low percentage of cells, these genes are expressed not only in senescent cells but also in other cellular conditions, such as inflammation or cellular stress^{3,4}. Nevertheless, p16⁺ cells are scattered in the UMAP representation in both myonuclei and mononucleated cells (see online portal), indicating that they have only minimal transcriptomic difference compared to non-expressing cells. This precludes, in our opinion, the prediction of *bona fide* senescent cells exclusively through transcriptomics. We have made this part clearer in our revised manuscript (see **page 4**). Yet, the reviewer's comment is excellent, and we have now explained in the Discussion section that these changes may facilitate the acquisition

of a more classical senescent-like phenotype under muscle damage in old individuals (see **page 20**).

REBUTTAL Figure 2.

Dot plots showing the expression levels for the representative cell-cycle inhibitor genes and *IL6*. The colour scale represents the average gene expression and dot size the percentage of cells expressing a given gene within the population.

Please see above our replies to **point 1** for further discussion on senescence.

"(3) The dataset itself contains both RNA and ATAC, but largely presented as separate datasets with limited efforts to integrate or examine epigenetic mechanisms except some preliminary peak to gene linkage analysis. First of all, could the authors do co-embedding to better utilize both data types to identify stable cell populations. Second, could the authors check the temporal dynamics by examining the timing of chromatin accessibility vs gene expression in addition to the pseudo time trajectory."

Our reply: We thank the reviewer for raising this point. The comment can be divided in two parts:

1) Perform co-embedding of sn/scRNA-seq and snATAC-seq

In response to the reviewer's suggestion, we conducted co-embedding analysis for myofibre, MuSC, immune cell, vascular cell, and stromal cell populations (see below, **REBUTTAL Figure 3**). While this revealed a good match in terms of identified cell types, we observed that co-embedding visualization could lead, in some cases, to a loss of resolution in terms of the identified subpopulations for given cell subtypes, such as FAPs or MuSCs, or to increased noise for populations with a low number of detected cells, such as immune cells. Additionally, the sequencing of 3-fold more nuclei/cells by sc/snRNA-seq reduced the contribution of snATAC-seq to the co-embedding visualization. Therefore, we opted not to include co-embedding visualization in the revised manuscript for the sake of clarity, but if the reviewer and/or editor consider it necessary we will be happy to include these panels as part of the supplementary

figures in the revised manuscript. Overall, our results demonstrate a good match between both types of data, which is also supported by the correlation heatmap analysis presented in the previous version (and here in the NEW Extended Data Figure 2b, 9d, 10d, and 11d).

Myofiber

MuSC

Vascular cell

Immune cell

Stromal cell

REBUTTAL Figure 3.

UMAP plot of integrated sc/snRNA-seq and snATAC-seq profiles in myofibre, MuSCs, vascular, immune, and stromal cell populations. Dots are coloured according to each annotated subpopulation (left panels) or omics modality (right panels).

2) Check the temporal dynamics of chromatin accessibility vs gene expression

Thanks for this great suggestion. In our previous manuscript, the temporal dynamics of myonuclei toward the ageing state were analyzed with two approaches, the second of which was not applied in the snATAC-seq dataset:

- i. A graph-based model, in which the profiled nuclei from the snRNA-seq and snATAC-seq datasets were combined and paired to generate the multimodal pseudotime trajectory. Further, the gene expression and the corresponding chromatin accessibility (aggregated as domains of regulatory chromatin) were projected along the trajectory. This revealed that stress-associated gene loci open their chromatin before the increase of gene expression, indicating a priming process that confirms a stepwise transition to overt myonucleus degeneration. This information was included previously and in this version in the NEW Extended Data Figure 7a-c.
- ii. A regression-based model in which the dataset from individual donors was aggregated to derive time-resolved summary statistics for inferring the age-associated changes. Using this approach, we analyzed the age-associated dysregulation of gene expression related to general muscle functions (*MYH1*, *MYH7*, *AMPD3*, *PCDHGA1*) in myofibres (see previous Extended Data Figure 6c). This analysis was not applied in the snATAC-seq, and we understood from the reviewer that we should do this. Thus, we aggregated the local chromatin accessibility (summed as gene score) in each individual for the corresponding genes in the myofibres, observing a similar trend as in the snRNA-seq but with higher variability. This information is included in NEW Extended Data Figure 6c, and we have revised the manuscript accordingly (see **pages 7 and 8**). The higher variability of chromatin accessibility may be attributed to a smaller number of profiled individuals in the snATAC-seq and/or the additional complexity of epigenetic regulation that involves multiple layers such as DNA or histone modifications. We have added a note for this in the discussion of the revised manuscript (see **page 20**).

"(4) In fact, old-fashioned bulk RNA seq may not be able to more comprehensively show sarcopenia-associated genes since it can not resolve different cell types and the differential expression could be attributed to cell type difference or gene module difference. It is the unique value to identify cell type specific alterations between young and old or normal vs sarcopenia that can offer more accurate mechanistic insights. However, to better address this question, could the authors use top DEG to identify major cell populations, and then go back to all the detected genes to perform cell-type-specific differential analysis between young and old."

Our reply: We thank the reviewer for agreeing about the advantages of single-cell omics technologies and the suggested analysis.

Following the reviewer's indication, we have performed the following: (1) Obtained DEGs from the publicly available bulk RNA-seq datasets of Perez *et al.* (GSE167186)⁵ and Migliavacca *et al.* (GSE111010)⁶, and (2) overlapped these DEGs with the cell type-specific genes in our study.

Overall, we detected only a limited number of DEGs (GSE111006 [frail vs healthy], 23; GSE111010 [frail vs healthy], 4; GSE167186 [frail vs old], 4; GSE167186 [old vs young], 215) in the bulk RNA-seq (see below, REBUTTAL Figure 4), which we attribute to the mixed cell composition. Moreover, after mapping the identified DEGs from the old vs young group in the GSE167186 dataset to each cell type of our study, we observed that a small subset of upregulated genes mostly corresponds to type I myofibres and (to a lesser extent) to FAPs. Moreover, cell type-specific downregulated genes were present in very low numbers (≤ 5) and correspond to type I/II myofibres and FAPs. Overall, these results highlight that gene expression levels measured by bulk RNA-seq are less sensitive than single-cell omics approaches at capturing DEGs belonging to minor cell populations.

REBUTTAL Figure 4.

Upper panel, upset plot comparing the list of DEGs obtained from different bulk RNA-seq datasets. Bottom panel, bar plots showing the number of cell type-specific DEGs from the bulk RNA-seq DEGs.

"(5) as mentioned by the authors, sexual dimorphism in biological ageing is common in various tissues and organs. Although the authors found cell composition changes upon ageing in this work are comparable between male and female donors, within each population, there seem to exist sex-dependent alterations, in particular, in FAP/Fibroblast and Type II cells. Any further analysis within these populations to examine sex as a biological variable."

Our reply: We thank the reviewer for raising this point. We understand that the reviewer is referring to the previous and NEW Extended Data Figure 2d. We have re-examined the

contribution of sex to the changes in cell composition in this panel using the snRNA-seq dataset, because it contains more individuals (see below, REBUTTAL Table 1). As shown in the table, two-way ANOVA analysis confirms the conclusions in previous and NEW Figure 1d that the changes in cell composition are highly related to age and not to sex (type II myofibre, $P = 0.0005$ in age, $P = 0.7090$ in sex; FAP, $P = 0.8250$ in age, $P = 0.9134$ in sex; Fibroblast-like cell, $P = 0.0105$ in age, $P = 0.3809$ in sex). Of note, the local true sign rate analysis in previous and NEW Figure 1d is more complete and considers not only the effects of sex and age but also ethnicity and technical covariates. We conclude that, based on our dataset, no major differences in cell populations are related to sex. However, we agree with the reviewer that this is a relevant question. We have a note in the Discussion (see page 20) explaining that future expansions of our HMA will help clarify this matter and further help refine our model.

2-way ANOVA test	% variation			P value		
	Age	Sex	Interaction	Age	Sex	Interaction
Type II	29.2	0.276	0.1204	0.0005	0.709	0.8052
FAPs	0.1441	0.03483	0.8686	0.825	0.9134	0.5879
Fibroblast-like cells	16.49	1.773	1.554	0.0105	0.3809	0.4118

REBUTTAL Table 1.

Description of the two-way ANOVA test performed to assess the contribution (% of variation) and statistical significance (P value) for the cellular composition changes in adult and old individuals (age factor) in males and females (sex factor).

"(6) Although some orthogonal validation experiments have been done, overall, it is rather limited or not so informative. In particular, given that the majority of this work is single-cell sequencing data, any additional data from spatial gene expression or protein mapping may better validate the findings and also identify where these cells and their associated alterations occurs in ageing."

Our reply: We acknowledge the reviewer's suggestion to incorporate spatial data to enhance the context and validation of our findings. While we agree that this could help refine our model, implementing spatial omics is a massive endeavor and would delay our manuscript's resubmission by several months. We discussed this with the editor, Dr. Therese Heemels, who indicated that although it is indeed an excellent suggestion, it would not be necessary to include spatial omics in our revised manuscript.

In our previous version, we provided multiple validations for age-related changes in cell type composition, myofibre type switching, myofibre subtypes with loss of sarcomere specification

(TNNT2⁺) or gain of signs of denervation (NCAM1⁺), early primed MuSC increase and cell communication changes (refer to Extended Data Figure 3, 4f, 5g, 8h, 13c, and Figure 2h, 5h). Although relatively limited in number, these validations, backed by adequate biological replicates, strongly support our findings.

Addressing the reviewer's concern within our time constraints (initial deadline of four weeks for resubmission), we have added a schematic to Extended Data Figure 16 illustrating the potential spatial distribution of age-related changes in muscle based on our findings. This addition is noted on page 20 of our revised manuscript. As mentioned in our previous and revised manuscript (see **page 20**), future HMA expansions will apply high-definition spatially resolved omics technologies relevant to understanding in more detail the impact of ageing on multicellular networks in the skeletal muscle niche. We hope that the reviewer will be satisfied with our explanations, and understand the limited time for resubmission considering that this is the third round of review.

References:

1. González-Gualda, E., Baker, A. G., Fruk, L. & Muñoz-Espín, D. A guide to assessing cellular senescence in vitro and in vivo. *FEBS J.* **288**, 56–80 (2021).
2. Moiseeva, V. *et al.* Senescence atlas reveals an aged-like inflamed niche that blunts muscle regeneration. *Nature* **613**, 169–178 (2023).
3. Beyfuss, K. & Hood, D. A. A systematic review of p53 regulation of oxidative stress in skeletal muscle. *Redox Rep.: Commun. Free Radic. Res.* **23**, 100–117 (2018).
4. Márton, M., Tihanyi, N., Gyulavári, P., Bánhegyi, G. & Kapuy, O. NRF2-regulated cell cycle arrest at early stage of oxidative stress response mechanism. *PLoS ONE* **13**, e0207949 (2018).
5. Perez, K. *et al.* Single nuclei profiling identifies cell specific markers of skeletal muscle aging, frailty, and senescence. *Aging (Albany NY)* **14**, 9393–9422 (2022).
6. Migliavacca, E. *et al.* Mitochondrial oxidative capacity and NAD⁺ biosynthesis are reduced in human sarcopenia across ethnicities. *Nat Commun* **10**, 5808 (2019).

Reviewer Reports on the Second Revision:

Referees' comments:

Referee #4 (Remarks to the Author):

The revised manuscript has nicely addressed the major questions and the additional supplementary figures 15-16 show quite interesting results in relation to senescence markers although this reviewer does understand the definition of bona fide senescent cells is still a topic of extensive on-going research interest. betaGal staining should be good enough to support the conclusions in this paper. Thanks! This reviewer would strongly recommend accepting this paper as it provides a highly valuable data resource to the community of aging research as well as some quite intriguing findings.

Author Rebuttals to Second Revision:

Point-by-point response to the referees' comments

Referee #4:

The revised manuscript has nicely addressed the major questions and the additional supplementary figures 15-16 show quite interesting results in relation to senescence markers although this reviewer does understand the definition of bona fide senescent cells is still a topic of extensive on-going research interest. betaGal staining should be good enough to support the conclusions in this paper. Thanks! This reviewer would strongly recommend accepting this paper as it provides a highly valuable data resource to the community of aging research as well as some quite intriguing findings.

Our reply: We thank the referee for thoroughly assessing our manuscript and for considering that our manuscript provides a highly valuable data resource to the ageing research community.